

# Global anthropogenic emissions of particulate matter including black carbon

Zbigniew Klimont[1], Kaarle Kupiainen[1,2], Chris Heyes[1], Pallav Purohit[1], Janusz Cofala[1], Peter Rafaj[1], Jens Borken-Kleefeld[1], Wolfgang Schöpp[1]

[1]International Institute for Applied Systems Analysis (IIASA), Laxenburg, 2361, Austria
[2]SYKE, Helsinki, Finland

*Correspondence to*: Zbigniew Klimont (klimont@iiasa.ac.at)

**Abstract.** This paper presents the first comprehensive assessment of historical (1990-2010) global anthropogenic particulate matter (PM) emissions including consistent and harmonized calculation of mass-based size distribution ($PM_1$, $PM_{2.5}$, $PM_{10}$)

as well as primary carbonaceous aerosols including black carbon (BC) and organic carbon (OC). The estimates were developed with the integrated assessment model GAINS, where source- and region-specific technology characteristics are explicitly included. This assessment includes a number of previously unaccounted or often misallocated emission sources, i.e., kerosene lamps, gas flaring, diesel generators, trash burning; some of them were reported in the past for selected regions or in the context of a particular pollutant or sector but not included as part of a total estimate. Spatially, emissions were

calculated for 170 source regions (as well as international shipping), presented for 25 global regions, and allocated to $0.5^\circ$ x $0.5^\circ$ longitude-latitude grids. No independent estimates of emissions from forest fires and savannah burning are provided and neither windblown dust nor unpaved roads emissions are included.

We estimate that global emissions of PM have not changed significantly between 1990 and 2010, showing a strong decoupling from the global increase in energy consumption and consequently, $CO_2$ emissions but there are significantly

different regional trends, with a particularly strong increase in East Asia and Africa and a strong decline in Europe, North America and Pacific. This in turn resulted in important changes in the spatial pattern of PM burden, e.g., European, North American, and Pacific contributions to global emissions dropped from nearly 30% in 1990 to well below 15% in 2010, while Asia's contribution grew from just over 50% to nearly 2/3 of the global total in 2010. For all considered PM species, Asian sources represented over 60% of the global anthropogenic total, and residential combustion was the most important sector

contributing about 60% for BC and OC, 45% for $PM_{2.5}$ and less than 40% for $PM_{10}$ where large combustion sources and industrial processes are equally important. Global anthropogenic emissions of BC were estimated at about 6.6 and 7.2 Tg in 2000 and 2010, respectively, and represent about 15% of $PM_{2.5}$ but for some sources reach nearly 50%, i.e., transport sector. Our global BC numbers are higher than previously published owing primarily to inclusion of new sources.

This PM estimate fills the gap in emission data and emission source characterization required in air quality and climate

modelling studies and health impact assessments at a regional and global level, as it includes both carbonaceous and non-carbonaceous constituents of primary particulate matter emissions. The developed emission data set has been used in several





regional and global atmospheric transport and climate model simulations within the ECLIPSE (Evaluating the Climate and Air Quality Impacts of Short-Lived Pollutants) project and beyond, serves better parameterization of the global integrated assessment models with respect to representation of black carbon and organic carbon emissions, and built a basis for recently published global particulate number estimates.

**1 Introduction**

Particulate matter (PM) or aerosols are solid and liquid particles small enough to remain airborne. PM can be directly emitted to the atmosphere (primary PM) or it can form from gaseous precursors (secondary PM). The size of PM stretches from clusters of molecules with a diameter of a few nanometers up to micrometer-sized abrasion products. This vast dimensional spectrum is reflected in the varying composition and characteristics of PM measured at source and receptor

sites. PM species are important constituents of the atmosphere and they play a role in the earth's climate system. Some PM species, i.e. black carbon, absorb visible light and warm the atmosphere, whereas other species, i.e., sulphates and organics reflect sunlight back to space and cool the climate (Bond et al., 2013). PM also serve as condensation nuclei for water vapour to eventually form cloud droplets. There is well-documented evidence that exposure to PM results in adverse effects on human health (e.g., Anenberg et al., 2012; Lim et al., 2012; WHO, 2004).

Integrated assessment models, such as the GAINS (Greenhouse gas - Air pollution Interactions and Synergies) model (Amann et al., 2011), utilize data on economic development and corresponding emissions, estimate atmospheric concentrations and further assess the impacts on climate, human health and ecosystems. When this information is combined with potentials and costs for controlling the emissions it is possible to study the cost-efficiency of different policies to reduce the undesirable effects and meet environmental objectives on climate, human health and ecosystem impacts. Such an

integrated modelling framework is particularly important for assessing the impacts of particulate matter owing to the multitude of sources, including primary and secondary, and effects on health and climate. All these aspects of PM call for consistent data to support the assessments of impacts and potential for formulating robust strategies to reduce emissions together with consequent concentrations and impacts.

This paper presents the first comprehensive assessment of historical (1990-2010) global anthropogenic particulate matter

(PM) emissions including consistent and harmonized calculation of mass-based size distribution ($PM_1$, $PM_{2.5}$, $PM_{10}$) as well as primary carbonaceous aerosols; black carbon (BC) and organic carbon (OC). The methodology draws on the earlier developed structure of the PM module in GAINS (Klimont et al., 2002b; Kupiainen and Klimont, 2004, 2007) but was extended to include new information as well as previously unaccounted sources, i.e., gas flaring, kerosene lamps, diesel generators.

A recent GAINS model development includes extension to include particulate number (PN) emissions (Paasonen et al., 2013). This builds on the emission methodology and estimates described in this paper making use of one of the datasets





(ECLIPSE *V5*) to calculate past and future PN emissions and their spatial distribution. The respective documentation and discussion paper is in review (Paasonen et al., 2016).

While the results presented in this paper focus on the outcomes included in the ECLIPSE *V5a* version of data, there were several datasets developed within the ECLIPSE project[1] (Stohl et al., 2015) and the key differences between the datasets are also briefly discussed. Table 1 gives an overview of the datasets that are accessible from the GAINS website[2]; the paper describing the projections is in review for this issue of ACP (Klimont et al., in preparation).

## 2 Method

The ECLIPSE emission data set was created with the GAINS (Greenhouse gas – Air pollution Interactions and Synergies; http://gains.iiasa.ac.at) model (Amann et al., 2011), which calculates emissions of air pollutants and Kyoto greenhouse gases (GHG; i.e., carbon dioxide ($CO_2$,), methane ($CH_4$), nitrous oxide ($N_2O$) and the three F-gases) in a consistent framework. The GAINS model holds essential information about key sources of emissions, environmental policies, and further mitigation opportunities for 170 country-regions. The model relies on international and national statistics of activity data for energy use, industrial production, and agricultural activities (see section 3) for which it distinguishes all key emission sources and control measures. Several hundred technologies to control air pollutant and greenhouse gases emissions are represented allowing simulation of implemented air quality legislation (see section 2.3).

Since previous work (Cofala et al., 2007; Klimont et al., 2002b, 2009, Kupiainen and Klimont, 2004, 2007; Shindell et al., 2012) we have reviewed recent literature, including non-peer reviewed studies, to improve characterization of the source sectors and control technologies in the GAINS model, update the assumptions about penetration of control measures, and to include previously unaccounted or poorly allocated sources. Emission sources that have been recently added or for which the emission calculation has been refined include: flaring of associated petroleum gas in the oil and gas exploration sectors, kerosene lamps for lighting (further development of estimates originally presented by Lam et al. (2012)), diesel generator sets, high emitting vehicles, international shipping, trash burning as well as brick kilns (see section 3).

Further improvements in the emission model have been made especially for China (Klimont et al., 2013; Wang et al., 2014; Zhao et al., 2013), where large changes have occurred recently as well as new data becoming available, but also for Europe where results of the consultation with national experts during the review of the EU National Emission Ceilings Directive were considered in the last datasets (Amann et al., 2015). Finally, the regional resolution of the global GAINS model has been improved by distinguishing more countries in Latin America where five regions (Argentina, Brazil, Chile, Mexico, all remaining Latin America) were replaced with 13 regions in version *V5a*, including most countries of South America,

---

[1] European Commission FP7 project ECLIPSE (Evaluating the Climate and Air Quality Impacts of Short-Lived Pollutants); Project no. 282688; http://eclipse.nilu.no
[2] http://www.iiasa.ac.at/web/home/research/researchPrograms/air/Global_emissions.html




Mexico, Central America, and the Caribbean; a full list of country-regions in the global GAINS application is included in the supporting information (SI).

**2.1 PM estimation method**

The methodology to derive particulate matter (PM) emission factors and calculate emissions relies on the methods documented in (Klimont et al., 2002b; Kupiainen and Klimont, 2004, 2007). However, apart from updates to emission factors a number of modifications and extensions have been introduced since, especially for carbonaceous particles. We summarize the principles below allocating more space to discuss extensions.

The emissions of PM in the GAINS model are calculated for several size classes: submicron fraction (particles with diameter smaller than 1 μm; $\leq PM_1$), fine fraction ($\leq PM_{2.5}$), coarse fraction ($PM_{2.5} > PM_{10}$), and large particles ($\geq PM_{10}$). $PM_{10}$ is calculated as the sum of fine and coarse fractions. Total suspended particles (TSP) as the sum of fine, coarse, and $\geq PM_{10}$ fractions. Additionally, black carbon (BC) and organic carbon (OC) are calculated.

The methodology includes the following steps:

(i) region- ($i$), sector- ($j$) and fuel- ($k$) specific "raw gas = unabated" emission factors for total suspended particles (TSP) are derived. For solid fuels (excluding biomass and use of solid fuels in small residential installations) the mass balance approach is used where ash content ($ac$) and heat value ($hv$) of fuels, and ash retention in boilers ($ar$) for given combustion technologies are considered Eq. (1):

$$ef(TSP)_{i,j,k} = \frac{ac_{i,j,k}}{hv_{i,j,k}}(1 - ar_{j,k}) \tag{1}$$

For liquid fuels, biomass, solid fuels used in small residential installations, industrial processes, mining, storage and handling of bulk materials, waste incineration, agriculture[3], and transport, TSP emission factors are taken from the literature;

(ii) considering fuel- and sector-specific size fraction profiles reported in the literature, "raw gas" emission factors for each of the size fractions and carbonaceous species are estimated;

(iii) the emission factors for organic carbon (OC), calculated in the previous step, are adjusted considering carbonaceous fraction in $PM_{2.5}$ and organic carbon (OM); see section 2.1.1 for discussion;

(iv) PM emissions are calculated for each size fraction and carbonaceous species applying the following equation Eq. (2), where also the application rates of control technologies ($X$) and size fraction specific emission removal efficiencies ($eff$) are taken into account:

$$E_{i,y} = \sum_{j,k,m} E_{i,j,k,m,y} = \sum_{j,k,m} A_{i,j,k} ef_{i,j,k,y}(1 - eff_{m,y}) X_{i,j,k,m}, \tag{2}$$

where $i,j,k,m$ are region, sector, fuel, abatement technology; $y$ size fraction, i.e., fine, coarse, PM_>10, or carbonaceous species (BC, OC); $E_{i,y}$ emissions in region $i$ for size fraction $y$; $A$ activity in a given sector, e.g., coal consumption in power

---

[3] For livestock, emission factors refer to housing period and therefore, information on the length of this period (one of the parameters in the GAINS model) is considered to derive annual animal- and country-specific values.



plants; *ef* "raw gas" emission factor; $eff_{m,y}$ reduction efficiency of the abatement option *m* for size fraction *y*, and *X* actual implementation rate of the considered abatement, e.g., percent of total coal used in power plants that are equipped with electrostatic precipitators. If no emission controls are applied, the abatement efficiency equals zero ($eff_{m,y} = 0$) and the application rate is one ($X = 1$). In that case, the emission calculation is reduced to simple multiplication of activity rate by the "raw gas" emission factor.

There are a few source sectors where additional assumptions are made in order to develop emission factors used in the calculation. Specifically, for gas flaring additional information about the composition of associated gas is used (see section 3.6.3 for more details), and to estimate emissions from high emitting vehicles (or superemitters) assumptions about region-specific shares of high emitters as well as technology and pollutant specific increments, compared to the average fleet emissions factors (excluding high emitters), are made (see section 3.4.1).

### 2.1.1 Adjustments of carbonaceous particle emission factors

While we principally follow the definition of black carbon (BC) given by Bond et al. (2013), i.e., *"...a distinct type of carbonaceous material that is formed primarily in flames, is directly emitted to the atmosphere, and has a unique combination of physical properties. It strongly absorbs visible light, is refractory with a vaporization temperature near 4000K, exists as an aggregate of small spheres, and is insoluble in water and common organic solvents"*, the available measurement studies have not been consistent in this respect, and it has not been possible to systematically follow the definition in developing the input data for emission estimates; this has also been discussed in our previous papers (Kupiainen and Klimont, 2004, 2007).

Organic carbon (OC) refers to the carbon fraction in numerous organic compounds that contain hydrogen and, usually, oxygen, and are emitted to the air as particles (Bond et al., 2013). To attain the total mass associated with the organic compounds, organic matter (OM), OC needs to be multiplied with a fraction that depends on the suite of compounds emitted and varies between the emission sources. We introduce source specific OM to OC fractions for primary emissions found from literature, varying between 1.3 and 2.1 (Aiken et al., 2008; Tissari et al., 2007; Turpin and Lim, 2001). Due to the lack of a formal definition and available measurement studies we have not attempted so far to separate emissions of "brown carbon", a group of absorbing compounds considered a subset of organic aerosol (Bond et al., 2013).

Emission factors of organic carbon ($ef_{OC}$) for each GAINS technology category are calculated using a mass balance equation Eq. (3). This equation has been introduced to ensure that the mass balance of the chemical species of particulate matter (black carbon and organic carbon) will still stay within physical limits of the PM mass metrics applied in GAINS. The calculation uses $PM_{2.5}$ as the limiting mass metric since the emissions of carbonaceous matter occur primarily in that size range. We introduce only few exceptions where larger carbonaceous particles are expected to be present, e.g., tyre wear.

$$ef_{OC} = (ef_{PM2.5} \times f_{carb} - ef_{BC}) \div f_{OM}, \tag{3}$$

where $f_{carb}$ is the mass fraction of the total carbonaceous matter, or black carbon and organic matter, in $PM_{2.5}$, $f_{OM}$ the average organic molecular weight per carbon weight in particular matter, $ef_{BC}$ the emission factor of BC, $ef_{PM2.5}$ the emission factor of





$PM_{2.5}$. Emission factors of BC and $PM_{2.5}$ as well as $f_{carb}$ and $f_{OM}$ are estimated based on emission measurement data. The final set of OC emission factors is checked for consistency with emission measurements.

Fraction of carbonaceous matter in $PM_{2.5}$ ($f_{carb}$) varies significantly between source sectors. Highest fractions are usually found in residential combustion and transport sectors in technologies with poor combustion, where over 90 percent of the

particulate matter is estimated to consist of carbonaceous matter. As the combustion process becomes more efficient and optimized, the fraction reduces drastically and, for example, in large modern power plants, which have optimized combustion processes and efficient air pollution abatement technologies, the fraction is typically negligible; see discussion in Kupiainen and Klimont (2007) and Sippula et al. (2009).

The average fraction of organic molecular weight per carbon weight ($f_{OM}$) varies also between different emission source

sectors and fuels. For combustion of biomass, including wood, we use $f_{OM} = 1.8$, which represents approximately the middle of the range (1.6 to 2.1) of $f_{OM}$ values available for combustion of different wood species in the literature (Aiken et al., 2008; Tissari et al., 2007; Turpin and Lim, 2001). For diesel and gasoline in transport sector, we use $f_{OM} = 1.3$, based on Aiken et al. (2008).

## 2.2 Model technology resolution

The GAINS model structure includes representation of key emission sources compatible with global and regional emission inventories but the calculation often distinguishes an additional level of detail where combustion technology (e.g., pulverized coal or grate firing boilers, fireplaces, various stoves, pellet boilers, etc.) as well as emission control technology (e.g., wet scrubbers, fabric filters, fan assisted stoves, diesel particulate filters, etc.) are explicitly distinguished (see also Eq. (2)). Such an approach has been an integral part of the GAINS model development for both particulate matter (e.g., Klimont et al.,

2002b; Lükewille et al., 2001) and other pollutants (e.g., Amann et al., 2011; Cofala and Syri, 1998; Klimont et al., 2002a); the details for PM are documented in Klimont et al. (2002b) and the current structure can be reviewed in the on-line application of the GAINS model[4]. This approach has also been used in other emission assessment studies and often referred to as 'technology-based' (e.g., Bond et al., 2004; Lu et al., 2011; Zhao et al., 2013).

Implementation of such technology resolution requires additional assumptions about the shares of activity in a given sector

falling into each subcategory and share of activity controlled with a specific mitigation measure. The following sections highlight and document briefly the assumptions for key sectors.

### 2.2.1 Residential combustion: cooking, heating, lighting

GAINS divides residential-commercial sector into several fuel–dependent categories (Table 2). The division is driven by varying emission characteristics and available control options (Table 3). While such a structure is fairly compatible with the

available emission measurements (see section 3.1), it is challenging to distribute fuel consumption into these categories as

---

[4] http://gains.iiasa.ac.at; select any of the accessible regional version to view the model structure





typically statistical data is available either as total residential sector or split into commercial/residential/other (e.g., IEA, 2015a, 2015b). We rely on a mix of sources and our own assessment to derive the respective shares. There have been several assessments at a global level where either total fuel demand for cooking and heating, allocation between various fuels, or stove types was attempted (Bonjour et al., 2013; Chafe et al., 2014; Fernandes et al., 2007). For Europe, such data are not

readily available; however, within the work on revision of air quality legislation we were involved in several rounds of stakeholder consultations where national experts representing various sectors reviewed GAINS assumptions (Amann et al., 2015) and all data can be viewed in the on-line model. For the US and Canada, a similar discussion and exchange took place within the work of the Arctic Council where the GAINS model was used to develop unified emissions and scenarios (AMAP, 2015). For Australia and New Zealand a number of local studies were used (Driscoll et al., 2000; Scott, 2005;

Todd, 2003).

The allocation of fuel between various categories varies between Europe, North America, and OECD Asia and Pacific where most solid fuel is used for heating (e.g., Chafe et al., 2015), and most of Asia, Africa and Latin America where cooking is the primary use. Consequently, nearly all solid fuels in South Asia, Africa, and Latin America are allocated to cooking stoves. For Asia, we draw on the past and ongoing collaboration on the development of the GAINS-Asia model (Amann et al., 2008;

Klimont et al., 2009; Zhang et al., 2006; Zhao et al., 2013) where assumptions on the split between heating and cooking as well as fuel used in medium size boilers were made, as well as several peer-reviewed publications (e.g., Venkataraman et al., 2010). For Latin America, information about this sector structure originates from the discussions with the authors of various assessments of effectiveness of clean cooking programs (e.g., Pine et al., 2011; Ruiz-Mercado et al., 2011).

The GAINS model includes a number of mitigation measures in this sector (Table 3), although some of them might be seen

more as different types of installations, e.g., various stove types already in place (for specific discussion of their assumed characteristics see Supplementary Information – section S2). While there has not been a lot of success in sustained replacement of traditional stoves with improved clean burning stoves (e.g., Foell et al., 2011; Pine et al., 2011; Ruiz-Mercado et al., 2011; Wickramasinghe, 2011), it is important to consider the varying level of implementation across the regions if such information is available. As with the allocation of fuel use (see discussion above), we rely on data and

assessments collected within several bilateral projects (e.g., Amann et al., 2008, 2015), peer-reviewed papers (e.g., Klimont et al., 2009; Lewis and Pattanayak, 2012; Li et al., 2016; Pine et al., 2011; Ruiz-Mercado et al., 2011; Shrimali et al., 2011; Silk et al., 2012; Troncoso et al., 2011), and published reports (Adria and Bethge, 2013; Germain et al., 2008; Scott, 2005; Todd, 2003).

One of the recent developments in the GAINS model was explicit distinction of use of kerosene for cooking and lighting

(Table 2); earlier all kerosene was allocated to cooking. This modification was driven by the study highlighting the potentially high contribution of kerosene lamps to black carbon emissions (Lam et al., 2012). The emissions depend on what type of lamp is used, and for historical data we distinguish between wick and hurricane lamps, with the former representing the majority (Lam et al., 2012; Mills, 2005). As a default we assume 80% kerosene wick lamps in South Asia and 50% in other developing world regions. For discussion of how total activity data for kerosene lighting is calculated see section 3.2.





### 2.2.2 Transport

The GAINS model distinguishes several source categories within the road and non-road transport sectors. Road transport is disaggregated into six vehicle categories: 2- and 4-stroke two-wheelers, passenger cars and vans, light duty vehicles, heavy duty trucks, and buses. The non-road mobile sources are grouped into nine broad categories: agriculture and forestry,

construction and mining, rail, inland navigation, coastal shipping, aviation (only landing and take-off), 2-stroke engines (e.g., in households, recreation, forestry, etc.), other land based engines. Each vehicle/machine category is associated with a fuel according to its propulsion type; several fuels are distinguished: diesel, gasoline, CNG, LPG, jet fuel or kerosene, heavy fuel oil as well as hydrogen and electricity. For each of the fuel-vehicle combinations, activity data (fuel consumption and km-driven for road vehicles) are sought and are usually available in national and international statistics for road transport

categories while are often incomplete, allocated under other sectors, or even lacking for non-road sources. For a complete list of transport sources and fuels see Table S8.1.

In order to reflect existing legislation, each fuel-vehicle combination is further subdivided by its average emission level. The key proxy for the emission level is the exhaust emission legislation in force in the country (or region) at the time when the vehicle type is put into service or to which level it is retrofitted. The associated emission factors describe the emission rates

for the pollutants averaged over the actual operating conditions, vehicle sizes, and machine types as well ages and model years within one emission standard. More details about the emission factors, control stages in GAINS, and discussion of high emitting vehicles are provided in section 3.4.

### 2.2.3 Large scale industrial combustion

The available statistical data allows for allocation of fuel into key sectors, like power plants and industrial boilers, but owing

to varying emission characteristics and often different legislation for different boiler types, the GAINS model distinguishes additionally a number of selected plant and boiler types (for more background discussion see Klimont et al. (2002b)). Specifically, the power sector is divided into existing (constructed before 2005), new and modern plants for which additionally large and small plants (grate firing) are distinguished. Industrial combustion is associated with several sectors for which also small boilers are included to capture the large numbers of often old and poorly controlled solid fuel grate

firing boilers in the developing countries (e.g., Wang et al., 2014; Zhao et al., 2013), for example in China they accounted for about 85% of all industrial boilers (Wang et al., 2009).

Finally, the GAINS model structure has been extended to distinguish diesel generator sets; previous GAINS regional and global assessments of PM or carbonaceous particles (Cofala et al., 2007; Klimont et al., 2009; Kupiainen and Klimont, 2007) included their fuel consumption in power and residential combustion sectors. The new structure allows for better

representation of emissions and mitigation opportunities, especially in regions with low reliability of electricity supply and poor emission standards, e.g., South Asia. The estimates of regional diesel generators fuel use is discussed in section 3.3.



### 2.2.4 Industrial processes

Most industrial processes are sources of particulate matter emissions. For the majority of them emissions are calculated using total production volumes without distinguishing specific stages of the processing chain. However, for a number of manufacturing processes we define default plant profile and distinguish between process and fugitive emissions, for details see Klimont et al. (2002). Additionally, for selected industries a more detailed structure was designed to reflect the significant differences between types of plants (kilns); this has been done for cement, coke, and brick manufacturing.

The key driver behind the extended structure for cement and coke manufacturing was developments in China where in the last decades strong growth resulted in often rapid transformation of the two sectors. For cement production rotary kilns with precalciner and shaft kilns are distinguished for which the activity split has been developed in collaboration with Tsinghua University (Zhao et al., 2013). The coke production sector in China experiences rapid transformation from traditional ovens to mechanized integrated coke ovens which have different emission characteristics; the changes in the structure of the sector are discussed by Huo et al. (2012). Currently, the information about the comparable technology split is not available for other countries for which emissions are calculated without such distinction.

*Brick manufacturing*

There are strong regional differences in brick manufacturing sector structure that is especially relevant in the developing world where a large share of the market is occupied by traditional, heavy-polluting kilns. Our earlier work focused on characterizing the brick sector in Asia, by far the largest producer, and therefore the distinguished kiln types reflected practices in Asia (Klimont et al., 2009; UNEP/WMO, 2011). However, such design of the model did not allow to correctly address the structure of this sector in other regions like Africa or Latin America and the Caribbean. We have reviewed regional and national assessment studies to identify typical regional profiles (distribution of production by kiln types) of the brick manufacturing sector, including also typically used fuels; such profiles change over time and it has been considered where such information was found. Table 4 shows the kiln structure included in GAINS and highlights key representative technologies assumed for different world regions. The overview of studies used to develop the respective assumptions is provided in the SI (S5). The overall brick production data are discussed in section 3.6.2 and SI (Table S5.2).

### 2.3 Emission legislation

We have collected information about existing international and national requirements with respect to emission limit values for stationary and mobile sources and estimated control technology implementation rates required to achieve respective standards in all GAINS regions. The interpretation of the laws and translation into the set of GAINS technologies with the associated emission rates under average operating conditions has been discussed previously in a number of papers and assessments addressing regional (Amann et al., 2015; Klimont et al., 2009; Kupiainen and Klimont, 2007; Wang et al., 2014) and global (Amann et al., 2013; Cofala et al., 2007; Rao et al., 2013; Riahi et al., 2012; UNEP/WMO, 2011) emissions.



For a number of sources there exist global databases summarizing current laws and emission limit values, including power plants (IEA, 1997; IEA CCC, 2012), transport (Delphi Inc., 2013, 2015; ICCT & Dieselnet, 2014), and the cement industry (Edwards, 2014). Additionally, specific regional and national laws and policy implementation studies were reviewed, i.e., for the European Union a number of Directives was considered (Crippa et al., 2016; EC, 2001a, 2001b, 2010; Krasenbrink and

Dobranskyte-Niskota, 2008), for Asia several peer-reviewed studies (Goel and Guttikunda, 2015; Guttikunda and Jawahar, 2014; Huo et al., 2011, 2012; Klimont et al., 2009; Liu et al., 2015; Lu et al., 2011; Wang et al., 2014; Zhang et al., 2006) as well as other sources (CAI-Asia, 2011; CPCB, 2007; IIDFC, 2009); for Latin America and Caribbean additional information was obtained for the brick sector (e.g., Stratus Consulting, 2014) and for Argentina, Brazil, and Mexico also for the transport sector (e.g., Ministério do Meio Ambiente, 2011).

In the course of development of the several ECLIPSE datasets, the legislation information and mostly the rates of enforcement and implementation of actual measures have been revisited. The key updates in version *V4a* (see Table 1) include consideration of the initial round of consultations with European Union member states experts within the review of the National Emission Ceiling (NEC) directive (Amann et al., 2012), which included comparison of GAINS estimates with the emissions officially reported to the Centre on Emission Inventories and Projections (CEIP; www.ceip.at) under the

Convention on Long-range Transboundary Air Pollution. A much more substantial update came with version *V5a* where for China the 12th Five Year Plan policies were introduced, resulting in revision of the implementation and enforcement rates of control measures for 2010 drawing also on analysis of progress in legislation implementation in China (e.g., Lin et al., 2010; Zhang et al., 2015). Furthermore, the legislation for cement industry was reviewed and updated (Edwards, 2014), emissions from international shipping were also calculated, the treatment of the non-road mobile machines were reviewed, and for

Latin America and Caribbean (LAC) the GAINS model has been revised to include nearly all single countries [5] and consequently required definition of control strategies reflecting current legislation for each country. Finally, also for the European Union an update was performed in *V5a* to include the latest status of discussion with the national experts (Amann et al., 2015), as well as new submissions of $PM_{2.5}$ emissions (also for the past years) to CEIP, especially for 2010.

## 2.4 Spatial and temporal distribution

The GAINS model calculation is performed for 170 regions globally and for Europe and Asia the calculation and results are directly available by country or even subnational level from the online version of the model (http://magcat.iiasa.ac.at) for all ECLIPSE data sets. At a global level, the emissions and activity data are available online at the resolution of 25 global regions (see Supporting Information (SI), S7) and key sources (http://magcat.iiasa.ac.at/gains/IAM/index.login); the structure is compatible with most of the global integrated assessment models. Additionally, the total annual emissions were gridded

and temporal (monthly) distributions were developed.

---

[5] Previous versions included five regions: Argentina, Brazil, Chile, Mexico, other LAC



The GAINS particulate matter emissions were distributed into $0.5^{o}$ x $0.5^{o}$ longitude-latitude grids and stored in *netCDF* format files available from http://www.iiasa.ac.at/web/home/research/researchPrograms/air/Global_emissions.html as well as from the ECLIPSE project web: http://eclipse.nilu.no. The files contain several layers (Table 5), reflecting key sectors (consistent with Representative Concentration Pathways (RCP) used in the Intergovernmental Panel for Climate Change

Fifth Assessment Report (IPCC AR5)), and a total emission layer. The spatial distribution was prepared from RCP-consistent proxies as used and further developed within the Global Energy Assessment project (GEA, 2012). These are in line with proxies applied within the RCP projections as described in Lamarque et al. (2010) and were modified to accommodate more recent information where available, e.g., population distribution, open biomass burning, effectively making them year-specific (Klimont et al., 2013; Riahi et al., 2012).

In the process of preparing gridded emissions we have developed additional layers which were merged into the sector layers listed in Table 5. The primary example, relevant for particulate matter emissions, is the flaring layer which has been developed by IIASA using the information on flare location areas developed in the collaborative project of NOAA, NASA, and the World Bank (Elvidge et al., 2009, 2011). This layer contains emissions from flaring in oil/gas exploration and it is for the first time that a global PM emission assessment includes this source with explicit spatial allocation (Fig. 1); this

dataset was used within the ECLIPSE project and highlighted the relevance of proper distribution of black carbon emissions from this source (Stohl et al., 2013). The flaring emissions are integrated in the *Energy* (Table 5) layer but a separate file with all emissions from flaring only is also available for download.

### 2.4.1 Temporal distribution

The GAINS model does not explicitly include any assumptions about temporal distribution and therefore all emissions are

calculated as annual totals. However, within the MACEB[6] and ECLIPSE projects we have developed monthly emission profiles for the gridded output, shares of emissions in different months in each grid, for a number of sources. The focus was on allocation of domestic heating and cooking emissions where the methodology combines the stove use assumptions from Streets et al. (2003) with the global gridded temperature fields from the CRU3.0 archive[7] of monthly mean temperatures (Brohan et al., 2006). The shares were developed for six years (2000-2006) and an average was eventually used as a

representative monthly fraction. Fig. S1 in SI compares this pattern with other existing estimates for selected countries. The importance of considering the temporal distribution of residential combustion emissions developed within ECLIPSE has been demonstrated in Stohl et al. (2013) for the Arctic.

For the energy sector, country-specific monthly patterns were created for selected regions based on available data; for Europe and Russia such data were originally developed in the GENEMIS project (Ebel et al., 1997) and are readily available

in the EMEP database; for North America we used the US-EPA Clearinghouse for Emission Inventories

---

[6] MACEB - **M**itigation of **A**rctic warming by **C**ontrolling **E**uropean **B**lack carbon emissions, European Union Life+ project no: LIFE09 ENV FI 572
[7] http://badc.nerc.ac.uk/data/cru/



(http://www.epa.gov/ttnchie1/emch/temporal/) and the US Energy Information Agency Monthly Energy Review (http://www.eia.gov/totalenergy/data/monthly/); for Thailand the information provided by (Vongmahadlek et al., 2008, 2009) was applied. For all other regions, the temporal distribution file includes constant emissions across the year.

5 The emissions from open burning of agricultural residues are seasonal since the activity is related to growing cycles and harvesting of different crop types. A global spatial and temporal representation was developed based on the timing and location of active fires on agricultural land in the Global Fire Database GFEDv3.1 (http://www.globalfiredata.org/Data/index.html) combined with annual emissions from GAINS. All active grid cells ($0.5^{o}$ x $0.5^{o}$) in the monthly data from 1997 to 2010 in GFED were summed up and normalized. Also for other agricultural activities several patterns were developed but they are more relevant for ammonia and methane emissions and therefore 10 discussed in Klimont et al. (in preparation).

## 3 Emission sources – activity data and emission factors

Here we highlight the contribution of key sources to total emissions and document the sources of activity data and emission factors used in the GAINS model for all relevant sources of particulate matter (PM) emissions, including discussion of differences between several published ECLIPSE datasets. The technology splits and air pollution legislation is discussed in 15 section 2.2 and 2.3.

The basic statistical data for energy consumption, industrial output, and agriculture originates from International Energy Agency (IEA, 2015a, 2015b), Eurostat (EUROSTAT, 2011), UN Food and Agriculture Organization (http://faostat.fao.org), and several national sources that have been used in the course of collaboration with several partners in Europe (e.g., Amann et al., 2012, 2015) and Asia (e.g., Amann et al., 2008; Purohit et al., 2010; Zhang et al., 2006; Zhao et al., 2013). For several 20 sectors more specific regional data were used; see discussion in the following source-specific sections. There are also differences in data used for various versions of the ECLIPSE dataset; an overview is provided in Table 1. For activity data, the most significant changes are due to update of the historical data in version *V5* and *V5a* where all IEA statistical data was imported at national level and processed for use in GAINS. Furthermore, for Europe the consultations with national experts during the National Emission Ceiling Directive (NEC) revision process led to a number of updates (including activity, 25 emission factors, penetration of control technologies) for the EU-28, specifically in *V4a* (Amann et al., 2012) and then in the *V5a* (Amann et al., 2015) version. Both of these updates were most significant for the year 2010 as new information became available.

The GAINS model database has been developed for five-year periods starting in 1990 and extending to 2050 and as shown in Table 1, different ECLIPSE versions include estimates for either the whole time horizon or selected five-year periods. 30 There is one exception; in the *V3* dataset we estimated also global emissions for 2008 and 2009. In order to calculate emission fields for 2008 and 2009 we have used a number of additional sources of information to develop scaling factors for emissions of the year 2005. The exercise was performed at the finest possible sectoral resolution compatible with GAINS but





for some regions only key aggregated sectors (see Table 5) were estimated. For most sectors, the country specific emission ratios were developed using officially reported emissions by US-EPA (http://www.epa.gov), Environment Canada (http://www.ec.gc.ca/inrp-npri/), within UNECE LRTAP Convention (http://www.ceip.at), and 2012 UNFCCC national inventory submissions (http://unfccc.int/). For countries where we found no submissions, emissions for key sectors (Table 5)

were linearly interpolated between 2005 and 2010. Additionally, for flaring in the oil and gas industry the emissions for 2008 and 2009 were calculated using GAINS methodology and data on activities available from the NASA report (Elvidge et al., 2011). Finally for open biomass burning we have used the data from the GFED v3.1 global database (http://www.globalfiredata.org/).

*What is not included and where to find it*

None of the ECLIPSE datasets includes estimated emissions from forest and savannah fires (note that emissions from open burning of agricultural residue are included; see section 3.7), which can be obtained from the GFED v3.1 global database (van der Werf et al., 2010) or a more recent version GFED v4 that was made available subsequently (Randerson et al., 2015). GFED contains emissions for BC, OC, $PM_{2.5}$, and total particle matter (TPM) for the period 1997-2014 in varying temporal and spatial distribution (including gridded dataset) depending on the version (http://www.globalfiredata.org/).

None of the ECLIPSE datasets includes emissions from international aviation but these can be acquired from the Representative Concentration Pathways (RCP) database available at e.g., http://tntcat.iiasa.ac.at:8787/RcpDb/. The data originates from a study by Lee et al. (2009) and were used in the development of the RCPs (Van Vuuren et al., 2011). However, only emissions of black carbon (BC) are included.

Versions *V3* and *V4a* do not include emissions from international shipping and at the time we recommended using datasets

developed for the RCP process (Buhaug et al., 2009; Eyring et al., 2010). Version *V5* and *V5a* include international shipping estimates for all PM species (the RCP set contains only BC and OC), which we have developed drawing on the QUANTIFY[8] project spatial distribution (Endresen et al., 2007) and activity data from Buhaug et al. (2009); for more details see section 3.4.2. The datasets for international shipping, aviation, and open burning have been extracted for use in the ECLIPSE project and can be downloaded (upon request) from the project website http://eclipse.nilu.no.

**3.1 Residential sector**

Several previous studies (e.g., Bond et al., 2004; Cofala et al., 2007; Kupiainen and Klimont, 2007; Lu et al., 2011; Venkataraman et al., 2005) showed that the residential sector is an important source of PM emissions at a regional and global level, especially of carbonaceous species. GAINS distinguishes a number of source categories for residential sector heating and cooking, i.e., fireplaces, stoves, single house boilers and medium sized boilers as well as a number of solid fuels, i.e.,

fuelwood, agricultural residues, dung, and coal as well as liquid and gaseous fuels, i.e. kerosene, fuel oil, LPG, and natural

---

[8] QUANTIFY - Quantifying the Climate Impact of Global and European Transport Systems; European Union Sixth Framework project (https://www.pa.op.dlr.de/quantify/)



gas; see Table 2. The data about fuel consumption used in the GAINS model originates primarily from IEA statistics but is enriched with additional data from regional statistics and studies. This includes regional, rather than national, statistics of coal use in China (Zhao et al., 2013) but most of all additional assessments of biomass use for cooking and heating in several regions; for US, Canada, Finland, Sweden, and Norway drawing on the collaboration within the Arctic Council (AMAP, 2015); for Australia and New Zealand (Driscoll et al., 2000; Scott, 2005); Asia (Amann et al., 2008; Klimont et al., 2009; Purohit et al., 2010; Venkataraman et al., 2010); and finally for Europe where exchange with national experts led to consideration of several local datasets in the GAINS model (Amann et al., 2015). The data used in the last version of ECLIPSE (*V5a*) for Europe are comparable with the independent fuel estimate by Denier van der Gon et al. (2015). Beyond the total fuel use, the split by fuel and installation types is of high relevance (see discussion in section 2.2).

The global fuel use for cooking and heating used in GAINS ranges from about 2100±200 Tg in 1990 to 2600±200 Tg in 2010 and compares well with the total fuel demand estimated in other global studies; for example, Fernandes et al. (2007) estimated total biofuel use in 2000 at 2460 Tg, which compares with GAINS value of 2200-2500 Tg (the range given owing to uncertainties in assumptions about heat value of various biofuels).

The emission factors aim to reflect real world emissions (e.g., MacCarty et al., 2007; Roden et al., 2006, 2009), i.e., incorporate emission measurements of diluted samples, and have been recently compared and updated for Europe (Boman et al., 2011; Pettersson et al., 2011; Schmidl et al., 2011; Tissari et al., 2008, 2009), specifically for modern stoves and boilers, Asia (Cao et al., 2006; Chen et al., 2009; Habib et al., 2008; Li et al., 2009; Parashar et al., 2005; Venkataraman et al., 2005; Zhi et al., 2008, 2009), Latin America (Johnson et al., 2008).

Emission factors and shares of BC and OC in particulate mass emissions from selected measurement literature together with the range of values used in the GAINS model are presented in Tables S2.1 – S2.4 in the SI (S2), where also a brief characterization of stove and boiler categories used in GAINS is provided.

### 3.2 Kerosene lamps

Most of the previous emission studies did not highlight particulate matter emissions from kerosene used for lighting, primarily because the information about emission factors and fuel use was either not available or sparse. Only after Lam et al. (2012) reported very high black carbon emission factors, indicating that this is potentially an important 'missing' source, has more work been done to distinguish between kerosene used for cooking and lighting; the new estimates suggest this source might contribute 5-10% of global BC emissions.

Approximately 250 million households (about 1.3 to 1.5 billion people, mostly in developing Asia and Sub-Saharan Africa) lacked access to reliable electricity to meet basic lighting needs in 2010 (IEA, 2012). These households often rely on fuel-based lighting, with the majority burning kerosene in wick-type lamps (Lam et al., 2012; Mills, 2005); their consumption was estimated at up to 25 billion litres of kerosene per year (Lam et al., 2012). Growing evidence suggests that these light sources pose risks to health (Pokhrel et al., 2010) and environment (Lam et al., 2012), and improvements to lighting may provide numerous welfare benefits to households (Jacobson et al., 2013).



Annual kerosene consumption ($K_i$) for lighting in GAINS region $i$ in year $y$ was estimated by using the following expression

$$K_{i,y} = \left(\frac{POP_{i,y}}{HS_{i,y}}\right)\left(1 - ele_{i,y}\right) * 365 \sum_{j=1}^{n}(N_{i,j,y}h_{i,j,y}CV_k f_{i,j,y}SC_j), \tag{4}$$

where, *POP* represents population, *HS* household size, *ele* electrification rate, *f* share of device type *j* (either wick lamps or hurricane lanterns), *N* number of kerosene lamps, *h* daily operating hours, *SC* specific kerosene consumption of a device, and

$CV_k$ the calorific value of kerosene.

The population data originates from (IEA/ETP, 2012), household size from (UN-Habitat, 2005), the electrification rates from OECD/IEA sources (IEA, 2007, 2011, 2012) and national data/reports (ESMAP, 2005; GOI, 2011; NSSO, 2007). For India, information about the share of lighting devices (i.e., wick lamps, hurricane lanterns), operating hours, specific kerosene consumption are derived from regional studies (Desai et al., 2010; Mahapatra et al., 2009; Purohit and Michaelowa, 2008).

Reported specific kerosene consumption in kerosene lamps varied from 0.005 to 0.042 litre per hour (e.g., Mills, 2003; Pode, 2010) and we assumed 0.006 and 0.02 litre per hour for wick lamps and hurricane lanterns, respectively. Further, we assumed that each household will use three lamps for 6 hours per day whereas the share of hurricane lanterns is 20 percent for South Asia and 50 percent for other regions.

In India, over 44 percent of rural and about seven percent of urban households reported kerosene as their primary source of

lighting in 2004–2005 (NSSO, 2007) and in the lowest four socioeconomic deciles, 60 percent of households use kerosene for lighting (Parikh, 2010). In several of the most populated African countries, including Uganda, Ethiopia, and Kenya, more than 60 percent of the population relies on kerosene as the primary lighting fuel (Apple et al., 2010; IFC/WB, 2008; UBOS, 2010).

Less is known of the quantity of kerosene used for lighting, since it is often difficult to differentiate kerosene used for

lighting from that used for other purposes, particularly cooking. The India Human Development Survey 2005 (Desai et al., 2010) results indicate that kerosene lighting accounts for approximately 65 percent (or 5-6 Tg year$^{-1}$) of residential kerosene consumption in India. Lam et al. (2014) observed that use of kerosene for lighting in electrified homes is substantial (due to intermittent and unreliable electricity supply), constituting an approximately equal share of demand as non-electrified households.

Particulate matter emission factors for kerosene lamps used in this work were derived from Lam et al. (2012). The $PM_{2.5}$ emission factor for kerosene lighting (1.92 g GJ$^{-1}$) is approximately 13 times higher compared to that for kerosene used for cooking (0.15 g GJ$^{-1}$), whereas the OC emission factor for kerosene lighting is roughly one third of the kerosene stove. Furthermore, particulate emissions from kerosene lamps are mostly BC (~92%) (Lam et al., 2016).

### 3.3 Diesel generators

At a global scale, diesel generator (DG) sets are not a large source of pollution but locally, and especially in the developing world, they could be responsible for a significant share of air pollutant emissions, especially nitrogen oxides and black carbon. DG set are the prevailing option for backup power in facilities where continuous power is essential, based on their



combination of reliability, durability, affordability, and overall efficiency (Shah et al., 2006). While increasing power deficit and instabilities in the electricity market resulted in rapid growth of the DG set market in several developing regions, DG have been in use all over the world as backup power facilities, primary electricity generation sources in small remote areas or at initial development stage of industrial plants, for irrigation purposes, etc. The DG sets range from small engines to large

generators, are operated on very variable fuel quality, and the emission limit values have been typically lagging behind those for mobile engines.

There is no direct statistical data on fuel use in DG sets as their consumption is typically part of the energy use reported within power plants, commercial, and potentially agricultural sector. Therefore, fuel consumption was estimated from data on number and size of diesel generators as well as regional studies. The resulting fuel use was compared and calibrated to the

diesel consumption reported in the power and commercial sector.

According to a market review in India, annual DG sales in 2010 were about 150,000 units and they are likely to grow at a rate of about seven percent (Frost & Sullivan, 2010) driven by chronic power shortages and prolific growth in industries, infrastructure, telecommunication, information technology (IT), and IT enabled services. The DG market spans from small (15 – 75 kVA) to large (375.1 – 2000 kVA) sets with estimated diesel consumption of about 5 to 6 billion litres between

2008 (Anand, 2012) and 2010[9]. This represents about 8-9%[10] of total diesel consumption (Anand, 2012; NIELSEN, 2013) and in peak periods up to 18% or even more in some regions (NIELSEN, 2013). In Nepal, electricity deficit has been estimated recently at almost 50% (NEA, 2012) massively increasing dependency on diesel generators. The share of diesel used for DG sets in Nepal is estimated at 15 percent for 2010 (World Bank, 2014a). In Nigeria, total electricity demand is estimated at between 8,000 and 10,000 MW while supply from the national grid is about 4,500 MW, which results in very

heavy reliance on DG sets operating most times between 15 – 18 hours a day (Triple E., 2013; World Bank, 2014b). For South Asia (except Nepal), Cambodia, Indonesia and Myanmar we have used the Indian share of diesel consumption in DG sets whereas in other developing countries, the share of diesel use for DG sets is assumed to be one fourth of the Indian share due to high electrification rates and relatively low power deficit. For sub-Saharan Africa, due to very high power deficit (up to 50 percent), in some regions we have used the share of diesel use in DG sets from Nepal (World Bank, 2014a).

For South Korea, diesel consumption in DG sets was less than 0.2 percent of total diesel consumption (KEEI, 2011). In EU-28, the share of diesel consumption in DG sets is less than 0.4% of the total diesel consumption; however, the share of heavy fuel oil (HFO) use in DG sets is more than 3% of the total HFO used in EU. Similarly, in United States and Japan the share of diesel consumption is small while the share of HFO is approximately 0.5% and 2%, respectively.

Stationary DG sets are frequently operated in harsh conditions and until recently were rarely subject to emission regulation.

Information on DG set emissions factors is fairly limited and not necessarily representative for all regions. GAINS model emission factors were developed on the basis of data reported in a number of studies (Anayochukwu et al., 2013; Corbett and Koehler, 2003; Gilmore et al., 2006; Lee et al., 2011; Lin et al., 2008; Shah et al., 2004, 2006; Shi et al., 2006; Tsai et al.,

---

[9] http://ppac.org.in/
[10] http://www.nipfp.org.in/newweb/sites/default/files/Diesel%20Price%20Reform.pdf





2010; Uma et al., 2004; US EPA, 1996). While it is possible to achieve emissions reductions from diesel combustion through engine modifications, post-combustion control technologies such as diesel particulate filters (DPFs), diesel oxidation catalysts (DOC), and fuel-borne catalysts (FBC) offer an array of options for mitigation or elimination of gaseous and particulate emissions, and can be utilized for both on- and off-road applications (Herzog, 2002; Yelverton et al., 2016). Shah

et al. (2007) observed that DOC and DOC+FBC technologies were effective in reducing mainly organic carbon (OC) emissions (56-77%) while DPFs showed excellent performance in reducing both elemental carbon (EC) and OC emissions (>90%). The emission factors and shares of BC and OC in particulate mass emissions from measurement literature together with the range of values used in the GAINS model are presented in Table S3.1 in SI.

## 3.4 Transport

Globally, the transport sector, including international shipping, is estimated to contribute about 10% of total anthropogenic $PM_{10}$ and $PM_{2.5}$ emissions and up to 25% of BC (Table 8). At a regional level, the role of transport in BC emissions varies strongly and, for example, in Europe and North America was estimated at over 60% in 1996 (Bond et al., 2004) and about 50% in 2005 (Kupiainen and Klimont, 2007) and 2010 in this study, while for East Asia its share grew from about 8 to 23% between 1990 and 2010 (this study). The key source of PM emissions in the transport sector is exhaust emissions from diesel

engines with typically light- and heavy-duty trucks playing the largest role; Europe is an exception as policies favouring diesel fuels, in terms of both tax rates and emission limits, resulted in large a share of diesel cars (Cames and Helmers, 2013). Non-exhaust emissions (brake, tyre, and road wear) represent a relatively small share, especially for carbonaceous particles, but their importance grows over time owing to ever more stringent exhaust emission limits.

The overall energy consumption in the transport sector was taken from Eurostat (EUROSTAT, 2011) statistics for the 28

European Union (EU) member states and from the International Energy Agency (IEA, 2015a, 2015b) for all other countries. Fuel consumption of road vehicles is allocated to the different vehicle types through triangulation with data on the active fleet, their average annual mileage, and their average fuel efficiency. The IEA statistics provide fuel consumption figures separately for rail, aviation, and domestic shipping, however, not for mobile machinery used in agriculture, forestry, industry, and construction and mining sectors. Unless national information is available, as is the case for European countries,

the US and Canada, we re-allocate 80% of diesel fuel consumption from the IEA categories "industry" and "agriculture" to construction and agricultural machinery, respectively. International shipping and aviation are not included in the GAINS model but were estimated for the ECLIPSE project separately; see section 3.4.2.

There is a vast literature on PM measurements of internal combustion engines used in road vehicles in both developing and developed countries, including also pre-regulation vehicles (e.g., Cadle et al., 2009; Cheung et al., 2009; Geller et al., 2006;

Kirchstetter et al., 1999; Liu et al., 2009; Subramanian et al., 2009; Yanowitz et al., 2000). Also old and often poorly maintained vehicle fleet is reflected in measurements of emission factors (e.g., Mancilla et al., 2012) as well as the share of so-called high-emitters (McClintock, 1999, 2007; Smit and Bluett, 2011; Yan et al., 2011, 2014); see further discussion in section 3.4.1. For Europe and the USA we draw the emission factors for road vehicles from established emission factor



models where experts already synthesized the information (HBEFA 3.1, 2010; Ntziachristos et al., 2009; US-EPA OTAQ, 2011). These emission factors are adjusted to conditions in other world regions.

Kupiainen and Klimont (2004, 2007), Bond et al. (2004), Maricq (2007) are examples of studies which summarized and compared emission factors for various vehicle categories. Most of exhaust PM is emitted in a submicron range, actually within 100 nm, and diesel vehicles typically emit several times more (mass-based) PM than equivalent gasoline engines (e.g., Maricq, 2007); exceptions are old vehicles running on leaded gasoline and pre-regulation 2-stroke mopeds (Klimont et al., 2002b; Kupiainen and Klimont, 2004) while latest gasoline direct injection engines have PM mass emissions comparable or even higher than latest diesel engines with particle filter, however, the absolute level is about one order of magnitude lower than for older generations. The carbonaceous particles represent the largest share with the elemental carbon fraction higher for diesel (50–70%) than for gasoline vehicles (30–40%) (e.g., Kupiainen and Klimont, 2007; Maricq, 2007). Non-exhaust emissions, i.e., brake and tyre wear as well as road abrasion were updated based on (Denier van der Gon et al., 2013; EEA, 2013; Harrison et al., 2012). Recent roadside measurements showed that tyre wear produces essentially coarse particles, with only a small contribution (<0.5%) in the $PM_{2.5}$ size fraction (Stein et al., 2012). Road abrasion emissions significantly increase when studded tyres are used, e.g., a common practice in Scandinavian and some Baltic countries. Higher abrasion during winter and spring conditions, average usage period, and application shares are factored into the average abrasion emission factor for the Nordic countries (Kupiainen et al., 2005; Kupiainen and Pirjola, 2011).

PM emission factors for the diverse non-road mobile machinery are much less well established, and only seldom available for developing countries. Moreover, most measurements refer to the mandatory duty cycles rather than real-life operating conditions. For Europe and North America we use emission factors based on (EEA, 2013; OTAQ, 2004; Schäffeler and Keller, 2008) and transfer to other world regions assuming that technology performs similarly and the under comparable operating conditions.

The contribution from diesel engines used in agriculture, construction, mining, rail, shipping, and as back-up generators has been increasing, not least because the emission legislation lags behind that for road transport, but has been receiving more attention recently (e.g., Kholod et al., 2016). Diesel generators and shipping are discussed in separate sections (3.3 and 3.4.2), more recent emission factors for diesel locomotives (e.g., Johnson et al., 2013; Tang et al., 2015) are compared with GAINS in Table S4.3, and emission factors for other non-road machinery used in GAINS were summarized earlier (Klimont et al., 2002b; Kupiainen and Klimont, 2004, 2007) and are also included in the supplementary information (SI). Emission factors for key diesel and gasoline engines in the transport sector from recent literature and the GAINS model are compared in Tables S4.1 to S4.5.

### 3.4.1 High-emitting vehicles

On-road remote sensing measurements of vehicles suggest that a relatively small fraction of the fleet is responsible for a relatively large fraction of emissions (e.g., Ban-Weiss et al., 2009; Cadle et al., 1997; Mazzoleni et al., 2004; Subramanian et al., 2009). In the literature, these vehicles have been referred to as: high emitters or high-emitting vehicles, heavy emitters,




super emitters, gross emitters, excess emitters or smokers, but in principle highlighting the same problem (Shafizadeh et al., 2004). Reasons for their poor emission performance are variable and can be traced back to malfunctioning or totally inoperative emission controls, low combustion efficiency of the engine, engine oil that is entering the combustion chamber, and/or leakage in the exhaust system between the engine and the emissions control devices (Jimenez et al., 2000; Mazzoleni

et al., 2004; Norris, 2001). The shares of high emitters and their contribution to total fleet emissions are variable across countries, with for instance only limited evidence in Europe for light duty vehicles (Borken-Kleefeld and Chen, 2015; Chen and Borken-Kleefeld, 2016) and more modern vehicles seem to have more durable emission controls (McClintock, 2007). Though there is no doubt in the existence of high emitting vehicles, quantifying their emissions is much more speculative.

According to Shafizadeh et al. (2004) two general definitions of high emitters can be identified from the literature: a group

of vehicles that (i) account for a certain fraction, e.g., 50 percent, of air pollutant emissions, or (ii) have emissions above a certain emission threshold or cut-off. The GAINS estimation of high emitter emissions is based on the second general definition. The calculation requires two sets of information: (i) the amplification factor which is the ratio between the high and normal emitter emission factors, and (ii) the share of high emitters in the whole vehicle fleet.

The technology-specific amplification factors, i.e., for Euro 1 to 6, were developed based on existing studies mainly from the

United States (Ban-Weiss et al., 2009; Durbin et al., 1999; Hsu and Mullen, 2007; Yanowitz et al., 2000) and Europe (Carslaw et al., 2011; Ekström et al., 2004) studying the 90-95[th] percentile as the cut-off between high and normal emitting behaviour. Similar datasets from Australia (Smit and Bluett, 2011), China (Guo et al., 2007), Thailand (Subramanian et al., 2009) and Mexico City (Jiang et al., 2005) were also studied in order to find which percentiles would represent the local fleets if the amplification factors identified, based on the 90-95[th] percentiles in the European and US studies, would be

applied also there. The identified percentiles then determined what share of the vehicle fleet corresponded to the amplification factors specified for the high emitting vehicles. A global coverage of the parameterization was developed using the available studies and databases listed above as benchmarks representative for larger groups of countries and regions. We acknowledge that this definition of the high emitting vehicle class is based on a statistical analysis only and currently does not have a technical definition. However, the motivation of the exercise is to single out a portion of the vehicle fleet that

might emit significantly more than the majority of the fleet and study the potential importance of such vehicles in total emissions. The amplification factors determined from the studies varied between pollutants, vehicle types and fuels. Table 6 demonstrates the derived amplification factors for light and heavy duty on-road vehicles that apply for all countries and all PM species, following the observations reported by Subramanian et al. (2009). We have noted the results by Lawson (2010) who showed that the OC/BC ratio might be different for high emitters than for normal vehicles but have not introduced

variable ratios for individual vehicle categories.

The default assumptions about the high-emitter shares are: about 5% for the EU-28, Japan, and Korea; 8% for Australia, Canada, and US; 5-10% for non-EU Europe, 12% for China (except some key cities with more modern fleet where 10% is assumed); 15% for India, and 20% for other developing Asia, Africa, Latin America. These assumptions are compatible with those used in other global studies (e.g., Bond et al., 2004, 2007, Yan et al., 2011, 2014). In addition we factor in that the



durability of the emission controls have increased. Therefore we assume that failure rates decline for the more modern technologies, i.e., above the equivalent of Euro 4, which translates to halving the percent of high emitters for such vehicles. For example, for Europe or Japan for most recent years this results in a lower overall rate of about 2%, which is consistent with assessments for the US and Europe (Chen and Borken-Kleefeld, 2014; McClintock, 2007).

### 3.4.2 International shipping and aviation

Particulate matter emissions from international shipping contribute about 3-4% of the global total, and while, unlike for $SO_2$ and $NO_x$, this is a rather small share, it is also comparable to the contribution of road transport (e.g., Lack et al., 2009). Aviation contributes only a very small proportion of global PM emissions, e.g., for black carbon its share was estimated at about 0.1-0.2% (Lee et al., 2009; Stettler et al., 2013) of which about 14% were during landing and take-off (LTO) (Stettler et al., 2013).

The GAINS model does not include emissions from these sources and the gridded ECLIPSE datasets *V3* and *V4a* refer to other sources, e.g., datasets developed for the RCP process (Buhaug et al., 2009; Eyring et al., 2010; Lee et al., 2009). However, the more recent ECLIPSE sets (*V5* and *V5a*) include international shipping estimates developed using activity data from Buhaug et al. (2009); fuel consumption data for 2007 were extrapolated to 2010 using GDP. Emissions are estimated for all PM species (the RCP set contains only BC and OC) using emission factors shown in Fig. 2 and spatially distributed drawing on the QUANTIFY project[11], i.e., based on global ship traffic data (Endresen et al., 2007). The fuel consumption data includes assumptions about region-specific regulation with respect to fuel quality, i.e., sulphur content of fuels.

The shipping PM emission and their chemical, physical, and optical properties have been analysed for various types of fuels, engines, and vessels, as well as operating conditions, e.g., load factors (Agrawal et al., 2008, 2010, Lack et al., 2008, 2009; Moldanova et al., 2009; Murphy et al., 2009; Petzold et al., 2008, 2010). Further studies reviewed and compared emission factors (Buhaug et al., 2009; Dalsoren et al., 2009; Lack and Corbett, 2012). The particulate matter emission profile, including BC and OC, presented in Fig. 2, was developed on the basis of the studies listed above.

### 3.5 Large scale combustion

Solid fuel combustion in large boilers used in power plants and industry has been a major source of primary particulate matter emissions and although efficient reduction technology exists and is typically required by law, about 15% of total global anthropogenic $PM_{2.5}$ emissions in 2010 originated from this source. At the same time, since 1990's emissions declined by over 30% and its share dropped from over 20% to 15%. Primary PM from combustion can be divided into two major categories: (i) ash, formed from non-combustible mineral constituents in fuel which vary from few to over 30% depending on fuel quality, and (ii) carbonaceous particles, e.g., char, coke and soot, which are formed by pyrolysis of unburned fuel molecules (e.g., Seinfeld and Pandis, 2012). The largest particles remain in the boiler and are removed with bottom ash while

---

[11] https://www.pa.op.dlr.de/quantify/



smaller (typically <100 μm) are entrained in combustion gas forming fly ash. Emissions of elemental and organic carbon from such installations are small due to high combustion temperature, oxidizing conditions, and long residence times (e.g., Ohlström et al., 2000); only about 2% of global total black carbon was estimated to originate from this source (Bond et al., 2004, 2013; Cofala et al., 2007).

The principal statistical data for energy use in the power sector and industry used in GAINS originates from International Energy Agency (IEA, 2015a, 2015b), Eurostat (EUROSTAT, 2011), and national sources, especially for Europe (e.g., Amann et al., 2012, 2015) and Asia (e.g., Amann et al., 2008; Purohit et al., 2010; Zhang et al., 2006; Zhao et al., 2013). The national sources and consultations were especially useful to distribute fuel use among different types of plants; see discussion in section 2.2.3.

The PM emission factors in GAINS are calculated considering region-specific fuel properties (heat value, ash content), installation-specific parameters (ash retention in boiler, size distribution), size-specific efficiency of control equipment (cyclones, wet scrubbers, electrostatic precipitators, fabric filters); see Eq. 1, Eq. 2, and discussion in section 2.1. Detailed review of measurement studies, methodology and assumptions applied in GAINS has been documented in a number of earlier reports and papers (Klimont et al., 2002b, 2009, Kupiainen and Klimont, 2004, 2007; Zhang et al., 2006; Zhao et al.,
2013). Key updates with respect to emission factors have been done for Europe within the work for the European Commission (Amann et al., 2015) and China, where latest information about efficiency and penetration of control measures was used (Zhao et al., 2013).

## 3.6 Industry

There are many industrial processes that emit particulate matter to the atmosphere and the origin of these emissions is often
more complex than that of stationary combustion since there are several process stages, fugitive sources, and the process designs vary significantly across the world. The particular processes will also differ with respect to emission characteristics, i.e., PM size distribution and chemical speciation. The GAINS model distinguishes tens of industrial processes, including several within the iron and steel sector, non-ferrous metals, cement and lime, petroleum refining, coal mining, gas flaring, and production of bricks, coal briquettes, mineral fertilizers, glass, carbon black, and pulp. Extensive discussion of these
sources, including their particulate matter and carbonaceous aerosols emissions and mitigation measures in GAINS is available from previously published reports (Klimont et al., 2002b; Kupiainen and Klimont, 2004). The estimates presented in this paper rely for most sectors on the characteristics presented in those reports, however with updated emission factors for a number of regions and specifically a new structure of the three sectors mostly relevant for carbonaceous particles, i.e., coke ovens, brick making, and gas flaring.
While there are well established PM control technologies applicable to most of the sources (Klimont et al., 2002b; Kupiainen and Klimont, 2004; Maithel et al., 2012) and typically, even in the developing world, there exists legislation prescribing emission limit values, this sector remains among the most uncertain in terms of emission estimation of total PM as well as carbonaceous aerosols. We estimate that at a global scale industrial processes contributed between about 13 and 20% of



PM$_{2.5}$ emissions in 1990 and 2010 and total emissions grew in this period by over 60%. Regional shares might be much larger, e.g., for China were estimated at over 30% in 2010 and grew by nearly a factor of three compared to 2000, or significantly lower, e.g., for Africa less than 5%. For most regions, key PM$_{2.5}$ sectors include cement and iron and steel production representing globally about 75% of industrial emissions of PM$_{2.5}$. For carbonaceous particles, this sector plays a

slightly less important role from the global perspective; Bond et al. (2004) estimated its contribution at about 13% to BC emissions, primarily from coking and brick making. This is broadly consistent with our assessment, although we estimate a somewhat lower share of about 10% globally, of which about a third comes from gas flaring, and there is very strong regional variation from less than a percent to over 20%, especially in regions with high oil production, e.g., Middle East, Russia.

The principal statistical data used in GAINS originates from international sources (Elvidge et al., 2009; EUROSTAT, 2011; IEA, 2015a, 2015b), and national sources, especially for Europe (e.g., Amann et al., 2012, 2015) and Asia (e.g., Amann et al., 2008; Heierli and Maithel, 2008; Huo et al., 2012; Purohit et al., 2010; Zhang et al., 2006; Zhao et al., 2013).

The PM emission factors used in the GAINS model have been discussed in previously published reports (Klimont et al., 2002b; Kupiainen and Klimont, 2004) and key updates concern the region-specific primary technology allocation and

implementation rates of control technologies – as discussed in section 2.2.4 and 2.3. For coke manufacturing (see 3.6.1), brick production (see 3.6.2), and gas flaring in oil and gas industry (see 3.6.3) more significant changes were introduced with new technology and region-specific emission factors.

### 3.6.1 Coke production

Total coke production grew by about a factor of two in the 1990-2010 period and most of the change took place after 2000

when China increased their production by about a factor four from just over a 100 Tg to about 400 Tg coke which represented over 60% of global production in 2010 (Huo et al., 2012) and see http://www.statista.com. China's coke sector undergoes a significant transformation moving from low efficiency and high emission indigenous ovens to highly mechanized recovery ovens, following the world trend (Huo et al., 2012; Polenske, 2006). Several of the other producing countries remained fairly constant or reduced their output in the last decade, e.g., US, Europe, former Soviet Union region,

and only few increased their production, e.g., India, but from the global perspective these changes were not very significant (http://www.statista.com).

There are only few measurements of PM emissions from coke plants and the established emission factors show a wide range. This is partly driven by the varying technology but also owing to the sources of emissions from coke manufacturing since they include several stack and fugitive sources. In the GAINS model, we have constructed a PM emission profile based on

the US EPA Compilation of Air Pollutant Emission Factors (AP-42)[12] and SPECIATE[13] (US EPA, 1995, 2002) as discussed

---

[12] https://www.epa.gov/air-emissions-factors-and-quantification/ap-42-compilation-air-emission-factors
[13] SPECIATE is the US EPA repository of volatile organic gas and particulate matter (PM) speciation profiles of air pollution sources: https://www.epa.gov/air-emissions-modeling/speciate-version-45-through-32




in Klimont et al. (2002b) and Kupiainen and Klimont (2004), and updated it with more recent measurements discussed in Huo et al. (2012) and Weitkamp et al. (2005). For uncontrolled ovens, GAINS emission factors for $PM_{2.5}$ range from about 2 to 4.8 kg t$^{-1}$ coke, the upper bound being representative for China and the range reflecting different oven types across the global regions. For BC and OC, the emission factor range is 0.28-1.3 kg t$^{-1}$ and 0.46-2.2 kg t$^{-1}$, respectively, with upper range values representing Chinese indigenous ovens. The PM emission factors for China are comparable to the ones used in recent Chinese studies (Huo et al., 2012; Lei et al., 2011) and the ratio of BC/OC of about 0.6 is also consistent with the estimates by Weitkamp et al. (2005).

### 3.6.2 Brick kilns

The brick making industry is dominated by production in the developing countries where over 95% of global output, estimated at about 1.5 trillion bricks per year (e.g., Schmidt, 2013), is produced and most of it in fairly inefficient and polluting kilns. In India, over 70% of kilns, or about 100,000, are clamp kilns, the least efficient kiln that remains widespread in the developing world. More than 1.2 trillion bricks per year are produced in Asia alone which is associated with use of over 100 million tons of coal as well as other fuels including agricultural residues, dung, and waste (Heierli and Maithel, 2008; Schilderman and Mason, 2009). The largest brick producing countries in Asia are China, India, Pakistan, Bangladesh and Vietnam (AIT, 2003; FAO, 1993; Heierli and Maithel, 2008; Maithel, 2014). Worldwide non-automated brick production, including artisanal brick kilns, in developing countries is about 1.25 trillion bricks per annum and is distributed between three main regions (i) China – about 700 billion bricks or 56%, (ii) India – about 150 billion bricks or 12%, (iii) Asia, Africa, South America & Mexico – about 400 billion bricks or 32%. In contrast, worldwide machine made brick production using automated kilns, is approximately 125 billion bricks, with Australia's brick production accounting for only 2 billion, UK 4 billion, USA 8 billion, China 100 billion, and other developed countries approximately 11 billion bricks. A summary of studies used to compile the brick production data is provided in SI (S5) along with the activity data used in ECLIPSE *V5a* for key global regions (Table S5.2).

Even though from the global perspective, the brick manufacturing sector does not represent a major share of particulate matter emissions, about 1-2% for $PM_{2.5}$ according to our estimates and less than 5% for BC (e.g., Bond et al., 2004, 2013), the regional impacts might be much more significant (Guttikunda et al., 2013; Le and Oanh, 2010; Skinder et al., 2014). And while many countries may have emissions standards, i.e., maximum permissible concentrations of several pollutants, including PM, the enforcement is difficult for several reasons including relatively few measurements available. Maithel et al. (2012), Weyant et al. (2014), and Rajarathnam et al. (2014) reported particulate matter measurements for key brick kiln production technologies in Asia (primarily India and Vietnam), and a few studies, focusing on toxics and black carbon, were performed in Mexico (Cardenas et al., 2012; Christian et al., 2010; Maíz, 2012); the latter covered several types of kilns including the Marquez kiln (MK) that is specific to Latin America. For the main brick producing technologies in South Asia, the PM emission factors derived from the above measurements are lower by over 30% for BC and 90% for $PM_{2.5}$ than previously estimated values (Weyant et al., 2014), which were used in several regional (Klimont et al., 2009; Lu et al., 2011;



Ohara et al., 2007; Zhang et al., 2009) and global inventories (Bond et al., 2004; Cofala et al., 2007; UNEP/WMO, 2011). Additionally, the BC/TC ratio appears higher than previously thought (Weyant et al., 2014).

The emission factor set used in GAINS to calculate ECLIPSE set is more in line with the currently available measurements although it was developed prior to the publication of measurements by Weyant et al. (2014); compare Table S5.1 in the SI,

where current GAINS emission factors for $PM_{2.5}$, BC, and OC are compared with the previous GAINS dataset and recent measurements by Weyant et al. (2014). Also the EC/TC ratio in GAINS, from about 0.67 for zig-zag, about 0.75 for clamps, downdraft, moving chimney BTK, and 0.93 for fixed chimney BTK, resembles the measurements by Weyant et al. (2014)

### 3.6.3 Gas flaring

Understanding venting, flaring, and associated gas utilization practices in the oil industry has been of high relevance for the

assessment of methane emissions while it was not considered as a potentially important source of air pollution. Consequently, non-$CO_2$ emissions from flaring of associated gas in oil industry were not part of previous inventories (e.g., Bond et al., 2013), including the datasets used in the IPCC assessments. We have developed the first global estimate of air pollutant emissions from this activity, including black carbon, which was used first in the studies focusing on the role of black carbon and other short-lived climate forcers in climate mitigation (Bond et al., 2013; Shindell et al., 2012; UNEP,

2011; UNEP/WMO, 2011; World Bank and ICCI, 2013). Within the ECLIPSE project, an update and future mitigation scenarios (Klimont et al., in preparation) were developed and used in several regional and global modelling exercises (Stohl et al., 2013, 2015).

Associated petroleum gas (APG) is gas that is associated with the oil in the reservoir and once oil is extracted, the dissolved gas follows and is commonly separated from the oil and either vented or flared. The volumes and composition of APG

depend on several factors including the nature of the oil reservoir, degree of depletion, etc. (PFC Energy, 2007; Røland, 2010). While the APG could be utilized, the lack of developed markets, missing infrastructure, no legislation, etc. resulted in very low recovery rates before 1980; only in the last decades has the flaring trend been decoupled from oil production but the level of gas utilization varies greatly among the producing regions. Globally, about 140-160 billion m³ (bcm) APG have been flared annually, which represents about 5% of 2009 global natural gas consumption or about 30% of European Union

demand (Elvidge et al., 2009). Regions where the largest volumes of gas are flared include Middle East, Russia, Northern Africa, Nigeria, Venezuela representing about 70-80% of the global total (Elvidge et al., 2009, 2013). There are significant uncertainties in estimates of flared volumes as metering is rare and official estimates differ significantly from remote sensing data or even between different official versions, e.g., for Russia governmental sources reported for 2006 about 15-20 bcm of APG flared while Global Gas Flaring Reduction Initiative (GGFR) estimates were about 40-60 bcm (PFC Energy, 2007).

The reported share of APG flared in Russia in 2006 varied from 27% (governmental sources) to 75% (NGOs) with 45% estimated by PFC Energy (2007) (Røland, 2010). For Nigeria, flaring volumes have been estimated or reported between 10 to 25 bcm indicating that up to 70% of APG is flared (Aghalino, 2009; Ite and Ibok, 2013). While for several countries APG utilization rates have been increasing (Elvidge et al., 2009; Haugland et al., 2013), Russia made relatively little progress in



recent years in spite of new legislation requiring a 95% recovery rate (PFC Energy, 2007; Røland, 2010). For US, flaring volumes increased by about a factor three between 2006 and 2011 owing to the boom in unconventional gas and oil production (Elvidge et al., 2013). GAINS activity data relies on the time series of gas flaring volumes developed within the GGFR initiative (Elvidge et al., 2007, 2011)

There is a very limited number of measurements of flaring emissions allowing the establishment of a representative set of emission factors where local flare operating conditions and APG properties could be considered. Some of the earlier published PM emission factors (about 2.6 g m$^{-3}$) referred to landfill (CAPP, 2007) or refinery flares (US EPA, 1995) and are generally considered inappropriate. A new technique for quantitatively measuring soot emission rates in flare plumes under field conditions has been reported by Carlton University group (Johnson et al., 2011) and while their average BC emission

factor of 0.51 g m$^{-3}$ (McEwen and Johnson, 2012) considers representative fuel mixtures, their measurements were performed on laboratory-scale flares, which might underestimate real-world emissions. The first ECLIPSE datasets include flaring emissions calculated with one BC emission factor of 1.6 g m$^{-3}$ gas flared, assuming that real-life flares perform much worse than laboratory measurements. In the later ECLIPSE set *V5a*, region-specific PM emission factors were developed considering a more recent study measuring emissions from flares in the Bakken region (Schwarz et al., 2015) which

confirmed the order of magnitude measured by McEwen and Johnson (2012) by establishing upper bound BC emission factor of 0.57±0.14 g m$^{-3}$. We have assumed that such emission rates are representative for well operated flares, i.e., OCED countries. For other countries we retained the previously used value of 1.6 g m$^{-3}$ but considered, where available, composition of flared gas that apart from methane includes several heavier hydrocarbons. The relationship between BC emission factors and heat value of flared gas has been proposed by McEwen and Johnson (2012) and was also applied in

estimates for Norway (Aasestad, 2013) and Russia (Huang et al., 2015).
The range of current BC emission factors in GAINS is ~0.5-1.75 g m$^{-3}$, the upper bound represents values for Russia, and the estimated heat value of APG varied from about 41 to 50 MJ m$^{-3}$. Huang et al. (2015) suggested even higher BC emission factors for Russia (2.27 g m$^{-3}$) assuming local APG composition with estimated heat value of about 75 MJ m$^{-3}$ and extrapolating linearly from the relationship from McEwen and Johnson (2012); well beyond the range presented there.

Finally, the most recent measurements of BC from flaring, also in Bakken field, estimate much lower overall emission factors of 0.13±0.36 g m$^{-3}$ and characterize flares without visible smoke (Weyant et al., 2016) and therefore, likely not representative for regions with visible high-density smoke, e.g., Russia, Nigeria, Middle East, Northern Africa (e.g., Aghalino, 2009; Elvidge et al., 2013; Pederstad et al., 2015). We assume that all PM from flaring is PM$_{2.5}$ and BC and OC represent about 78% and 16%, respectively. These assumptions are broadly consistent with the results of McEwen and

Johnson (2012) who reported BC/OC share of 80/20 and Fortner et al. (2012) measuring 4-20% of OC and over 95% of PM within PM$_{2.5}$.



### 3.7 Agricultural waste burning

Bond et al. (2004) estimated that globally about 7 and 15% of anthropogenic (excluding forest and savannah fires) BC and OC emissions originated from this source in 1996; our own estimates point to a slightly lower share in carbonaceous particles emissions but mostly because our total, not agricultural burning, estimates are higher. At the same time for several regions this source might be even more important, e.g., for Brazil we estimate up to 15% of PM$_{2.5}$ and 10% of BC. These emissions have been also linked with heavy smog and haze episodes (e.g., Mukai et al., 2015; Stohl et al., 2007).

Typically assessment of global emissions from open field burning of agricultural residues is based either on a compilation of national reports/sources (e.g., Bond et al., 2004; EC-JRC/PBL, 2010) or on remote sensing data which characterize the magnitude and spatial distribution of open biomass burning including agricultural, savannah, and forest fires (e.g., van der Werf et al., 2010; Wiedinmyer et al., 2011), however, it has been shown the latter underestimates small open fires (e.g., Randerson et al., 2012). Niemi (2007) compared various datasets for all open biomass sources and developed the first global activity set for the RAINS model drawing on EDGAR3.2FT2000 (Van Aardenne et al., 2005). This database has been further extended and updated to accommodate other data sources allowing gaps to be filled for several countries. Specifically, we have used estimates from the global studies (Bond et al., 2004), a number of regional estimates (Cao et al., 2008; Oanh et al., 2011; Pettus, 2009), reporting of emissions to EMEP (http://www.ceip.at), and bilateral discussions within the revision of the European air pollution policy (Amann et al., 2015). Our global estimate of open burning of agricultural residue has been fairly constant in the assessment period varying from about 485 to 515 Mt between 1990 and 2010; this estimate is comparable with 475 Mt for 1996 by Bond et al. (2004) and higher than the original EDGAR3.2FT2000 of 252 Mt of residue burned in 2000.

To derive particulate matter emission factors, we have relied on Andreae and Merlet (2001), Turn et al. (1997), and Hegg et al. (1997); the latter for the OM/OC ratio which we assumed 1.7 as discussed in Kupiainen and Klimont (2004). The default emission factors used in GAINS (all values in g kg$^{-1}$) are 8.5 for TSP, 7.1 for PM10, 6.3 for PM$_{2.5}$, 5.6 for PM$_1$, 2.62 for OC and 0.83 g kg$^{-1}$ for BC. Using data from Turn et al. (1997), these values were adjusted for specific regions considering typical type of crops; for example, for regions with a high share of rice production (primarily Asia) the values of BC and OC factors were estimated at 0.6 and 2.2 g kg$^{-1}$.

### 3.8 Waste

Open burning of solid waste is a widespread method, especially in the developing world, to reduce the volume or odours of dumped or uncollected municipal solid wastes (EAWAG, 2008) and it has been identified as significant source of particulate matter and hazardous air pollutants to the atmosphere (Christian et al., 2010; Hodzic et al., 2012; Kumar et al., 2015; Wiedinmyer et al., 2014). The estimated magnitude of emissions and contribution to PM concentrations vary widely across the studies, ranging from a few percent to nearly 50% of the total contribution in particular regions. While large uncertainties



remain owing to only few measurements and difficulties in finding reliable data on waste collection, recycling, and disposal rates, the open burning of residential waste is a potentially important source of PM, especially in the developing world.

To estimate the region-specific share of the municipal solid waste (MSW) that is burned, we used a mass balance approach described in the IPCC Guidelines for National Greenhouse Gas Inventories (IPCC, 2006b). As a starting point, we used the

IPCC reported data on MSW generation and management and assumed that the category "*other MSW management, unspecified*" represents the upper limit for the open burning of residential waste. However, the IPCC values were not used directly in many cases, because the IPCC unspecified fractions are in some cases relatively high, up to 60 percent, and also because not all unspecified mass is necessarily burned. We have additionally used information on percentages of commonly used MSW disposal methods in other studies (CEPMEIP, 2002; EAWAG, 2008; Neurath, 2003); the final fraction of open

burning from the total waste produced in the developed world was estimated to vary between 0.5 and 5% and for the developing world the region-specific fractions were estimated at 10-20%. The GAINS model estimate of the global MSW is about 1500 to 2150 Tg in the period 1990 to 2010 of which about 115 to 160 Tg were estimated as openly burned. While the total waste generation rate is consistent with other studies (e.g., Christian et al., 2010; Wiedinmyer et al., 2014), the open burned fraction differs significantly owing to different assumptions about the fraction burned and practices in urban and rural

areas. For example Bond et al. (2004) and Wiedinmyer et al. (2014) estimated that 33 and 970 Tg of waste are burned; the latter is still about six times larger than GAINS. We were not able to consider the results of Wiedinmyer et al. (2014) in GAINS yet, but a comparison at the national level shows that GAINS has significantly lower estimates for most of the developing countries as well as Europe; for the latter, GAINS is consistent with national reporting and often a factor 5 to 10 lower than Wiedinmyer et al. (2014). For the US and Canada GAINS has a factor of 2-3 higher estimates (also consistent

with the US EPA and Environment Canada).

The PM emission factors used in GAINS were derived from Akagi et al. (2011) and Christian et al. (2010) and are consistent with the ones used by Wiedinmyer et al. (2014). These are (all in g kg$^{-1}$) 9.5 for $PM_{10}$, 8.74 for $PM_{2.5}$, 6 for $PM_1$, 5.27 for OC, and 0.65 g kg$^{-1}$ for BC.

### 3.9 Other sources

The GAINS model also includes several other sources of PM which at a larger scale represent rather small contribution but could be of relevance locally. These are mostly non-combustion (fugitive) emission sources and include animal livestock, storage and handling of bulk industrial and agricultural products, arable land related agricultural activities, and construction works. Additionally, emissions from cigarette smoking, barbeques, and fireworks are considered.

The predominant sources of PM from animal housing include feed and faecal material, bedding, skin, hair, mould, and

pollen. Size-specific PM emission factors were developed in GAINS drawing on the results of measurements done in Europe (e.g., ICC &SRI, 2000; Louhelainen et al., 1987; Takai et al., 1998) which are discussed in more detail in Klimont et al. (2002b). The values presented in that report were adapted considering region-specific length of the housing period (time animals spend indoors) which is a regional parameter in the model, also relevant for estimation of ammonia emissions. For



dairy cows the $PM_{10}$ factors range from 0.22-0.43 kg animal$^{-1}$ per year, for beef 0.11-0.43 kg animal$^{-1}$, for poultry about 0.05 kg animal$^{-1}$, and for pigs 0.4-0.45 kg animal$^{-1}$. The share of $PM_{2.5}$ is about 22% with the exception of pigs where it was estimated at about 17%; no BC or OC emission factors were assumed. Emissions from arable farming include harvesting, ploughing, tilling, etc. The GAINS $PM_{10}$ emission factor varies from 0.8 to 2 kg ha$^{-1}$ and the $PM_{2.5}$ is assumed to represent

about 22% of $PM_{10}$. These revised numbers, compared to the earlier GAINS values discussed in Klimont et al. (2002b), draw on the more recent work in Germany and France discussed within the EU air quality consultation (Amann et al., 2015).

Emissions from storage and handling of bulk industrial (coal, iron ore, fertilizers, cement, other) and agricultural products as well as from construction activities are estimated using emission factors discussed in Klimont et al. (2002b). For the latter, some updates were made based on national consultations within work on the revision of the EU air quality policy (Amann et

al., 2015) and the recent range for $PM_{10}$ is 0.07 – 0.22 Gg per million m$^2$ of constructed floor space, with a share of $PM_{2.5}$ assumed at 12% and no primary carbonaceous particles.

For cigarette smoking we assume a $PM_{2.5}$ emission factor of 0.01 – 0.0165 kg capita$^{-1}$ (equal to $PM_{10}$) and a share of BC and OC as 0.5% and 60%, respectively (Klimont et al., 2002b). Also for barbeques, a per capita emission factor is established, i.e., 0.02 – 0.075 kg capita$^{-1}$ with a share of BC and OC assumed as about 15% and 50%, respectively (Klimont et al.,

2002b).

## 4 Results and discussion

Global, regional and sectoral emissions of particulate matter (PM) distributed into several size bins ($PM_{10}$, $PM_{2.5}$, $PM_1$) as well as into black and organic carbon are shown in Table 7-8 for 2010 and Fig. 3-4 for the period 1990-2010; Table S6.2-S6.6 in the SI show global emissions of PM species for 25 global regions in the period 1990-2010. To our knowledge, these

estimates represent the first global dataset of anthropogenic emissions where size-specific mass PM calculation, including BC and OC, was performed using a uniform and consistent estimation framework. Emissions are also allocated into a 0.5$^{\circ}$ x 0.5$^{\circ}$ (longitude-latitude) grid and available freely for a number of datasets[14]. Finally, the PM estimates are consistently linked with the emissions of other air pollutants and greenhouse gases for the same time period as well as their future projections developed with the GAINS model (Klimont et al., in preparation).

Total emissions of particulate matter (including open burning based on GFED3.1 database but excluding windblown dust) in 2010 are estimated at about 111 Tg for $PM_{10}$, 81 Tg for $PM_{2.5}$, 71 Tg for $PM_1$, 9.5 Tg for BC, and 33 Tg of OC. Anthropogenic contribution dominated all species except OC and OM, i.e., about 55% of $PM_1$, $PM_{2.5}$, and $PM_{10}$, 75% of BC, and 40% for OC and OM (Table 7). For all considered PM species, sources in Asia represented over 60% of the global anthropogenic total (Table 7) with residential combustion being the most important sector although its share declines with

increasing particle size: about 60% for BC and OC, 45% for $PM_{2.5}$ and less than 40% for $PM_{10}$ for which large combustion sources and industrial processes are equally important (Table 8).

---

[14] http://www.iiasa.ac.at/web/home/research/researchPrograms/air/Global_emissions.html





In contrast to several local and regional atmospheric modelling studies, the global modelling community has been relying so far on the assumption that anthropogenic $PM_{2.5}$ emissions are sufficiently well represented by the sum of black carbon and primary organic PM, often referred to as POM. This total fine PM mass is typically estimated as BC+1.4*OC[15]. Combining such estimates with windblown dust and open biomass fires to arrive at the total $PM_{2.5}$ might be sufficient from the perspective of global climate impacts of primary PM aerosols; however, the health impacts could be severely underestimated in some regions where the non-carbonaceous share of anthropogenic fine particulate matter is significant (Fig. 3).

We argue that assessment of health impacts due to PM using results of the global emission projections developed in the first place for climate simulations, e.g., Representative Concentration Pathways (RCP), which included anthropogenic BC and OC, windblown dust, and open fires but not the non-carbonaceous component of primary $PM_{2.5}$ and $PM_{10}$ emissions originating from combustion, industrial processes, and some fugitive sources, might lead to inconsistent results and underestimation of PM concentrations and regional impacts. This study provides the first global assessment of the role non-carbonaceous particle emissions play in total anthropogenic $PM_1$, $PM_{2.5}$, and $PM_{10}$ mass emissions and could prove more appropriate to use in global modelling studies of health impacts as well as climate. Moreover, while at the global level, the ratio of anthropogenic emissions of $PM_1$ and $PM_{2.5}$ to (BC+POM) is about 1.3 and over 1.6, there are important differences between the regions and the emission ratios have been changing over time (Fig. 3). For example, in 2010 we estimate for Asia an emission ratio of two for $PM_{2.5}$/(BC+POM) while for North America the same ratio is about 1.5 (Fig. 3, Table 7). In Europe, including Russia, this ratio has changed from about three in the early 1990's, where primary PM emissions from poorly controlled coal power plants and heavy industry (not a large source of carbonaceous particles – compare Fig. 4) dominated the total, to below two in 2010 (Fig. 3). Even when the emissions from open biomass burning (forest and savannah fires) are taken into account, and most of these occur far from densely populated areas, the total $PM_{2.5}$ mass emissions are over 20% larger than the BC+POM (Table 7).

We estimate that about 75% of global anthropogenic emissions of $PM_{10}$ are $PM_{2.5}$ and while there was only little change in that ratio (slight increase) in the last decade at the global level, more significant variation has been observed across sectors (Fig. 4). Combustion of liquid fuels, biomass, and waste produces typically over 90% of $PM_{2.5}$ in $PM_{10}$ but for several industrial processes, power and industrial boilers burning coal, and coal production, distribution and storage, emissions of $PM_{2.5}$ represent only 40-60%. Carbonaceous particles (BC+OM) emissions play a key role in $PM_{2.5}$ representing over 60% with the largest contribution from residential combustion (about 80%) and transport and agriculture (each about 10%). Nearly 90% of $PM_{2.5}$ emissions from residential boilers and cooking and heating stoves is BC+OM of which over 20% is BC. Similarly high share of BC+OM is estimated for transport sector but it varies between about 95% for road transport and 80% for non-road vehicles, however, the share of BC is much larger than for residential combustion: $35 - 45\%$ of $PM_{2.5}$ emissions from transport (including non-exhaust) is BC. Few of the smaller sources, agricultural residue and trash burning, have also large share of BC+OM (over 80%) but rather small contribution of BC. Combustion of solid fossil fuels in power

---

[15] The 1.4 has been the most commonly used OM/OC ratio (Aiken et al., 2008)



and industrial boilers as well as most industrial processes (except brick manufacturing in traditional kilns and possibly coke making) are characterized by very low share of carbonaceous particles (below 5%).

## 4.1 Regional distribution and temporal trends

Total anthropogenic emissions of $PM_{2.5}$ and BC in 2010 have a similar spatial distribution (Fig. 5). Emission densities are generally the highest in Asia, however, there are some important differences in the contributions of various sectors to both species as well as across regions. Residential combustion plays a key role but appears far more important for BC where it represents nearly 60% of the global total (Table 8) and an even higher share for Asia and Africa; for $PM_{2.5}$ this sector contributes globally about 45%. While for $PM_{2.5}$ the energy and waste sector (incl. agricultural burning) and industry make most of the remaining emissions (25% and 17.5%, respectively), they represent just over 10% of BC emissions (Table 8 and Fig. 5). Industrial emissions appear much more important in Asia (Fig. 5) and while there are several processes contributing to $PM_{2.5}$ emissions, for BC brick and coke production make the most and represent up to 12% of Asian emissions, globally about 6%. Some sector contribution patterns are similar across continents, for example, for North America, Latin America, and Europe transport and the residential sector dominate BC emissions while for $PM_{2.5}$ it is mostly energy and the waste sector, except Europe where also residential combustion appears important (Fig. 5). For Africa, residential combustion is the key source of all PM with the exception of a few areas like Republic of South Africa or oil producing countries where the energy sector is an important source. It is particularly striking to see the difference in the source contributions to BC emissions in Africa and Asia where the most important source is the residential sector, but while in Africa other sources are barely visible, for Asia there are important contributions from the transport and industry (Fig. 5). The other feature worth highlighting is the difference in relative importance of the transport sector for $PM_{2.5}$ and BC emissions (about 8% and 24% at a global level, respectively) which is clearly visible in the third row of maps in Fig.5.

We estimate that global emissions of PM have changed little in the period 1990-2010 showing a strong decoupling from the global increase in energy consumption and consequently $CO_2$ emissions (Fig. 3). However, there are very different regional emission trends with a particularly strong increase in East Asia and Africa and a strong decline in Europe, North America and Pacific. The development of $PM_{10}$ and $PM_{2.5}$ emissions is fairly similar with a slightly faster growth of $PM_{2.5}$ (+8%) than $PM_{10}$ (+4%) at a global level. The difference is mostly due to reductions of industrial emissions in Europe and Russia following the political and economic transition in Eastern Europe that started already in the mid-80s. This economic restructuring resulted in closure or transformation of inefficient and polluting heavy industries which in turn brought in about 55 and 60% reduction of $PM_{2.5}$ and $PM_{10}$ emissions between 1990 and 2010; most of which was achieved before 2000 (Fig. 3). Also North American and Pacific emissions declined in this period by about 30%. In contrast, $PM_{10}$ and $PM_{2.5}$ emissions in East Asia and Africa increased by about 40-50% and those of Other Asia and Latin America by about 10%. The stark differences in regional trends resulted in important changes in the spatial pattern of PM burden. Europe, North American, and Pacific contribution to global emissions dropped from nearly 30% in 1990 to well below 15% in 2010 while Asia's contribution grew from just over 50% to nearly 2/3 of the global total in 2010 (Fig. 3, Table 7, Table S6.2-S6.3).




For black carbon (BC), the regional changes were less dramatic but the global emissions are estimated to grow by about 15% by 2010 compared to 1990, mostly driven by increases in Asia (about 30%) and Africa (over 40%) (Fig. 3, Table 7, Table S6.5-S6.6). BC emissions in Europe, North America, and Pacific declined by about 30% but their share in the global total are estimated at below 15% in 2010 (from about 24% in 1990).

## 5  4.2 Comparison with other studies

This is the first assessment of the global anthropogenic emissions of $PM_{10}$, $PM_{2.5}$, and $PM_1$ using a consistent bottom up approach across all the sources and regions and therefore only limited comparison to other work at a global level can be made. In fact, the only global set where $PM_{10}$, $PM_{2.5}$, BC, and OC were published is the so called 'mosaic inventory', developed within the UNECE Task Force on Hemispheric Transboundary Air Pollution (HTAP) where a compilation of EDGAR and several regional inventories were put together (Janssens-Maenhout et al., 2015) for 2010. For most of the species the HTAP_v2 is lower than ECLIPSE *V5a* by about 20-30% except OC where the agreement is good (Table S8.1 in SI). It is difficult to easily conclude on the reasons for observed differences as the methods are not fully comparable and HTAP_v2 is a compilation where single products rely on different methods. However, as further discussion shows, the largest discrepancy for $PM_{10}$ and $PM_{2.5}$ is for China as well as Europe and Russia; combined they represent about 90% and over 50% of all the difference for $PM_{10}$ and $PM_{2.5}$. There have been a number of global studies of BC and OC emissions as well as several regional assessments of $PM_{10}$, $PM_{2.5}$, BC, and OC which we discuss in a more detail below.

A seminal work by Bond et al. (2004) established a benchmark global inventory of BC and OC emissions for the year 1996 that was later updated to 2000 (Bond et al., 2013) and was also used as the basis for the development of BC and OC emission in the RCP scenarios (Lamarque et al., 2010; Van Vuuren et al., 2011). Bond et al. (2004) provided a thorough review of BC and OC estimates to date and has been used as the primary reference since. We compare our results with Bond et al. (2004, 2013) in Table 9 and Fig. 6 for 1995 and 2000. At a global level, recent GAINS calculation (*V5a*) shows higher values, which is mostly due to inclusion and re-estimation of few sources: kerosene wick lamps, gas flaring, use of regional coal statistics for China; Fig. 6 shows the role of these sources in GAINS estimates for 2000, ECLIPSE version totals (see also Fig. S6.1 in SI), and compares them to the range presented in Bond et al. (2013). Even though the global totals fall within the same range, especially when considering the role of newly calculated emissions from kerosene lamps (version *V4a* did not include them), there are often larger differences at a source-sector level, particularly for residential combustion where largest uncertainties exist in fuel consumption, its allocation between uses and technologies, and emission factors (Table 9). Excluding kerosene lamps and gas flaring, which were not included in Bond et al. (2004, 2013), GAINS global estimates are larger by less than 5% and 15% for 1995 and 2000 than Bond et al. (2004, 2013). This difference is mostly due to residential sector where comparable source categories are larger in GAINS by 40-60% but the overall balance is partly offset by emissions from industrial coal use (including coke and brick production as well as industrial boilers) that are larger in Bond et al. (2004, 2013) (Table 9).



Emission characteristics for kerosene lamps, gas flaring, and diesel generators have been included in GAINS only recently (most of the previously published global work has not included these sources). For kerosene wick lamps we followed on the work of Lam et al. (2012) but developed an independent assessment of activity data and estimated global BC emissions from this source at 706 Gg in 2005. Our estimates are higher than the previous assessment of 270 Gg (Lam et al., 2012) and 580

Gg (Jacobson et al., 2013) because of larger kerosene consumption in our study, but compare well to Elisabeth (2013) who calculated 702 Gg BC from this activity. For gas flaring we estimated global BC emissions at about 270 Gg and 210 Gg in 2005 and 2010. A recent study of flaring emissions for Bakken field (Weyant et al., 2016) extrapolated their results to global estimates of $20\pm6$ Gg BC, assuming the same range of emission factors as measured by them at the Bakken field. This is over ten times less than our estimates but we argue that the Bakken flares are not necessarily representative for some of the

other regions where strong variability and potentially high soot emissions have been shown by (Conrad and Johnson, in review; Johnson et al., 2011) and also speculated in Huang et al. (2015). We found no global estimates of PM emissions from diesel generators and our estimate of 113 Gg for $PM_{2.5}$ and 50 Gg for BC in 2010 confirms that it appears to be a rather small source from a global perspective, and although, important locally, it is expected that in the near future with reliable access to grid electricity use of DG sets will be limited particularly in residential, commercial and industrial sectors.

Granier et al. (2011) compared global and regional estimates of BC developed within global and regional modelling activities or inventories for the period 1980-2010. We compare the range presented in that study with the inventory used during development of RCP scenarios (Lamarque et al., 2010) and the GAINS model calculation for version *V5a*, highlighting the role of the newly included and re-estimated sources (Fig. 7). At a global level, the GAINS range overlaps with the span of estimates presented in other studies, although the GAINS total is actually higher than all previous estimates

and the post 2000 trend is also different, implying a slight increase in emissions rather than a decline or stabilization shown in earlier studies. As shown in comparison to Bond et al. (2004, 2013) (Table 9), the GAINS values are higher primarily due to inclusion of kerosene lamps and gas flaring but also because of more recent statistical data for 2010 than used in the previously published work. Fig. 7 also includes results of selected global and regional studies which were not explicitly referred to in Granier et al. (2011); these are marked with 'black star' symbols and included in Table S8.1 in SI. The values

for 1996 and 2000 refer to the Bond et al. (2004, 2013) and for 2010 to HTAP_v2 inventory (Janssens-Maenhout et al., 2015), none of which included emissions from kerosene wick lamps.

Fig. 7 shows also a similar comparison for selected countries: China, India, and US; note that the ranges presented in Granier et al. (2011) for regions/countries do not necessarily add up to the global total as the former included also selected regional studies which were not part of the comparison of the global totals. For China, a continuing growth in BC emissions has been

reported in all investigated studies. GAINS is comparable with the RCP input (Lamarque et al., 2010) for 1990-1995, while for the last decade is consistently higher or at the top of the range, which in Granier et al. (2011) is representative of the upper estimates in the RCP scenarios rather than specific inventories. However, a number of recently published studies for



China reported rather high BC, e.g., Zhang et al. (2009) estimated about 1.8 Tg for 2006, HTAP_v2 (based on the MEIC[16] system developed by the Tsinghua University (Beijing, China)) 1.76 Tg for 2010, Lu et al. (2011) 1.84 Tg for 2010, and 1.92 Tg for 2008 using a top-down approach (Kondo et al., 2011); these results and other recent regional studies are marked with 'black star' symbols in Fig. 7 and included in Table S8.1. Several authors estimated also $PM_{10}$ and $PM_{2.5}$ emissions for

China and these compare reasonably well with GAINS, although are systematically lower by up to 15% with exception of the HTAP_v2 mosaic inventory (Janssens-Maenhout et al., 2015) which is lower by nearly 25% for 2010 (Table S8.1 in SI); the latter inventory relies on the data from the MEIC system where more optimistic assumptions about the penetration and achieved efficiency of wet scrubbers and electrostatic precipitators in industry are made. For India, all inventories suggest emissions have been increasing in the investigated period but there is a very large spread of estimates. Current GAINS

estimates are higher than Lamarque et al. (2010) and the range shown by Granier et al. (2011) (Fig. 7) – the overlap in the last decade is because the upper values are based on the earlier GAINS model estimates (e.g., Klimont et al., 2009) which are consistent with ECLIPSE set. Some recent papers have shown similar BC emissions as GAINS (e.g., Janssens-Maenhout et al., 2015; Lu et al., 2011; see also Table S8.1) but overall the range of published emission estimates for PM species for India varies greatly between studies, e.g., for BC from about 350 Gg to over 1000 Gg (Table S81). A lot of that variability links to

different assumptions about biomass use for cooking (Venkataraman et al., 2005), efficiency of PM abatement in power and industry, large uncertainty in agricultural burning activity (Venkataraman et al., 2006). For the US, all studies indicate a declining trend in BC emissions (Fig. 7). However, in contrary to China and India, GAINS emissions are in the lower range of existing estimates (Fig. 7, Table S8.1) and difference in emissions from non-road machinery and agricultural (or prescribed) burning appears to be the key reason for observed discrepancies.

For Europe (including European part of Russia), the published studies of BC and OC (Bond et al., 2004; Kupiainen and Klimont, 2007; Schaap et al., 2004; see Table S8.1) compare well showing differences within ±10% or less with exception of EDGAR (Janssens-Maenhout et al., 2015) which shows much lower emissions but does not include any Russian territory. At the level of whole of Europe, GAINS calculates similar $PM_{10}$ and $PM_{2.5}$ emissions as officially reported to UNECE LRTAP Convention (www.ceip.at) while EDGAR estimate is nearly 40% lower for both species, however, does not include

Russia (Table S8.1). There have been only few published estimates of PM emissions in Russia (Table S8.1). For $PM_{10}$ and $PM_{2.5}$ in 2010, GAINS calculates higher emissions than EDGAR (Janssens-Maenhout et al., 2015) or national inventory submitted to LRTAP Convention (www.ceip.at) which covers only European part of Russian Federation; remarkably the total EDGAR estimate is similar to the national submission for European part. The main reasons for discrepancy are significantly larger GAINS emissions from industrial processes, residential combustion (these are very low in the national

submission – less than a quarter of EDGAR and GAINS estimates), agricultural burning, as well as inclusion of gas flaring. The uncertainties in volume of gas flared and actual emission factors are major reasons for the difference in estimated BC

---

[16] MEIC - Multi-resolution Emission Inventory for China; http://www.meicmodel.org





emissions in GAINS and Huang et al. (2015) who derived a much higher emission factor for this activity; for other sectors both studies report a fairly similar emissions of BC for 2010.

Yan et al. (2011) developed projections of $PM_{10}$ emissions from road transport sector (exhaust only). Their $PM_{10}$ estimates for 2000-2010 were about 1.65-1.75 Tg with a contribution from high emitters of about 0.3 Tg. The ECLISPE *V4a* results

are comparable Yan et al. (2011), while in *V5* and *V5a*, updates to the emission factors (reflecting more recent measurements, poor fuel quality, and maintenance) and penetration rates of control measures for developing countries (often delayed or postponed implementation of legislation) led to higher estimates of about 2.4-2.6 Tg, including  high emitters (0.4-0.5 Tg). Total GAINS model estimates for road transport also include non-exhaust emissions (brake, tyre, road abrasion) which add up to around 0.6 Tg $PM_{10}$.

Wiedinmyer et al. (2014) developed a new assessment of global emissions from burning of waste, including particulate matter. That study suggests that all current estimates largely underestimate emissions from this activity. Compared to GAINS, their emissions are nearly seven times larger and would make open burning of waste one of the key categories contributing between 10-15% of BC and $PM_{2.5}$ and nearly 30% of OC; considering anthropogenic sources. For example, waste burning could be responsible for three times more emissions of BC, OC, and $PM_{2.5}$ than agricultural waste burning or

about a third of the total transport sector emissions. Current GAINS estimates of 2010 emissions from open waste burning are about 1.4, 1.3, 0.1, 0.75 Tg $PM_{10}$, $PM_{2.5}$, BC, OC, while Wiedinmyer et al. (2014) calculated for the same species 12, 12, 0.632,  5.1 Tg. Obviously, large uncertainties remain in activity data and actual emission factors (see discussion in section 3.8) but this activity deserves more attention in the future.

### 4.3 Uncertainty in emission estimates

The completeness and quality of information about emission inventories varies across the regions, sectors, and species. The underlying information about several key PM sources like residential solid fuel combustion, brick production, and residual waste burning is often of poor quality or non-existing and that applies to both activity data and emission factors. In order to create a comprehensive emission data set, the national information is often supplemented with model estimates that rely on default parameterization; in fact, even many of the national inventories draw on the international data sets of emission

factors (e.g., EEA, 2013; US EPA, 1995) owing to lack of local measurements. Finally, the level of enforcement of existing laws, as well as the real-life performance of control technology is seldom sufficiently well-known and we tend to assume rather optimistically that both deliver and work as planned which has been shown to be often false (e.g., Stoerk, 2016; Xu, 2011; Xu et al., 2009), as, more recently, in  the so-called Dieselgate affair (e.g., Lange and Domke, 2015; US EPA, 2015a, 2015b). Consequently, the level of uncertainty, or confidence, varies widely across source sectors and regions.

We have not performed a formal uncertainty analysis for emission estimates in this study, but results of analysis from other studies are helpful and indicative of the expected uncertainties for various species and regions. For example, the global BC and OC inventory developed by Bond et al. (2004) included an uncertainty analysis of total emissions providing regional 'low-high' estimates for 1996. For BC emissions from anthropogenic sources, the range was 3.1-10 Tg yr$^{-1}$ (-30% to +120%)



and for OC 5.1-14 Tg yr$^{-1}$ (-40% to +130%). Estimates from the GAINS model presented in this study sit well within these ranges. The TRACE-P emission inventory for Asia (Streets et al., 2003) estimated the uncertainties for emission of carbonaceous particle (BC and OC) as 95% confidence intervals to be 160-500% for the developing countries and 80-180% for developed countries in Asia (Streets et al., 2003).

5    As indicated earlier, emissions of PM, including carbonaceous aerosols, belong to the most uncertain among the air pollutants as they form usually under poor combustion conditions in small inefficient installations burning poor quality fuels, which brings variability to the emission characteristics. Additionally there is very little information globally about local emission factors. Considering local data and knowledge about emission sources and their emission factors could significantly reduce uncertainties (Zhang et al., 2009).

10   In addition to the emission characteristics, the activity data also is a source of uncertainty. While for major industrial and transport sectors there are well documented and regularly updated national and international sources of activity data (e.g., IEA, 2015a, 2015b), the activities behind the major PM source categories, for example poor quality fuels in cook stoves or brick kilns, as well as local vehicle fleets, are not well known. For commercial fuels, however, the uncertainty has been estimated to vary from 2-3% for OECD countries to 5-10% for non-OECD (IPCC, 2006a).

15   A significant part of total aerosol emissions originate from open biomass burning, including forest fires, savannah, and agricultural residue burning (e.g., Reddington et al., 2015). Estimation of activity data and actual emission factors are bound with significant uncertainties which include, among others, amount of biomass burned and interannual variability (Chen et al., 2013; van der Werf et al., 2006; Wiedinmyer et al., 2011), drivers and impact of change in agricultural fires (Morton et al., 2008), and emission factors (Castellanos et al., 2014). The uncertainty ranges estimated by Bond et al. (2004) for BC and 20   OC emissions from open biomass burning  were 1.6 to 9.8 Tg yr$^{-1}$ (-45% to +185%) for BC and 31 to 58 Tg yr$^{-1}$ (-40% to +110%) for OC.

The uncertainties of emission estimates developed with integrated assessment models like GAINS are similar to the estimates for bottom-up inventories discussed above, at least at a regional scale. In fact, they could be even lower considering that they typically rely on a harmonized data set and include a simultaneous calculation of emissions of several 25   species using the same principal activity and technology data. Additionally, the error compensation, which is especially relevant if calculated emissions are the sum of a large number of equally important source categories (where the errors in input parameters are not correlated with each other), can lead to a further reduction of overall emission uncertainty (Schöpp et al., 2005). The GAINS model uncertainties, calculated in Schöpp et al. (2005), are consistent with the values reported by Streets et al. (2003) for developed countries. This analysis has also shown that at a finer scale the understanding of local 30   circumstances are critically important to reduce uncertainty, and while the emission factors were estimated to be the key factor determining uncertainty in historical emissions, at least for aerosol emissions, the uncertainty in activity assumptions becomes more important for the uncertainties in projected emissions.



## 5 Conclusions

To our knowledge, the estimates represent the first global dataset of anthropogenic emissions where size specific mass PM calculation, including BC and OC, was performed using a uniform and consistent estimation framework including a number of previously unaccounted or often misallocated emission sources, i.e., kerosene lamps, gas flaring, diesel generators, trash
burning that have been systematically evaluated for each region. Spatially, emissions were calculated for 170 regions and allocated to 0.5$^o$ x 0.5$^o$ longitude-latitude grids and are available either from the on-line GAINS model[17], where assumptions and results can displayed for 25 global regions (see section S7 in SI) or gridded emissions can be downloaded from the project website[18]. The ECLIPSE datasets do not include independent estimates of emissions from forest fires and savannah burning, windblown dust, and unpaved roads.

We estimate that global emissions of PM have not changed much between 1990 and 2010 but there are significantly different regional trends with North America, Pacific, and Europe reducing emissions by 30 to over 50% and Asia and Africa increasing by about 30%. Our new global estimate of BC emissions suggests higher numbers than previously published owing primarily to inclusion of new sources. Globally, over 75% of anthropogenic $PM_{10}$ and $PM_{2.5}$ originates from residential combustion, power plants and industry while for BC residential combustion and transport represent more than
75% but the importance varies across regions with Europe and North America having transport as key and rest of the world residential combustion.

We argue that this PM estimate reduces the gap in source coverage required in air quality and climate modelling studies and health impact assessments at a regional and global level as it includes both carbonaceous and non-carbonaceous constituents of primary particulate matter emissions, however, additional efforts need to be made to address several fugitive sources of
anthropogenic dust, e.g., unpaved roads. The ECLIPSE emission data sets have been used in several regional and global atmospheric transport and climate model simulations (AMAP, 2015; Eckhardt et al., 2015; Gadhavi et al., 2015; Lund et al., 2014; Quennehen et al., 2016; Stohl et al., 2013, 2015; Wobus et al., 2016; Yttri et al., 2014) where various aspects of several particulate matter species were addressed. The emissions developed during ECLIPSE also served as basis for a recently published global particulate number estimates (Paasonen et al., 2016).

We envisage development of further datasets drawing on the experience of the ECLIPSE exercise. The future versions will be available via the same on-line platform where additional documentation will be placed too. As a matter of fact, the GAINS model and the ECLIPSE dataset and scenarios have been already used as a starting point to develop emission data and mitigation strategies for the recently published International Energy Agency (IEA) World Energy Outlook special report on air pollution (IEA, 2016). Furthermore, the elements of the ECLIPSE data have been part of the contribution towards
improved representation of carbonaceous aerosols in the large scale integrated assessment models used in the development of the Shared Socio-economic Pathways (SSP) (O'Neill et al., 2014; Rao et al., in press; Riahi et al., in press)

---

[17] http://magcat.iiasa.ac.at/gains/IAM/index.login
[18] http://www.iiasa.ac.at/web/home/research/researchPrograms/air/Global_emissions.html



*Acknowledgements:* The research leading to these results has received funding from the European Union Seventh Framework Programme (FP7/2007-2013) under grant agreement no 282688 – ECLIPSE (Evaluating the Climate and Air Quality Impacts of Short-Lived Pollutants). We acknowledge the funding received from UNEP under the Small Scale Funding Agreement (SSFA) [SLP 2294-2H73-1111-2261] which allowed to improve the resolution of the GAINS model in Latin America and the Caribbean. We would like to thank Prof Qiang Zhang from Tsinghua University (Beijing, China) for the spatial distribution of Chinese power plants in 2000, 2005, and 2010 (that information was used in developing gridded emission fields) as well as Imrich Bertok and Robert Sander from IIASA for dedicated database programming support.

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



**Table 1.** Overview of the ECLIPSE emission data sets available to date

| Version | Release date | Period covered | Comments; key features |
|---|---|---|---|
| V3 | Nov 2013 | 2005, 2008, 2009, 2010 | Estimates for 2008 and 2009 based on activity proxies and trends in internationally reported emissions; activity data for 2010 based on the IEA World Energy Outlook 2011 (IEA, 2011) |
| V4a | Jan 2014 | 2005, 2010, *2030, 2050* | Major updates of EU-28 data (Amann et al., 2012) |
| V5 | Apr 2014 | 1990-2010[a], *2015-30[a], 2040, 2050* | IEA and FAO statistical data reimported for the period 1990-2010, international shipping included |
| V5a | Jul 2015 | 1990-2010[a], *2015-30[a], 2040, 2050* | China 12th 5-year plan included, improved regional resolution for Latin America, update of: global cement legislation, gas flaring, OC/OM ratios for residential combustion in Asia, Africa, Latin America, EU-28 update (Amann et al., 2015) |

[a] Estimated in 5-year intervals

**Table 2.** Residential-commercial sector fuel and source structure in GAINS

| Fuels | Non-specific | Lighting | Three-stone | Fireplace | Stove [a] | Household boiler Manual | Household boiler Auto | Medium boiler Manual | Medium boiler Auto |
|---|---|---|---|---|---|---|---|---|---|
| Gaseous fuels | • | | | | | | | | |
| Liquid fuels | • | • | | | | | | | |
| Charcoal | • | | | | | | | | |
| Coal | | | | | • | • | • | • | • |
| Biomass | | | | | | | | | |
|  - Fuelwood | | | • | • | • | • | • | • | • |
|  - Agricultural residue | | | • | | • | | • | | • |
|  - Dung cake | | | • | | • | | | | |

[a] distinguishing cooking and heating stoves as separate categories



**Table 3.** Mitigation measures distinguished in the residential-commercial sector in GAINS

| Control option | Non-specific | Lighting | Three-stone | Fireplace | Stove Cooking | Stove Heating | Household boiler Manual | Household boiler Auto | Medium boiler Manual | Medium boiler Auto |
|---|---|---|---|---|---|---|---|---|---|---|
| Improved | • | | | • | • | • | • | | | |
| New | | | • | | • | • | • | | | |
| Fan stove | | | | | • | | | | | |
| Coal briquettes | | | | | • | • | | | | |
| Hurricane lamp | | • | | | | | | | | |
| LED [a] lamp | | • | | | | | | | | |
| Pellets | | | | | | • | • | • | • | • |
| Cyclone | | | | | | | | | • | • |
| ESP [b] | | | | | | • | | • | • | • |

[a] Light Emitting Diode
[b] Electrostatic precipitator

**Table 4.** Brick sector technology structure assumed in GAINS for different regions

| Kiln type | East Asia [a] | South-East Asia [b] | Central Asia | Africa | Latin America and Caribbean | Other |
|---|---|---|---|---|---|---|
| Traditional clamp | • | • | • | • | • | |
| Downdraft | | • | • | • | • | |
| Moving chimney Bull's trench | | • | | | | |
| Fixed chimney Bull's trench | | • | | | | |
| Zig-Zag | | • | | • | | |
| Vertical shaft brick kiln | • | • | | • | • | |
| Marques kiln | | | | | • | |
| Hoffman kiln | • | • | • | • | • | |
| Tunnel kiln (coal) | • | • | • | • | • | • |
| Tunnel kiln (gas, oil) | • | • | • | • | • | • |

[a] Excluding OECD countries which are included in 'Other'
[b] Including Middle East





**Table 5.** Overview of sectoral layers included in the gridded ECLIPSE emissions of PM

| Sector layer | Included activities |
|---|---|
| Energy [a] | Power plants, energy production/conversion, fossil fuel distribution |
| Industry | Industrial combustion and processes |
| Residential | Residential and commercial combustion sources |
| Transport [b] | Road and non-road transport sources; including tyre and brake wear, road abrasion |
| Waste | Waste disposal, including trash burning |
| Agriculture | Livestock and arable land operations (ploughing, harvesting) |
| Agriculture (open burning) [c] | Open burning of agricultural residues (excluding forest and savannah burning) |
| Total | The sum of the above sectors |
| Shipping [d] | International shipping; available in version *V5* and *V5a* |

[a] Includes associated petroleum gas flaring which is also available as a separate gridded layer

[b] Does not include resuspension and international air and shipping; for the latter recommendation to use the RCP datasets, except for version *V5* and *V5a* where international shipping was also included

[c] The gridding proxy has been acquired from the GFED3.1 (van der Werf et al., 2010)

[d] Available as a separate file where all pollutants' emissions are included; the resolution of this layer is 1°x1°

**Table 6.** Particulate matter amplification factors for high emitting light- and heavy-duty diesel and gasoline vehicles used in the GAINS model

| | Light duty | | Heavy duty | |
|---|---|---|---|---|
| | diesel | gasoline | diesel | gasoline |
| No control | 3 | 6 | 3 | 4 |
| Euro 1/I | 3 | 6 | 3 | 4 |
| Euro 2/II | 5 | 6 | 5 | 10 |
| Euro 3/III | 5 | 10 | 5 | 10 |
| Euro 4/IV | 5 | 10 | 5 | - |
| Euro 5/V | 10 | 10 | 10 | - |
| Euro 6/VI | 10 | 10 | 10 | - |





**Table 7.** Regional emissions of particulate matter in 2010, ECLIPSE *V5a*, Gg year[-1]

| | $PM_{10}$ | $PM_{2.5}$ | $PM_1$ | BC | OC | OM |
|---|---|---|---|---|---|---|
| Africa | 9161 | 7973 | 6959 | 1347 | 3023 | 5207 |
| East Asia | 27172 | 20241 | 15291 | 2622 | 4974 | 7996 |
| Europe and Russia | 6027 | 4105 | 2781 | 660 | 897 | 1399 |
| Latin and Central America | 3736 | 2947 | 2358 | 508 | 994 | 1617 |
| North America | 1964 | 1268 | 917 | 249 | 382 | 594 |
| Pacific | 609 | 347 | 220 | 62 | 75 | 115 |
| South-West and Central Asia | 11982 | 9174 | 7654 | 1686 | 2796 | 4667 |
| International shipping | 1856 | 1758 | 1612 | 120 | 398 | 517 |
| International aviation [a] | 30 | 30 | 28 | 10 | 10 | 13 |
| Global anthropogenic | 62537 | 47843 | 37819 | 7264 | 13548 | 22125 |
| Forest and savannah fires [b] | 48207 | 33014 | 33014 | 2268 | 19489 | 31363 |
| Global total | 110744 | 80858 | 70834 | 9532 | 33037 | 53489 |

[a] Values are middle of the range estimates referring to the ranges reported in Settler et al. (2013), Yim et al. (2015), and based on global fuel consumption and ranges of emission factors from Kinsey (2009)
[b] GFED3.1 without agricultural waste burning; $PM_{10}$ value based on TPM (total particulate matter); $PM_1$ not available in GFED – here assumed equal $PM_{2.5}$

**Table 8.** Sectoral emissions of particulate matter in 2010, ECLIPSE *V5a*, Gg year[-1]

| | $PM_{10}$ | $PM_{2.5}$ | $PM_1$ | BC | OC | OM |
|---|---|---|---|---|---|---|
| Agriculture | 6555 | 3848 | 2883 | 337 | 1313 | 2364 |
| Residential combustion | 23078 | 21857 | 20742 | 4163 | 8852 | 15329 |
| Industrial processes | 12162 | 8340 | 4135 | 462 | 633 | 823 |
| Large scale combustion | 11561 | 6420 | 3812 | 136 | 164 | 248 |
| Oil & gas, mining | 1706 | 571 | 412 | 226 | 93 | 120 |
| Transport – road | 3339 | 2925 | 2524 | 1349 | 1116 | 1451 |
| Transport – non-road | 861 | 823 | 795 | 363 | 217 | 283 |
| Waste | 1388 | 1272 | 876 | 97 | 751 | 977 |
| International shipping | 1856 | 1758 | 1612 | 120 | 398 | 517 |
| International aviation [a] | 30 | 30 | 28 | 10 | 10 | 13 |
| Global anthropogenic | 62537 | 47843 | 37819 | 7264 | 13548 | 22125 |
| Forest and savannah fires [b] | 48207 | 33014 | 33014 | 2268 | 19489 | 31363 |
| Global total | 110744 | 80858 | 70834 | 9532 | 33037 | 53489 |

[a] Values are middle of the range estimates based on the ranges reported in Settler et al. (2013), Yim et al. (2015), and based on global fuel consumption and ranges of emission factors from Kinsey (2009)
[b] GFED3.1 without agricultural waste burning that is included based on GAINS estimates in category 'Agriculture'; $PM_{10}$ value based on TPM (total particulate matter); $PM_1$ not available in GFED – here assumed equal $PM_{2.5}$



**Table 9.** Comparison of global anthropogenic emissions of BC by sector, Gg year$^{-1}$

| | 1995 | | 2000 | |
|---|---|---|---|---|
| | Bond et al. (2004)[a] | This study (*V5a*) | Bond et al. (2013) | This study (*V5a*) |
| Diesel engines – road | 792 | 872 | 840 | 980 |
| Diesel engines – off-road | 579 | 415 | 470 | 432 |
| Residential combustion | 2046 | 3703 | 1880 | 3891 |
| *of which:* | | | | |
| *Biomass cooking* | *1481* | *1660* | *1290* | *1711* |
| *Biomass heating* | | *411* | *260* | *392* |
| *Residential coal* | *480* | *710* | *330* | *908* |
| *Other* [b] | *85* | *922* | [c] | *880* |
| Agricultural burning | 328 | 323 | 330 | 326 |
| Industrial coal [d] | 642 | 282 | 740 | 315 |
| Other [e] | 610 | 612 | 600 | 649 |
| Global anthropogenic | 4997 | 6206 | 4870 | 6594 |

[a] Estimates for 1996

[b] GAINS includes oil appliances and kerosene lamps – the latter are estimated in GAINS at 750 and 692 Gg BC in 1995 and 2000

[c] Other residential sources (oil) included in category 'Other'

[d] Includes coke and brick production, coal boilers and furnaces

[e] Includes power plants, gas flaring, waste, gasoline engines in transport; for Bond et al. also oil use in residential sector





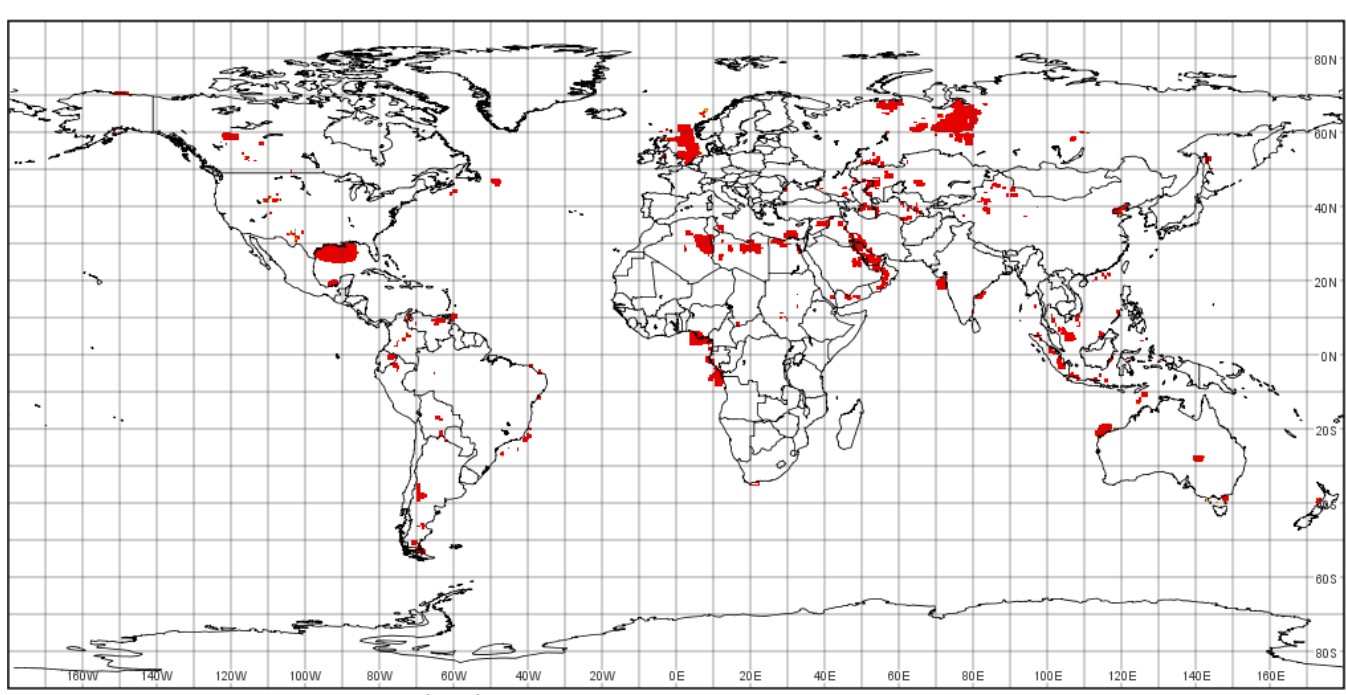

**Figure 1.** Global distribution of grids (0.5°x0.5°) for which flaring of associated petroleum gas emissions were calculated; derived from the 2009 data from Elvidge et al. (2011).





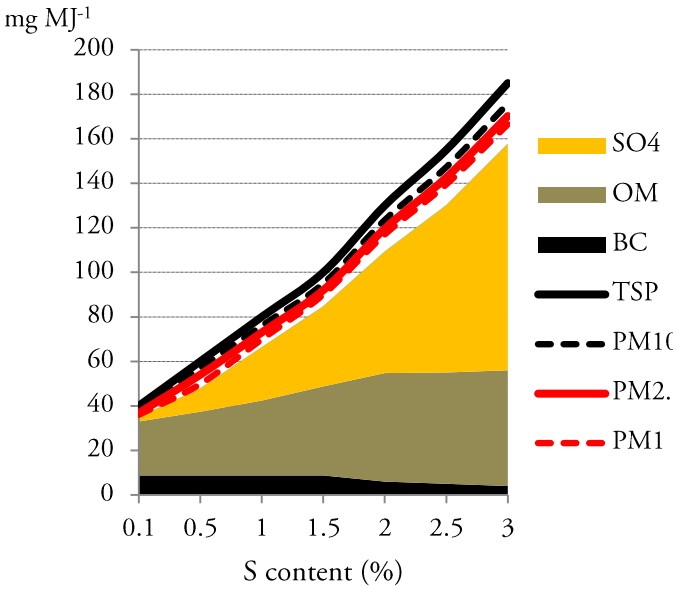

PM emission factors
(mg MJ$^{-1}$)

| | Sulphur content (%S) | | | | |
|---|---|---|---|---|---|
| | 0.1 | 0.5 | 1 | 2 | 3 |
| PM$_{10}$ | 38 | 57 | 76 | 124 | 176 |
| PM$_{2.5}$ | 37 | 54 | 73 | 120 | 167 |
| BC | 8.8 | 8.8 | 8.8 | 6 | 4 |
| OC | 17 | 20 | 24 | 35 | 37 |

**Figure 2.** Particulate matter emission factors for shipping used in the GAINS model




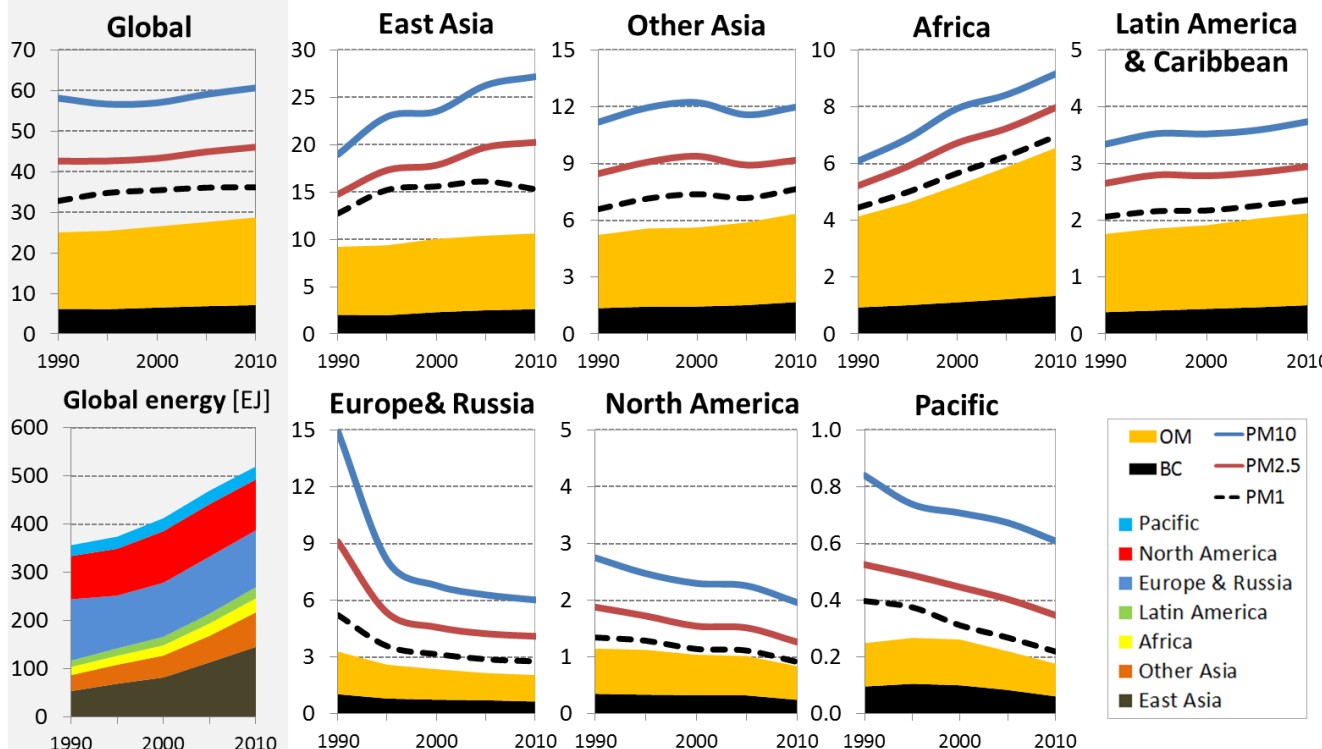

**Figure 3.** Global and regional emissions of PM species [Tg] and global energy consumption [EJ] in the period 1990-2010, ECLIPSE *V5a*.





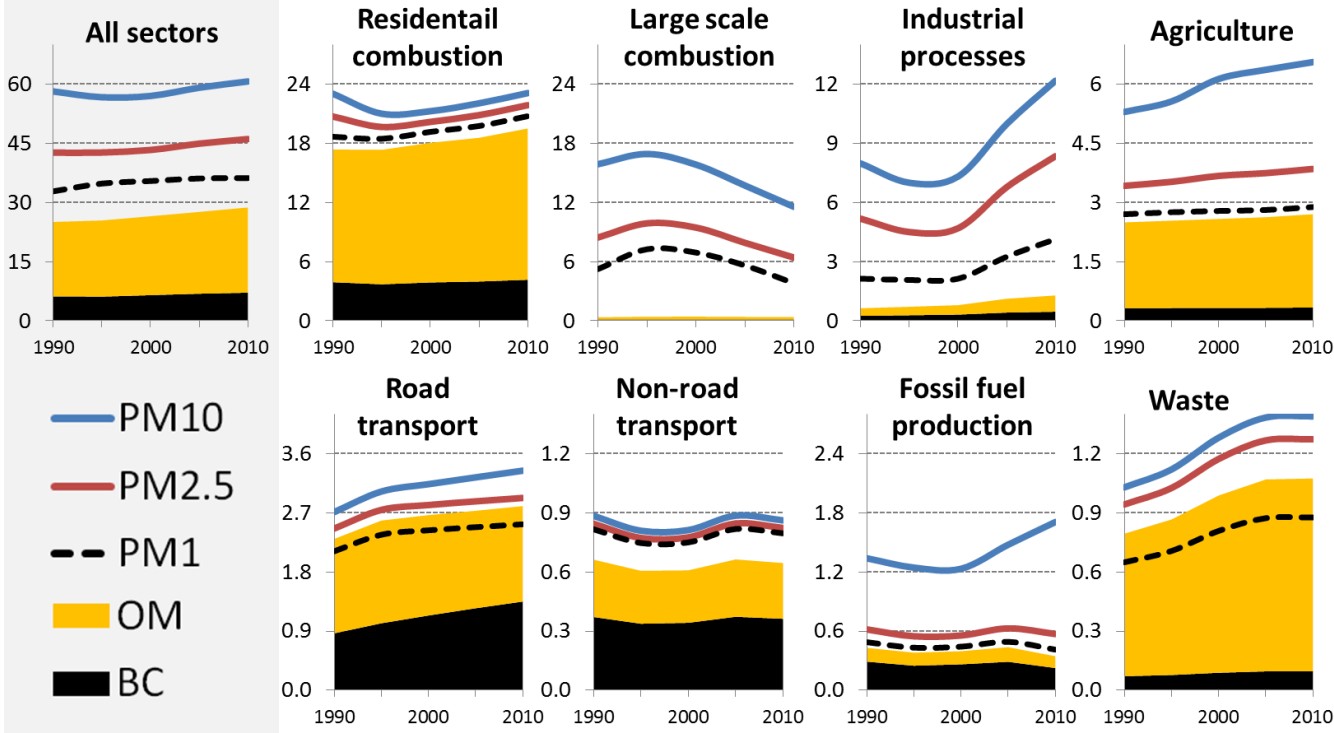

**Figure 4.** Global and sectoral emissions of PM species [Tg] in the period 1990-2010, ECLIPSE *V5a*.



**Figure 5.** Global distribution of emissions of $PM_{2.5}$ (left) and BC (right) in 2010 [Gg year$^{-1}$ per grid] from land-based sources, ECLIPSE *V5a*; the scale is the same across sectors but there is a factor ten between $PM_{2.5}$ and BC



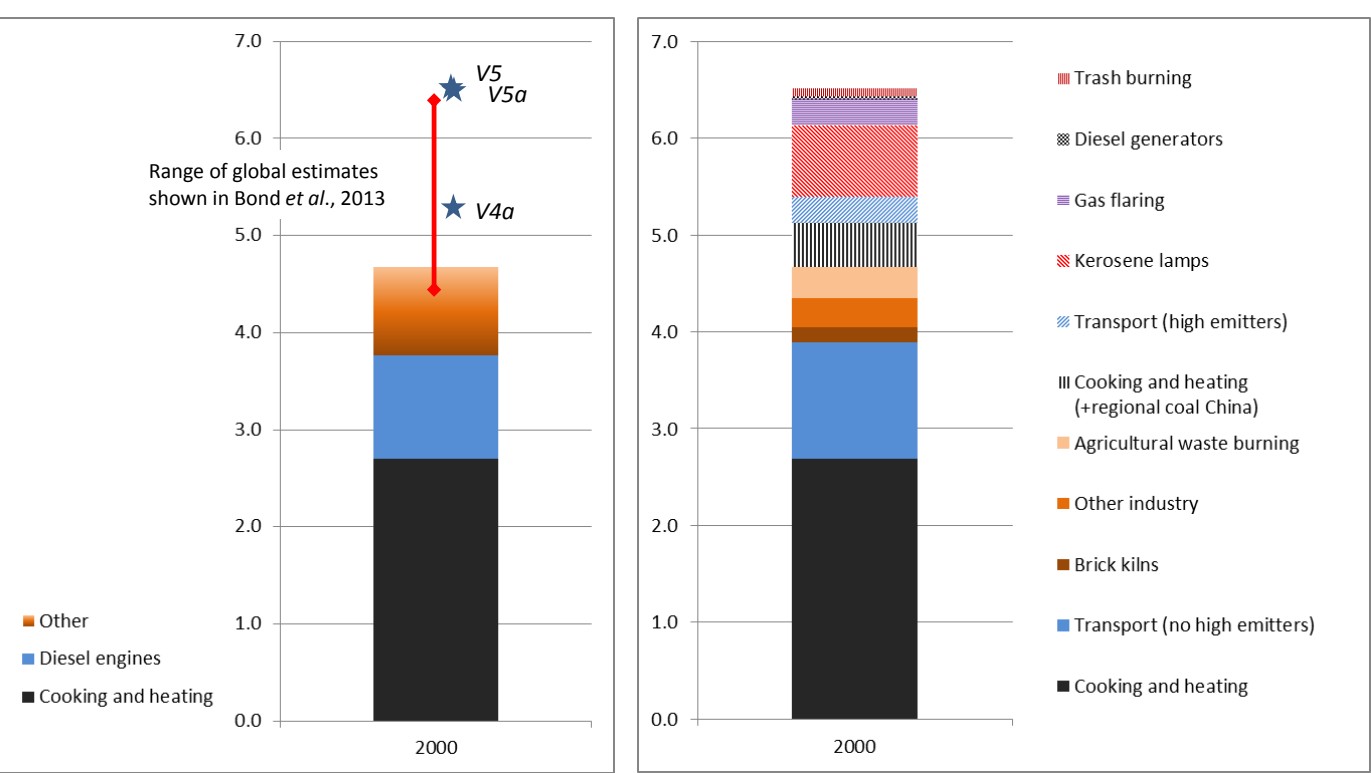

GAINS; excluding 'new/re-estimated' sources          GAINS; all sources

5    **Figure 6.** Source-sector distribution of global anthropogenic emission of BC estimated with the GAINS model (ECLIPSE V5a) for the
year 2000, Tg year[-1]







**Figure 7.** Comparison of black carbon emission in this work (ECLIPSE *V5a*) with Lamarque et al. (2010) and Granier et al. (2011)