# Peer review of "Global anthropogenic emissions of particulate matter including black carbon"

_Atmospheric Chemistry and Physics, 2016_

## Referee Comment (RC1) · Anonymous Referee #1 · 21 Nov 2016

The paper reports on the development of a global inventory for spectated, primary particulate matter emissions. To my knowlege, this is the most thorough effort of its type and is an important contribution. Overall, the methodology and assumptions are well described.

The description of the methodology is strong, and the overall results are well described. The paper could use more presentation of intermediate results. The calculations often incorporate substantial technology and emission control details, however the results of this detail are only presented at a fairly aggregate level. Some key intermediate results here would be very useful to present. In particular, for some key sector/fuel combinations, I suggest that the emission factors over time for different regions (perhaps PM or BC in the main paper, and other species in the supplement). One key sector is on-road diesel, for example, where aggregate emission factors have changed over time

in many regions. Other key sectors might be residential biomass, off-road diesel, etc. Where these emission factors are largely constant, this could be mentioned in the text without a figure, but where these have changed some figures and/or tables and some discussion would be useful.

The main presentation focuses on emissions by region, and global emissions by sector. There is a lot of material here, and this is a reasonable choice for the main paper, however many readers will want to see emissions by sector for specific regions. It would be useful to present the equivalent of Figure 3 for the different regions in the supplement. (Hopefully the codes that generated that figure can readily be generalized.). As noted below the authors should also provide an electronic supplementary file with more detailed emissions. Some further suggestions for details that would be useful to supply are below.

Overall, its not clear how equipment vintages are treated. There is a mention of old/new power plants, but not as much discussion for other sectors. For vehicles, for example, is the model based on aggregate emission factors by year, or are vintages of vehicles tracked over time? This should be clarified in the manuscript.

Specific comments Section: 2.2.1 Residential combustion: cooking, heating, lighting

It is not clear if assumptions such as the use fraction of different technologies and also splits between end-use services (e.g. cooking, heating in particular) are constant over time within a given country or region?

There is quite a bit of good work here and some sort of summary later on in terms of the evolution of aggregate emission factors (either over time – if the above assumptions change over the historical period) would be useful. For example, do PM emissions per unit biomass fuel use for cooking change over time in a given region? (Some of this could be in the supplement, with main points summarized in the text). While there are too many details to do this comprehensively, it would be important to summarize where emission factors for important sectors change over time (either as a result of

different technology fractions, or emission controls). This is clearly going to be the case for diesel fuel use in road transportation in many regions, but how much change was assumed for the various regions. How about off road, agriculture, or residential sectors? Did controls or technology mix (beyond shifts in type of fuel use) have a noticeable impact on emission factors in these regions?

Pg 14 - "the independent fuel estimate by Denier van der Gon et al. (2015).". van der Gon et al. state "A consistent set of PM10, PM2.5 and PM1 emission data for Europe was obtained from the GAINS model...", so these do not appear to be independent.

pg 16 - "The resulting fuel use was compared and calibrated to the diesel consumption reported in the power and commercial sector." Not clear what this means.

pg 17, last portion of diesel generator section. There is some discussion of emission reduction options, but no mention the extent to which these were assumed to be applied in the emission estimates.

pg 17 - line 25. IEA energy statistics contain separate lines for agriculture and construction - while these are not available for all years, these data seem to be becoming more complete in more recent years, even for non-OECD countries. It appears that this data is not used? Some further explanation would be useful.

pg 17 - line 31 "Also old and often poorly maintained vehicle fleet is reflected in measurements of emission factors" - not clear what is meant here.

pg 20 line 14. This "fuel consumption data for 2007 were extrapolated to 2010 using GDP". The result of this assumption should be compared to the fuel consumption estimates from IMO, who compared both bottom up and top down methodologies. ("Third IMO Greenhouse Gas Study 2014")

Page 23 top regarding coke ovens. It is not quite clear what the technology representation is here. "uncontrolled ovens" are mentioned, is the split between controlled and uncontrolled? Is this assumed to change over time? It appears there is little detail in

terms of emission factors available in the literature (and emissions seem likely to depend on site specific characteristics in any event), so some comment might be useful on what are the most important data that would be needed to improve estimates.

Page 26. Might be useful to also mention that Agricultural waste burning can also be seasonally concentrated, so that it might be particularly important in come months.

Page 26 line 12. "This database has been further extended and updated" perhaps edit to clarity that this refers to the data presented in the current paper.

Page 27 "3.9 Other sources" Is dust included? Its not mentioned until the discussion section.

Page 28 line 13 - " for barbeques, a per capita emission factor is established, i.e.,...". Presumably this varies by region?

Page 32 line 19-20. Probably useful note here that the emissions in Granier et al., past about 2000/2005 were based on projections, where as the estimates here up to 2010 are (to the extent available) based on reported data and practices.

Page 35 line 3, this presumably is a typo? "confidence intervals to be 160-500% for the developing countries"

Page 35 while "error compensation" is, indeed important, it might be useful to note that this might only be partial compensation. (e.g., some errors, such as measurement or enforcement issues, could be correlated across sectors.)

Wasn't sure about the meaning of this sentence: "In fact, they could be even lower considering that they typically rely on a harmonized data set and include a simultaneous calculation of emissions of several species using the same principal activity and technology data." I would presume most country-level inventories are similar in this respect?

Would be useful to mention in this section that there is also uncertainty in the speciation

fractions (but this is constrained across species since these must sum to <= 1).

Page 36 In addition to this "Our new global estimate of BC emissions suggests higher numbers than previously published...", perhaps useful to also mention something about BC trends over time here (since there is substantial interest in BC, and it looks like BC trends can be different from PM trends).

Supplement When GAINS values are listed in the tables, these are sometimes listed as ranges. I assume these are not uncertainty ranges (as in some of the other ranges in the table), but are GAINS central values and that the range represents the range used in different GAINS regions? It would be useful to clarify this.

I suggest providing a more detailed summary of the emissions data. It would make this data more readily useful to the community to have an electronic file (either csv or excel) that provides emissions of the various species by country/region and by sector and fuel (I appreciate that some aggregation with regard to sector/fuel might be necessary). I realize much of this (or perhaps all of this) would be available on-line, but providing this in a supplement will be more accessible and also provide for an archival record of these important results.

---

## Referee Comment (RC2) · Anonymous Referee #2 · 20 Jan 2017

The paper by Klimont et al. describes in details an emission dataset, ECLIPSE, which is already used by the atmospheric modeling community. This paper therefore provides very useful information on the emissions provided by this inventory.

A few comments on the paper:

The title mentions that the paper focuses on anthropogenic emissions. However, the paper also discusses open fires. Since the paper is already very long, it might be better to only focus on anthropogenic emissions as stated in the title. The inclusion of emissions from fires (which come from other authors) is a bit confusing.

Abstract and line 24, page 2: the abstract claims this paper is " the first comprehensive assessment of historical (1990-2010) global anthropogenic particulate matter (PM)...". However, the EDGAR4.3 inventory described in Crippa et al. (2016) provides emis-

Interactive
comment

sions for 1970-2010 for PM2.5 and PM10. The statement about being "the first comprehensive assessment" is true for PM1, but not for the other species. Please rephrase.

Page 18, line 6: " exceptions are old vehicles running on leaded gasoline and pre-regulation 2-stroke mopeds . . . while latest gasoline direct injection engines have PM mass emissions comparable or even higher than latest diesel engines with particle filter, however, the absolute level is about one order of magnitude lower than for older generations. This sentence is not clear. What does "absolute level" refer to?

Page 26, line 20: The authors use quite old data for emission factors for agriculture waste burning. Akagi et al. (Atmos. Chem. Phys., 11, 4039–4072, 2011) have published a more recent and detailed review of all data available on emission factors. The authors should indicate why they did not use this more recent review.

Page 29, lines 1-4: these lines should be rephrased. Many recent global chemistry-transport and chemistry-climate models now include detailed aerosols schemes, and PMs distribution are calculated as the sum of the mass of all the components included in the models. Maybe a few older models use the "BC + 1.4 OC" formula to calculate the mass of PM, but the recent models are much more advanced and calculate the mass of PMs in a more accurate way.

Page 31, lines 14-15: The sentence starting with "combined . . ." is unclear

Page 25 of the supplement: the authors should add in their table the TNO-MACC and TNO-MACCII (Kuenen et al., ACP, 2014) inventories, which provide emissions of PM for Europe and neighboring countries. The TNO-MACC inventories are now becoming a reference for atmospheric modeling in Europe, and these emissions should be mentioned in the paper.

Page 25 of the supplement: The emissions provided by US EPA are given as the sum of anthropogenic and wildfires. The dataset provided by EPA (note that the las release of the emissions is 2016 and not 2011 as mentioned in the supplement) provides emissions with and without wildfires. It would be better to include the emissions without wildfires, in order to be consistent with the other data in the table.

---

## Author Response (AR2)

**Responses to reviewers of the paper by**
**Klimont et al. ([http://www.atmos-chem-phys-discuss.net/acp-2016-880/](http://www.atmos-chem-phys-discuss.net/acp-2016-880/))**

**RESPONSES TO Reviewer 1:**

We thank the reviewer for a very thorough and insightful review which we have used to improve the manuscript and provide additional results in further extended Supporting Information set. The responses to specific points raised by the reviewer are provided below.

REVIEWER:
The calculations often incorporate substantial technology and emission control details, however the results of this detail are only presented at a fairly aggregate level. Some key intermediate results here would be very useful to present. In particular, for some key sector/fuel combinations, I suggest that the emission factors over time for different regions (perhaps PM or BC in the main paper, and other species in the supplement). One key sector is onroad diesel, for example, where aggregate emission factors have changed over time in many regions. Other key sectors might be residential biomass, off-road diesel, etc. Where these emission factors are largely constant, this could be mentioned in the text without a figure, but where these have changed some figures and/or tables and some discussion would be useful

RESPONSE: Indeed, the model considers explicitly implementation of particular technologies to achieve required emission standards. We produced now a specific output for selected sectors (on-road diesel, off-road diesel, power –coal, industry-coal, residential biomass) and all key regions. On the basis this output we present a few examples where changes were relevant within the characterized period.

REVIEWER:
The main presentation focuses on emissions by region, and global emissions by sector. There is a lot of material here, and this is a reasonable choice for the main paper, however many readers will want to see emissions by sector for specific regions. It would be useful to present the equivalent of Figure 3 for the different regions in the supplement. (Hopefully the codes that generated that figure can readily be generalized.). As noted below the authors should also provide an electronic supplementary file with more detailed emissions. Some further suggestions for details that would be useful to supply are below.

RESPONSE: We added now two additional outputs:

- The set of tables presenting emissions for each of the 25 IMAGE regions by key sectors (power-coal, power-other fuels, industry, coke ovens, fossil fuel production, residential – biomass, res –coal, res – other fuels, road transport –diesel, road transport – other fuels, non-road – diesel, non-road – other fuels, other). This is an Excel file that will be part of the SI.

- A set of figures for key regions (as in Figure 3) with emissions for PM species over time as in Figure 4. We believe that the reviewer meant Figure 4 for key regions rather than Figure 3 since the latter is already regional. These additional figures are in the SI.

REVIEWER:
Overall, its not clear how equipment vintages are treated. There is a mention of old/new power plants, but not as much discussion for other sectors. For vehicles, for example, is the model based on aggregate emission factors by year, or are vintages of vehicles tracked over time? This should be clarified in the manuscript.

RESPONSE: The equipment vintages are dealt in GAINS in two ways. Explicit assumptions for power sector where activity (energy use) for existing and new (post 2000) power plants are included, and specifying pace of equipment replacement, using technical lifetime of technologies in GAINS database which is defined for each control technology, in the so called 'control strategies' where share of fuel use for a given technology is given. While for many sectors, the add-on control technologies can be applied at any time and vintage plays smaller role, for transport new standards most of the time are synonymous with a new vintage year of a particular vehicle. Vehicle lifetimes are region specific and affect the fleet turnover which in turn determines how quickly a new technology penetrates the market. As for vehicle emission factors, in GAINS they represent for each of the categories, e.g., EURO 1, EURO 2 or any other standard, the average lifetime emission rate, including typical deterioration factors. We add additional text in the manuscript to explain how vintages of installations and technologies are dealt with – see section 2.2.2 and 2.2.3.

REVIEWER:
Specific comments Section: 2.2.1 Residential combustion: cooking, heating, lighting: It is not clear if assumptions such as the use fraction of different technologies and also splits between end-use services (e.g. cooking, heating in particular) are constant over time within a given country or region?

RESPONSE: There is an explicit region-specific assumption in the model about the fuel use for cooking and heating. For Europe and North America solid fuels in residential sector are allocated to heating unless specific information exist, e.g., Switzerland provided their estimates for cooking. We are aware that there is some cooking (or cooking and heating using the same devices) in several countries but there is no data and the cooking share is most likely very small compared to demand for heating. For Asia, Africa, and Latin America cooking dominates although for some countries like China and few of the Latin American countries heating plays an important role; similarly in some provinces (states) in India, Pakistan, or Nepal. The data for that originates from the GAINS-Asia related studies as well as the recent work in Latin America with support of CCAC and UNEP (final report in preparation for publication; see summary for policy makers http://www.ccacoalition.org/en/resources/integrated-assessment-short-lived-climate-pollutants-latin-america-and-caribbean-summary). The share of cooking/heating is assumed constant in the 1990-2010 period as we have not found any data allowing to change that

assumption. The manuscript has been modified to reflect the above discussion by adding additional references (see 2nd paragraph of section 2.2.1) and a specific statement about the constant ratio of cooking/heating (see the last sentence in section 2.2.1 which reads "*The ratio of cooking/heating is assumed constant in the 1990-2010 period as we have not found any data allowing to change that assumption*"

As far as technology split is concerned; there is no firm statistical data but for several countries our exchange with national experts led to adjustment of assumptions how the technology (stoves/boilers/automatic boilers/pellet stoves/fireplaces) shares changed over time. Again, there is no model behind but an attempt to reflect on available information from local studies or expert contacts. For example sales statistics, for example for pellet stoves and boilers, resulted in adjustment of shares of biomass used in such installations in several European countries where strong growth has been observed towards the end of the period under investigation and continues into the future (another paper). Also for China, trends towards cleaner coal stoves and more household coal boilers (in specific provinces) were taken into account. The manuscript has been modified including explicit statement that the technology shares change over time and addition of new references (see 1st paragraph in section 2.2.1).

Finally, residential use of kerosene was split into cooking and lighting and the shares change over time depending on the regional access to electricity, as described in the manuscript (see section 3.2).

REVIEWER:
Specific comments Section: 2.2.1 Residential combustion: cooking, heating, lighting: There is quite a bit of good work here and some sort of summary later on in terms of the evolution of aggregate emission factors (either over time – if the above assumptions change over the historical period) would be useful. For example, do PM emissions per unit biomass fuel use for cooking change over time in a given region? (Some of this could be in the supplement, with main points summarized in the text). While there are too many details to do this comprehensively, it would be important to summarize where emission factors for important sectors change over time (either as a result of  different technology fractions, or emission controls). This is clearly going to be the case for diesel fuel use in road transportation in many regions, but how much change was assumed for the various regions. How about off road, agriculture, or residential sectors? Did controls or technology mix (beyond shifts in type of fuel use) have a noticeable impact on emission factors in these regions?

RESPONSE: As already mentioned in one of the earlier points, we are developing implied emission factors for a number of sectors where changes in structure of the sector or increased penetration of control technology made an impact. Examples include heavy duty trucks where we estimate for BC implied emissions factor declined globally by nearly 20% (2010 to 1990) but in several regions like North America, Western Europe, Japan the reduction was over about 60-65%, Central Europe about 40-50%, but for Russia less than 10% and for most other regions no significant change was estimated (for several regions like China the impact of legislation is visible only in 2015 and later). Another example includes coal power plants where globally emission factors for PM2.5 dropped by about 45% with NA, Western

Europe, Japan having over 80% decline and even for China we estimated over 70% reduction while in Russia and several FSU countries only 20-30% decline. For industrial coal use lower reductions were achieved with exception of Eastern Europe and some FSU where collapse of heavy industry in the period 1990-2000 resulted in decline of emission factors by over 90% compared to the 1990 period. Finally, for residential heating (fuelwood) the 'global average factor' declined by about 15% which is mostly due to moderate changes in North America, Japan but mostly Western Europe where nearly 40% decline was estimated owing to strong increase in biomass use but in new installations including hundred thousands of pellet stoves and boilers. We will include brief discussion of these trends and illustrate it on a chart in the manuscript – see sections 221, 222,223,224 and figures 1,2,3,

REVIEWER:
Pg 14 - "the independent fuel estimate by Denier van der Gon et al. (2015).". van der Gon et al. state "A consistent set of PM10, PM2.5 and PM1 emission data for Europe was obtained from the GAINS model...", so these do not appear to be independent

RESPONSE: We need to point out that our statement refers to the 'independent fuel use' estimates by van der Gon et al. Indeed, they relied on GAINS emission factors but estimated fuelwood consumption independently rather than relying on statistical data. Therefore, we believe that this is an accurate statement.

REVIEWER:
pg 16 - "The resulting fuel use was compared and calibrated to the diesel consumption reported in the power and commercial sector." Not clear what this means.

RESPONSE: For DG sets, fuel use is estimated from the number and size of DG sets in some regions (i. e. Nepal, Nigeria). For some regions, share of diesel used in DG sets as a percentage of total diesel consumption in the country/region is available (i. e. India). Once the total diesel consumption by country/region is estimated then the fuel use for DG sets in GAINS model is allocated from commercial and power sector. We have revised the statement in the manuscript that reads now: "*The resulting fuel use was compared to the IEA statistics for the power and commercial sector and adjusted if necessary so that the overall energy use is consistent with the IEA.*"

REVIEWER:
pg 17, last portion of diesel generator section. There is some discussion of emission reduction options, but no mention the extent to which these were assumed to be applied in the emission estimates.

RESPONSE: Currently we have not applied any of the post-combustion control measures for DG sets in the developing world where they matter most; at least regionally. We modify that section and add

explicit statement that no post-combustion measures are implemented for developing countries in this period.

The last paragraph of section 3.3 includes now the following sentence: "*While it is possible to achieve emissions reductions from diesel combustion through engine modifications and post-combustion measures, we assume that in the period 1990-2010 DG sets operating in the developing world lack any such controls. In case new information will become available and for future implementation of respective policies, the GAINS model includes a number of post-combustion control technologies such as diesel particulate filters (DPFs), diesel oxidation catalysts (DOC), and fuel-borne catalysts (FBC) offering reduction of gaseous and particulate emissions (Herzog, 2002; Yelverton et al., 2016).*"

REVIEWER:
pg 17 - line 25. IEA energy statistics contain separate lines for agriculture and construction - while these are not available for all years, these data seem to be becoming more complete in more recent years, even for non-OECD countries. It appears that this data is not used? Some further explanation would be useful.

RESPONSE: In fact we do make use of this information but this (IEA statistics) data is for total diesel consumption in agriculture or industry and not specifically in mobile machinery. For example, in agriculture diesel is used also for irrigation purposes (we attempt to allocate this to diesel generation sets in GAINS as the operating conditions are similar) but also in heating boilers used for example for drying or heating. The text in the manuscript does specifically refer to mobile machinery information and therefore we believe it is a valid statement.

REVIEWER:
pg 17 - line 31 "Also old and often poorly maintained vehicle fleet is reflected in measurements of emission factors" - not clear what is meant here.

RESPONSE: We agree, this statement is unclear and we replace it with: "*For all world regions we assume that a certain fraction of vehicles is badly maintained (e.g., Mancilla et al., 2012) or their emission controls tampered which is reflected as the share of so-called high-emitters (McClintock, 1999, 2007; Smit and Bluett, 2011; Yan et al., 2011, 2014); see further discussion in section 3.4.1.*"

REVIEWER:
pg 20 line 14. This "fuel consumption data for 2007 were extrapolated to 2010 using GDP". The result of this assumption should be compared to the fuel consumption estimates from IMO, who compared both bottom up and top down methodologies. ("Third IMO Greenhouse Gas Study 2014")

RESPONSE: We compared the result of our extrapolation for 2010 (1056 Mt $CO_2$ (13.83 EJ)) with the data in IMO 3rd GHG Study 2014 showing: [Table 1]

- For 2010: Total shipping: 915 mio t CO2 / International shipping: 771 mio t CO2
- Average 2007-2012: 1015 mio t CO2 / 846 mio t CO2

Our extrapolation is slightly higher than the average for the period 2007-2012 and about 10% higher than the reported fuel use in 2010 by IMO. We have added a remark and reference to IMO in the manuscript.

REVIEWER:
Page 23 top regarding coke ovens. It is not quite clear what the technology representation is here. "uncontrolled ovens" are mentioned, is the split between controlled and uncontrolled? Is this assumed to change over time? It appears there is little detail in terms of emission factors available in the literature (and emissions seem likely to depend on site specific characteristics in any event), so some comment might be useful on what are the most important data that would be needed to improve estimates.

RESPONSE: Indeed, coke industry PM emissions are poorly understood and this is of special importance for China in the last decade. We highlight the poor data availability with respect to measurements and add at the end a sentence about the assumption regarding the change in emission factor over time, i.e., "*Owing to lack of specific data for various world regions, we assume little change in emissions factors over time for the developing world, although the transition in China reported in Huo et al. (2012) was considered, and for OECD countries emission factor trend follows reported emissions, where available.*"

REVIEWER:
Page 26. Might be useful to also mention that Agricultural waste burning can also be seasonally concentrated, so that it might be particularly important in come months.

RESPONSE: We have rewritten this part which now reads: "*At the same time, for several regions this source might be even more important, e.g., for Brazil we estimate its contribution at up to 15% of PM2.5 and 10% of BC emissions. Finally, agricultural burning has a strong seasonal pattern (see also section 2.4.1.) and has been also linked with heavy smog and haze episodes (e.g., Mukai et al., 2015; Stohl et al., 2007).*"

REVIEWER:
Page 26 line 12. "This database has been further extended and updated" perhaps edit to clarity that this refers to the data presented in the current paper.

RESPONSE: We have made that explicit now by joining this sentence and the next one that reads now: "*Niemi (2007) compared various datasets for all open biomass sources and developed the first global activity set for the RAINS model drawing on EDGAR3.2FT2000 (Van Aardenne et al., 2005) which we have*

*further extended and updated to accommodate other data sources allowing gaps to be filled for several countries.*"

REVIEWER:
Page 27 "3.9 Other sources" Is dust included? Its not mentioned until the discussion section

RESPONSE: We have added a sentence at the end of the first paragraph of this section: "*Note that windblown dust and emissions from unpaved roads are not included (see also introduction to section 3)*." As indicated here, there are few words of explanation about what is included and what not in the introductory section of the section 3 of the manuscript.

REVIEWER:
Page 28 line 13 - " for barbeques, a per capita emission factor is established, i.e.,...". Presumably this varies by region?

RESPONSE: Unfortunately, we found very little data allowing differentiating between the countries and therefore for most countries the same factor is used, except for few countries in Europe where national experts contributed their input leading to adjustment. We added a comment in the text reflecting that.

REVIEWER:
Page 32 line 19-20. Probably useful note here that the emissions in Granier et al., past about 2000/2005 were based on projections, where as the estimates here up to 2010 are (to the extent available) based on reported data and practices.

RESPONSE: Indeed, the numbers reported for 2010 represent various projections in Granier et al; in fact including also RCP numbers; this justifies also the widening emission range shown towards the end of the period they have investigated. We add a comment about it at the end of the line the reviewer referred to, which now reads: "….stabilization shown in earlier studies; *note that values reported in Granier et al. (2011) for 2010 were results of projections.*"

REVIEWER:
Page 35 line 3, this presumably is a typo? "confidence intervals to be 160-500% for the developing countries"

RESPONSE: As a matter of fact it is not a typo; see table 8 and discussion on page 18 of Streets et al (2003) manuscript. However, we choose to delete this statement from the paper as in view of the newer work (Bond et al, 2004, 2013), quoted already, where also a similar emission methodology was used as

here, this particular statement does not add any specific insight unless elaborated further discussing specifically reasons for very high uncertainties estimated in that study.

REVIEWER:
Page 35 while "error compensation" is, indeed important, it might be useful to note that this might only be partial compensation. (e.g., some errors, such as measurement or enforcement issues, could be correlated across sectors.)

RESPONSE: Indeed, the poor enforcement might be one of the factors that cuts across the sector while for measurements of emission factors it could be also the case, often measurement techniques and teams performing them will be different reducing such potential. Our statement in the paper highlights the fact that the error compensation 'works' when errors are not correlated and so it is either known or assumed based on well-founded knowledge. We have added a respective comment in the paper and now the whole statement reads: "*Additionally, the error compensation, which is especially relevant if calculated emissions are the sum of a large number of equally important source categories (and where the errors in input parameters are not correlated with each other), can lead to a further reduction of overall emission uncertainty (Schöpp et al., 2005). A careful assessment of the assumption about correlation between input parameters is essential as for example poor enforcement of legislation or measurements errors could affect several source sectors in a similar way.*"

REVIEWER:
Wasn't sure about the meaning of this sentence: "In fact, they could be even lower considering that they typically rely on a harmonized data set and include a simultaneous calculation of emissions of several species using the same principal activity and technology data." I would presume most country-level inventories are similar in this respect?

RESPONSE: Indeed, this statement is more relevant for the multipollutant inventory (whole GAINS framework) rather than PM alone. We delete this sentence in the final manuscript even though from our experience working with several regional and national inventories we see that often the methods applied to for example PM10 or PM2.5 are not the same as for BC or OC. The latter are often derived using simply shares of PM2.5 rather than absolute emissions factors representative for a given technology. Such shares are often derived from a large set of measurements representing a category of installations rather than a specific one for which PM2.5 emissions were calculated. In that way additional uncertainty is introduced.

REVIEWER:

Would be useful to mention in this section that there is also uncertainty in the speciation fractions (but this is constrained across species since these must sum to <= 1).

RESPONSE: Thank you for pointing it out, we add a sentence highlighting this point on page 35, line 9 [in the original submission] and it reads: "*Allocating total PM emissions into different size bins or chemical species (here BC and OC) is associated with uncertainties that for a specific source are determined by the measurement. Among others, Bond et al. (2013) discussed specific issues related to BC and OC aerosols, while for PM size distribution there exist specific analysis for particular measurement equipment (e.g., Armas et al.,2007; Coquelin et al., 2013) and most of the studies reporting measurements of size distribution estimate uncertainties for each size category. While the sum of all the PM species is constrained by the total mass, the single size distribution values rely on a large number of measurements reducing the overall uncertainty. Exceptions are source-sectors for which very few measurements exist, e.g., coke ovens, fireworks, handling of bulk materials.*"

REVIEWER:
Page 36 In addition to this "Our new global estimate of BC emissions suggests higher numbers than previously published...", perhaps useful to also mention something about BC trends over time here (since there is substantial interest in BC, and it looks like BC trends can be different from PM trends).

RESPONSE: More detailed discussion of this is actually provided in section 4.1 on page 30 of the original manuscript. Here, in conclusions we have added a statement about the different PM2.5 and PM10 trends vs BC and slightly reformulated the concerned paragraph (originally p.36, from line 10) and it reads now: "*We estimate that global emissions of PM have not changed much between 1990 and 2010 but there are significantly different regional trends with North America, Pacific, and Europe reducing emissions by 30 to over 50% and Asia and Africa increasing by about 30%. While these regionally varying developments are clearly visible in PM2.5 and PM10 estimates, the BC regional changes were somewhat less dramatic, mostly because trends in power and industrial sector emissions of PM are much less relevant for total black carbon emissions. Globally, over 75% of anthropogenic PM10 and PM2.5 originates from residential combustion, power plants and industry while for BC residential combustion and transport represent more than 75% but the importance varies across regions with Europe and North America having transport as key and rest of the world residential combustion. Our new global estimate of BC emissions suggests higher numbers than previously published owing primarily to inclusion of new sources.*"

REVIEWER:
Supplement When GAINS values are listed in the tables, these are sometimes listed as ranges. I assume these are not uncertainty ranges (as in some of the other ranges in the table), but are GAINS central values and that the range represents the range used in different GAINS regions? It would be useful to clarify this.

RESPONSE: Indeed, these are ranges representing the spread of values across different regions rather than uncertainty ranges. While this is written for example on page 4 before the Table S2.1, we add a respective comment next to other tables in the SI.

REVIEWER:
I suggest providing a more detailed summary of the emissions data. It would make this data more readily useful to the community to have an electronic file (either csv or excel) that provides emissions of the various species by country/region and by sector and fuel (I appreciate that some aggregation with regard to sector/fuel might be necessary). I realize much of this (or perhaps all of this) would be available on-line, but providing this in a supplement will be more accessible and also provide for an archival record of these important results.

RESPONSE: As indicated in the responses to the initial comments of the reviewer, we have developed an additional set of tables with sectoral emissions (including split across key fuels) for the 25 global regions and all considered PM species over time. This is now included as the MS Excel file in the SI.

**RESPONSES TO Reviewer 2:**

We thank the reviewer for useful comments which have been helpful in improving the manuscript. The responses to specific points raised by the reviewer are provided below.

REVIEWER:
The title mentions that the paper focuses on anthropogenic emissions. However, the paper also discusses open fires. Since the paper is already very long, it might be better to only focus on anthropogenic emissions as stated in the title. The inclusion of emissions from fires (which come from other authors) is a bit confusing.

RESPONSE: Indeed, the paper documents the methodology for PM estimation in GAINS focusing on anthropogenic sources, including also open burning of agricultural waste, but at the same time documents also the complete dataset of ECLIPSE emissions. The latter includes explicit information about what has been used in modelling exercises (see for example Stohl et al., 2015; Eckhardt et al., 2016) and here the reference to the open fires, or more specifically forest and savannah fires is referred to. We allocate one section in the paper to agricultural burning (not all open fires) [see section 3.7] and make explicit references to work on forest and savannah fires on page 13 in the introduction to section 3. We feel that it is justified to provide full documentation of sources used in the entire ECLIPSE set, including forest savannah fires and allocate a page to discuss specific aspect of agricultural burning for which national and regional inputs were used beyond remote sensing data of GFED.

REVIEWER:
Abstract and line 24, page 2: the abstract claims this paper is " the first comprehensive assessment of historical (1990-2010) global anthropogenic particulate matter (PM): : :". However, the EDGAR4.3 inventory described in Crippa et al. (2016) provides emissions for 1970-2010 for PM2.5 and PM10. The statement about being "the first comprehensive assessment" is true for PM1, but not for the other species. Please rephrase.

RESPONSE: As a matter of fact, the word 'comprehensive' is referring here to the comprehensive assessment of several PM species (as well as forming the base for the development of the particulate number inventory referred to in the abstract) within one system assuring that consistent framework is used for the assessment of all of the considered species including PM1, PM2.5, PM10, PMTSP, BC, OC, and OM. But in order to avoid any possible misinterpretation or confusion we simply delete the word 'first' in the abstract as well as in the introduction.

REVIEWER:
Page 18, line 6: " exceptions are old vehicles running on leaded gasoline and preregulation 2-stroke mopeds : : : while latest gasoline direct injection engines have PM mass emissions comparable or even higher than latest diesel engines with particle

filter, however, the absolute level is about one order of magnitude lower than for older generations. This sentence is not clear. What does "absolute level" refer to?

RESPONSE: Thank you for pointing this out. We will rephrase this sentence making clear that the 'absolute level' refers here to the modern (Euro5 and Euro 6) diesel vehicles which have reduced their PM emissions significantly compared the pre or early control stages. Below an example for Italy to illustrate the point with COPERT data for emission factor for PM10 [in g/km]

| g/km | No-control | Advanced controls |
|---|---|---|
| Two-stroke | 0.176 | 0.018 (Euro 3) |
| Light duty gasoline | 0.0024 | 0.0010 (Euro 3 and younger) |
| Light duty diesel | 0.216 | 0.0018 (Euro 5 and younger) |

Interpretation:
=> old 2-strokes are as bad polluters as uncontrolled diesel cars.
Modern diesel cars have reduced their emission rate by a factor 100, such that they are today at the level of or even lower than modern gasoline cars.

The updated manuscript includes a modified sentence reflecting the above explanation. The new sentence in section 3.4 (middle of the 4th paragraph) reads: "*It is important to highlight that properly functioning particulate filters reduce PM emissions significantly and consequently the absolute level of the latest diesel vehicles is about two orders of magnitude lower than for older generations.*"

REVIEWER:
Page 26, line 20: The authors use quite old data for emission factors for agriculture waste burning. Akagi et al. (Atmos. Chem. Phys., 11, 4039–4072, 2011) have published a more recent and detailed review of all data available on emission factors. The authors should indicate why they did not use this more recent review.

RESPONSE: The paper by Akagi et al (2011) is included in the references and in fact we have considered it while comparing and deriving emission factors for this work. Our emission factors derived from several studies listed compare well with the ones provided in the review by Akagi since they mostly refer to the same work already listed in our previous text. We add explicit reference to Akagi paper in the revised text.

REVIEWER:
Page 29, lines 1-4: these lines should be rephrased. Many recent global chemistrytransport and chemistry-climate models now include detailed aerosols schemes, and PMs distribution are calculated as the sum of the mass of all the components included in the models. Maybe a few older models use the "BC + 1.4 OC" formula to calculate the mass of PM, but the recent models are much more advanced and calculate the mass of PMs in a more accurate way.

RESPONSE: Thank you for pointing this out, in fact we would be interested to know which models are these so that we could an example reference. The mentioned paragraph was meant to highlight two elements, the issue of oversimplifying the total carbonaceous mass in PM where often 1.4 ratio was used to convert OC to POM and the fact that the BC+POM often represents total fine anthropogenic PM

in global (not regional) models. Taking into account your comments we have revised one of the sentences in this paragraph that reads now: "*This total fine PM mass has been typically estimated as BC+1.4\*OC and only recently a number of models included more detailed aerosol schemes accounting for varying BC/OC ratios while still largely neglecting the anthropogenic dust component (e.g., Philip et al., 2017)*"

REVIEWER:
Page 31, lines 14-15: The sentence starting with "combined : : :" is unclear

RESPONSE: We have rewritten that sentence, specifically the second part starting with 'combined', and the whole sentence reads now: "*However, as further discussion shows, the largest discrepancy for $PM_{10}$ and $PM_{2.5}$ is for China as well as Europe and Russia; the sum of the differences in these three regions represents about 90% and over 50% of all the difference for $PM_{10}$ and $PM_{2.5}$.*"

REVIEWER:
Page 25 of the supplement: the authors should add in their table the TNO-MACC and TNO-MACCII (Kuenen et al., ACP, 2014) inventories, which provide emissions of PM for Europe and neighboring countries. The TNO-MACC inventories are now becoming a reference for atmospheric modeling in Europe, and these emissions should be mentioned in the paper.

RESPONSE: Thank you for this suggestion. We have included MACCII reference and also added the respective emission estimates to the table

REVIEWER:
Page 25 of the supplement: The emissions provided by US EPA are given as the sum of anthropogenic and wildfires. The dataset provided by EPA (note that the las release of the emissions is 2016 and not 2011 as mentioned in the supplement) provides emissions with and without wildfires. It would be better to include the emissions without wildfires, in order to be consistent with the other data in the table.

RESPONSE: Thank you for suggesting the review of this numbers. We retrieved new numbers from the US EPA website (https://www.epa.gov/air-emissions-inventories/national-emissions-inventory-nei) for both 2011 and 2014. This allows now for a better comparison to our numbers constructing a similar sector set excluding wildfires. The reference to this EPA source is also included in the manuscript.

[revised manuscript text omitted]

**S1 Comparison of temporal distribution patterns**

Fig S1.1 shows a comparison of the temporal patterns (it is an aggregate as the actual patterns are grid specific) for residential combustion sector, applied in the ECLIPSE project, with other data for selected countries.

[Figure]

**Figure S1.1.** Comparison of monthly distribution of emissions used in ECLIPSE with profiles from EDGAR (EC-JRC/PBL, 2010), EMEP (http://emep.int/mscw/), national Finish model FRES (Karvosenoja, 2008), and US EPA.

**S2 Particulate matter emission factors for residential combustion**

The GAINS model distinguishes three principal solid fuel stove categories: *traditional*, *improved* and *new stoves*. *Traditional heating stoves* using wood or coal as fuel have simple grate based firebox designs with usually only primary air supply and no heat storing components. Consequently there is restricted availability of air for combustion and poor mixing of air and pyrolysis gases. *Traditional stoves* in general have very high PM emission factors compared with more advanced technologies, but within this category the variability in the emission factors is also large. For example highest emission factors for traditional wood stoves have been measured in situations with restricted combustion air supply that leads to lower burn rate (Jordan and Seen, 2005). Such conditions might prevail when the user wants a lower heat supply to the room. *Improved stoves* have secondary air supply and heat storing components in the firebox construction that improve the combustion performance and reduce emissions of PM compared with the *traditional stoves*. *New stoves* represent the most advanced stove models on the market that have firebox, construction and airflow characteristics that optimize combustion efficiency. Additionally, an electrostatic precipitator (ESP) can be fitted into the latest stoves, which further improve the PM emission performance. GAINS distinguishes also *wood pellet stoves*. Pellets are a very homogenous fuel and combustion is more optimized than batch fired wood log stoves and thus also the PM emissions are lower than with wood log stoves.

A stove heats the surrounding room, but a boiler heats water to be circulated through a piping system to heat an entire house (Johansson et al., 2004). In *old-type wood log boilers* up-draught combustion is commonly used, which resembles the combustion in a stove; *modern wood boilers*, however, use downdraught combustion and often have an isolated burn-out zone (Johansson et al., 2004). In contrast to stoves, wood boilers can be connected to a water tank to store heat, which allows the boiler to be run at a regular heat output and to certain extent optimizing the combustion conditions. Storage tanks are common in modern wood boilers and also old boilers may be equipped with them, leading to lower emissions and higher efficiencies (Johansson et al., 2004). The single family house boilers are typically smaller than 50 kW$_{th}$, the larger residential boilers are allocated to a category *medium size boilers* where manual and automatic boilers are distinguished (Klimont et al., 2002; Kupiainen and Klimont, 2004, 2007). Such boilers might be an important emission source, especially when many of them are fired with coal, but there are not a of lot measurements available. The GAINS model relies on studies discussed previously (EEA, 2013; Klimont et al., 2002; Kupiainen and Klimont, 2004, 2007) but for a number of countries in Europe updates were made drawing on national information provided within EU consultations (Amann et al., 2015) and recent measurements in China where 100,000s of such installations are used in both residential as well as industrial sector (Wang et al., 2009).

GAINS distinguishes also open fireplaces as a separate category which is of relevance mostly in North America and some European countries, even though in Europe less than 5% of fuelwood would be used in such installations (Klimont et al., 2002; Kupiainen and Klimont, 2004, 2007).

Field Code Changed

Here we summarize the published measurements of emission factors for cooking and heating stoves boilers and compare them to the current ranges of region- and technology-specific GAINS values. The focus is on studies that appeared after the original development of the GAINS particulate matter module (Klimont et al., 2002; Kupiainen and Klimont, 2004, 2007).

5    **Table S2.1:** Summary of PM emission factors for residential wood boilers.

| Emission factors (mg/MJ) | | | | Shares (%) | | | References |
|---|---|---|---|---|---|---|---|
| PM | TC[a] | BC | OC | TC[a] | BC | OC | |
| **wood log** | | | | | | | |
| 1300 (350-2200)[b] | 715 | | | 55 | | | (Boman et al., 2008) old, no accumulator, large fuel charge |
| 120 (73-260)[b] | 60 | | | 50 | | | (Boman et al., 2008) old, no accumulator, adjusted fuel charge |
| 95 (87-100)[b] | 48 | | | 50 | | | (Boman et al., 2008) old, with accumulator |
| 44 (11-450)[b] | 18 | | | 42 | | | (Boman et al., 2008) modern, with accumulator |
| 37 | 27 | 12 | 16 | 75 | 32 | 43 | (Gaegauf et al., 2005), 35 kW apartment house |
| 70-700 | | 20 | 30-335 | | | | GAINS[c], >50 kW, uncontrolled boiler |
| 230-1300 | | 75-200 | 75-600 | | | | GAINS[c], <50 kW, old uncontrolled boiler |
| 80-520 | | 32-50 | 22-230 | | | | GAINS[c], <50 kW, improved |
| 40-260 | | 13-37 | 12-100 | | | | GAINS[c], <50 kW, new/modern |
| **wood chip** | | | | | | | |
| | | | | 44 | 23 | 21 | (Schmidl et al., 2011) 40 kW moving grate, start-up |
| | | | | 5 | 1 | 4 | (Schmidl et al., 2011) 40 kW moving grate, full load |
| | | | | 35 | 33 | 2 | (Schmidl et al., 2011) 40 kW moving grate, part load |
| 85 | 8 | 2 | 6 | 9 | 2 | 7 | (Gaegauf et al., 2005) 70 kW, institute building |
| **wood pellet** | | | | | | | |
| 20 | | 0.1 | 0.9 | | 0.5 | 5 | (Lamberg et al., 2011a) efficient combustion |
| 12 (3-29)[b] | | 0.8 (0-14)[b] | 0.3 (0-3)[b] | | 6 (0-51)[b] | 2 (2-11)[b] | (Lamberg et al., 2011b), 25 kW, nominal load |
| 16 | | 1 | 0.1 | | | | (Tissari et al., 2008), 20 kW, nominal load |
| 24 | | 3 | 0.2 | | | | (Tissari et al., 2008), 20 kW, partial load |
| 49 | 35 | 24 | 11 | 72 | 49 | 23 | (Gaegauf et al., 2005) 10-32 kW, apartment house |
| 8-25 | | 0.8-1 | 0.4-1 | | | | GAINS[c], >50 kW |
| 20-68 | | 5 | 2.5-10 | | | | GAINS[c], <50 kW |

[a] Total Carbon (TC)
[b] (min-max)
[c] PM value refers to PM2.5

**Table S2.2:** Summary of PM emission factors for residential heating wood stoves.

| Emission factors (mg/MJ) | | | Shares (%) | | Reference |
|---|---|---|---|---|---|
| PM | BC | OC | BC | OC | |
| **traditional** | | | | | |
| 673-1373 | 24-72 | 263-623 | 2-7 | 39-53 | (Alves et al., 2011) |
| 300-1400 | - | - | 2-9 | 35-50 | (Gonçalves et al., 2011) incl. cold start |
| 90-900 | - | - | 2-9 | 35-48 | (Gonçalves et al., 2011) incl. hot start |
| 750-1060 | - | - | - | - | (Jordan and Seen, 2005), full airflow |
| 1560-1700 | - | - | - | - | (Jordan and Seen, 2005), half airflow |
| 1870-3000 | - | - | - | - | (Jordan and Seen, 2005), closed airflow |
| 128-400 | 20 | 157 | 8 | 64 | (McDonald et al., 2000) |
| - | 39-43 | 70-390 | 5-14 | 47-67 | Studies in Kupiainen& Klimont (2007) |
| 150[a] - 930 (2400)[b] | 32[a] - 100 | 60[a] - 435 (1200)[b] | 4-22 | 41-50 | GAINS; the PM value represents PM2.5 |
| **improved** | | | | | |
| 22-180 | - | - | - | - | (Boman et al., 2008) |
| 86-105 | 9-11 | 52-58 | - | - | (Fine et al., 2004) |
| 130 | 88 | 39 | 68 | 30 | (Gaegauf et al., 2005) |
| 60-160 | - | - | 11-37 | 20-43 | (Gonçalves et al., 2010) |
| 75-97 | 15-28 | 17-35 | 24-32 | 27-39 | (Schmidl et al., 2011) |
| 38-350 | - | - | - | - | (Pettersson et al., 2011) |
| - | 56-79 | 11-16 | - | - | Studies in Kupiainen& Klimont (2007) |
| 55[a] - 372 | 30[a] - 95 | 11[a] -133 | 25-55 | 19-35 | GAINS; the PM value represents PM2.5 |
| **new** | | | | | |
| 67-122 | 13-15 | 43-67 | - | - | (Fine et al., 2004), catalytic |
| 72-89 | 21-33 | 16-32 | 30-37 | 22-36 | (Schmidl et al., 2011) |
| 30[a] - 186 | 9[a] - 30 | 8[a] - 67 | 18-30 | 28-35 | GAINS; the PM value represents PM2.5 |
| **pellet** | | | | | |
| 10-66 | - | - | - | - | (Boman et al., 2008) |
| 15-47 | - | - | - | - | (Boman et al., 2011) |
| 17 | 0.7 | - | 4 | - | (Frey et al., 2014) |
| 20 | 0.1 | 0.9 | 0.5 | 5 | (Lamberg et al., 2011b) |
| 3-29 | 0-14 | 0.1-3 | 0-51 | 2-11 | (Lamberg et al., 2011a) |
| - | - | - | 14 | 11 | (Schmidl et al., 2011) |
| 47-129 | 0.5-1.3 | 0.3-5.2 | 1-2 | 1-9 | (Sippula et al., 2007) |
| 10[a] - 47 | 1.3[a] - 4 | 2[a] - 7 | 10-17 | 12-17 | GAINS; the PM value represents PM2.5 |

[a] The lowest values represent Swiss data

[b] Norwegian wood stove

**Table S2.3:** Summary of PM emission factors for cookstoves using biofuels.

| Emission factors (mg/MJ) | | | References |
|---|---|---|---|
| PM | BC | OC | |
| traditional | | | |
| 530 | 44 | 250 | (Just et al., 2013) |
| 106 | 50 | 44 | (Roden et al., 2009), 3-stone, lab measurements |
| 515 (300-1000) [a] | 83 (10-210) [a] | 254 (90-660) [a] | (Roden et al., 2009), Honduras, field measurements |
| 510 (280-510) [b] | 65-75 (40-75) [b] | 229 (125-229) [b] | GAINS [c] |
| improved | | | |
| 150 | 80 | 20 | (Just et al., 2013), rocket stove |
| 270 (100-500) [a] | | | (Li et al., 2009), improved stoves, PM2.5 |
| 394 (120-700) [a] | 102 (6-325) [a] | 208 (60-460) [a] | (Roden et al., 2009), improved no chimney, field measurements |
| 205 (105-270) [b] | 50-75 (27-75) [b] | 63 (31-68) [b] | GAINS [c] |
| new | | | |
| 255 (40-720) [a] | 116 (6-660) [a] | 93 (33-370) [a] | (Roden et al., 2009), improved with chimney, field measurements |
| 56-102 | 11-21 | 19-34 | GAINS [c] |
| fan assisted | | | |
| 86 (25-125) [a] | 33 (6-100) [a] | 38 (4-71) [a] | (Roden et al., 2009), fan assisted, lab measurements |
| 54 | 33 | 14 | (Just et al., 2013), gasifier with fan |
| 17 | 4 | 9 | GAINS [c] |

[a] (min-max)

[b] central value for fuelwood and in brackets the whole range including also dung and agricultural residues

[c] the PM value represents PM2.5

**Table S2.4:** Summary of PM emission factors for coal cooking and heating stoves

| Emission factors (mg/MJ) | | | References |
|---|---|---|---|
| PM | BC | OC | |
| traditional | | | |
| 805 (214-1360)[a] | 250 (11-540)[a] | 400 (116-710)[a] | (Zhi et al., 2009), portable stove, bituminous coals |
| | 332 (10-610)[a] | 472 (129-822)[a] | (Chen et al., 2009), simple low-efficiency stove without chimney, bituminous coals |
| 351 | 135 | 108 | GAINS [b] (cooking) |
| 315-495 | 90-220 | 160-200 | GAINS [b] (heating) |
| improved | | | |
| | 466 (6-1377)[a] | 248 (35-551)[a] | (Chen et al., 2009), high-efficiency stove with chimney |
| 492 | 183 | 200 | (Zhang et al., 2008), steel stove, brown coal |
| 36 | 1 | 16 | (Zhang et al., 2008), steel stove, bituminous coal |
| 408 (155-685)[a] | 40 (2-140)[a] | 230 (78-470)[a] | (Zhi et al., 2009), bituminous coals |
| 246 | 132 | 60 | GAINS [b] (cooking) |
| 315-350 | 82-200 | 88-112 | GAINS [b] (heating) |
| new | | | |
| 270 | 23 | 96 | (Li et al., 2016), average for bituminous coals |
| 176 | 108 | 32 | GAINS [b] (cooking) |
| 158-248 | 73-176 | 48-60 | GAINS [b] (heating) |
| briquettes | | | |
| | 16 (2-33)[a] | 329 (71-668)[a] | (Chen et al., 2009), simple low-efficiency, no chimney |
| | 4 (0.5-9)[a] | 219 (27-423)[a] | (Chen et al., 2009), high-efficiency, with chimney |
| 184 | 3 | 80 | (Zhang et al., 2008), steel stove |
| 440 (98-930)[a] | 12 (2-23)[a] | 233 (67-460)[a] | (Zhi et al., 2009), traditional portable stove |
| 202 (90-346)[a] | 2 (0.5-6)[a] | 124 (36-217)[a] | (Zhi et al., 2009), improved stove with chimney |
| 17 | 0.4 | 6.5 | (Li et al., 2016), semi-coke briquettes |
| 23-135 | 0.3-1 | 9-55 | GAINS [b] |

[a] (min-max)

[b] the PM value represents PM2.5

**S3 Summary of particulate matter emissions factors for diesel generators**

Note that the ranges presented for GAINS represent the spread across GAINS regions or technologies (if a category refers to an aggregate across several measures) defined in the GAINS model.

**Table S3.1:** Summary of PM emission factors for diesel generator sets

| Emission factors (mg/MJ) | | | Shares (%) | | Reference |
|---|---|---|---|---|---|
| PM | BC | OC | BC | OC | |
| 69-189 | | | | | Uma et al. (2004), 10 kW (higher value), 40 kW (lower value) |
| 139 | | | 66% | | Bond et al. (2004) |
| 13/22 | | | | | Gilmore et al. (2006), ICE 10 kW, with/without DPF |
| | | 116-585 | | | Watson et al. (2006) [a] |
| 59-190 | 12-54 | 30-120 | 31% | 51% | Shah et al. (2007) [b] 300 kW 1985 Detroit Diesel V92, 2-str |
| 45-219 | 30-145 | 8-56 | 67% | 21% | Shah et al. (2007) [b] 350 kW 2000 Cat 3406C, 4-str |
| 22-143 | 10-80 | 6-37 | 53% | 25% | Shah et al. (2007) [b] 300 kW 1985 Detroit Diesel V92, 2-str, DOC |
| 59-203 | 28-145 | 4-16 | 67% | 8% | Shah et al. (2007) [b] 350 kW 2000 Cat 3406C, 4-str, DOC |
| 23-190 | 9-96 | 10-81 | 49% | 36% | Shah et al. (2007) [b] 300 kW 1985 Detroit Diesel V92, 2-str, DOC+FBC |
| 4-26 | 2.5-19 | 1-3 | 76% | 15% | Shah et al. (2007) [b] 350 kW 2000 Cat 3406C, 4-str, passive-DPF |
| 1-3 | 0.8-2 | 1-6 | 67% | 49% | Shah et al. (2007) [b] 350 kW 2000 Cat 3406C, 4-str, active-DPF |
| | | | | 20-70% | Watson et al. (2008) |
| 14-42 | | | | | Zhu et al. (2009) [c] |
| 174-433 | | | | | Tsai et al. (2010) [d] |
| 55 | | | | | Anayochukwu et al. (2013) |
| GAINS emission factors; the PM value represents PM2.5 | | | | | |
| 96 | 40 | 28 | 41% | 29% | No control |
| 48-64 | 20-26 | 14-19 | 41% | 29% | Controlled, no DPF |
| <1-3 | 0.5-2 | 0.3-0.8 | | | Controlled, with DPF |

[a] Higher value with 10% load and lower value with 100% load for a 100 kW DG set
[b] Lower value with 100% load and higher value with 10% load, share of BC/OC is average of all loads
[c] Average of 14 military diesel generators with rated capacities of 10, 30, 60, and 100 kW under different load conditions. The fleet average EFs are 1.2+/-0.6 g/kg for PM.
[d] Higher value with no load and lower value with 10 kW

**S4 Transport sector**

Note that the ranges presented for GAINS represent the spread across GAINS regions or technologies (if a category refers to an aggregate across several measures) defined in the GAINS model.

**Table S4.1:** Comparison of selected measured emissions factors and ranges used in the GAINS model for diesel and gasoline cars and light duty vehicles.

| | Emission factors (mg/MJ) | | | Reference |
|---|---|---|---|---|
| | PM | BC | OC | |
| Diesel | | | | |
| Pre-/early regulation | 44-67 | 9-17 | 13-34 | (Subramanian et al., 2009) |
| Euro 1 | 67 | 17 | 13 | (Subramanian et al., 2009) |
| Euro 2 | 30-33 | 7-16 | 8-12 | (Cheung et al., 2009; Subramanian et al., 2009) |
| Euro 3 | 10-29 | | | (Graham, 2005) |
| Euro 4 | 6-11 | 3-8 | 1-2 | (Cheung et al., 2009; Geller et al., 2006) |
| Euro 4 with DPF | 0.2-0.3 | 0.02-0.1 | 0.02-0.06 | (Dwyer et al., 2010; Louis et al., 2016) |
| Pre-/early regulation | 56-133 | 38-76 | 21-51 | GAINS [a] |
| Euro 1 | 22-50 | 16-35 | 5-11 | GAINS [a]; for developing countries the values only marginally lower than pre/early regulation |
| Euro 2 | 15-40 | 12-32 | 3-6 | GAINS [a]; for developing countries the values only marginally lower than pre/early regulation |
| Euro 3 | 11-29 | 10-22 | 1-2 | GAINS [a] |
| Euro 4 | 5-20 | 4-17 | 0.5-1.6 | GAINS [a] |
| Euro 4 with DPF | 0.5-1 | 0.1-0.3 | 0.1-0.6 | GAINS [a] |
| Gasoline | | | | |
| Pre-/early regulation | 4-10 | 0.5-2 | 2-10 | see studies in Kupiainen and Klimont (2004, 2007) |
| Euro 1, 2 | 1-4 | 0.6-1.5 | 0.3-1.6 | see studies in Kupiainen and Klimont (2004, 2007) |
| Euro 3 | 0.2-2 | 0.01-0.2 | 0.2-0.6 | (Cheung et al., 2009; Geller et al., 2006; Graham, 2005) |
| Euro 4 | | 0.001-0.4 | | (Louis et al., 2016) |
| Pre-/early regulation | 6 | 1 | 3-4 | GAINS [a] |
| Euro 1, 2 | 1-4 | 0.2-1 | 0.3-1.7 | GAINS [a] |
| Euro 3, 4 | 0.3-1.1 | 0.05-0.5 | 0.1-0.4 | GAINS [a] |

[a] the PM value represents PM2.5

**Table S4.2:** Comparison of selected measured emissions factors and ranges used in the GAINS model for diesel heavy duty vehicles

| | Emission factors (mg/MJ) | | | Reference |
|---|---|---|---|---|
| | PM | BC | OC | |
| Diesel heavy duty trucks | | | | |
| Pre-/early regulation | 28-33 | | | (Herner et al., 2009; Yanowitz et al., 2000) |
| | 44-244 | 4-50 | 15-122 | (Subramanian et al., 2009), Bangkok, Thailand |
| | 30-50 | | | (Liu et al., 2009), on-road measurements in China |
| Euro I | 11 | | | (Yanowitz et al., 2000) |
| | 22 | 4 | 9 | (Subramanian et al., 2009), Bangkok, Thailand |
| | 10-20 | | | (Liu et al., 2009), on-road measurements in China |
| Euro II | 22-44 | 2-9 | 7-22 | (Subramanian et al., 2009), Bangkok, Thailand |
| | 7-17 | 16 | | (Liu et al., 2009; Wang et al., 2011), on-road measurements in China |
| Euro III | 3-7 | 9 | | (Liu et al., 2009; Wang et al., 2011), on-road measurements in China |
| Euro IV | | 4 | | (Wang et al., 2011), on-road measurements in China |
| Pre-/early regulation | 34-107 | 17-53 | 10-37 | GAINS [a] |
| Euro I | 21-71 | 17-53 | 6-19 | GAINS [a] |
| Euro II | 11-44 | 7-30 | 2-10 | GAINS [a] |
| Euro III | 10-27 | 8-25 | 2-7 | GAINS [a] |
| Euro IV, V | 2-7 | 2-5 | 0.3-1 | GAINS [a] |
| Euro VI | 0.1-0.4 | 0.01-0.06 | 0.06-0.15 | GAINS [a] |

[a] the PM value represents PM2.5

**Table S4.3:** Comparison of selected measured emissions factors and ranges used in the GAINS model for non-road machinery.

| | Emission factors (mg/MJ) | | | Reference |
|---|---|---|---|---|
| | PM | BC | OC | |
| Diesel locomotives | | | | |
| Pre-/early regulation | 49-67 | | | (Dincer and Elbir, 2007; Johnson et al., 2013; Tang et al., 2015) |
| Regulated | 20-40 | 20 | | (Dincer and Elbir, 2007; Galvis et al., 2013; Johnson et al., 2013; Tang et al., 2015) |
| | 30 | 14 | | (Galvis et al., 2013) |
| | 20 | 15 | | (Jaffe et al., 2014) |
| | 37 | 21 | | (Krasowsky et al., 2015) |
| pre-regulated | 49-98 | 24-45 | 12-25 | GAINS [a] |
| regulated (stage I) | 26-49 | 11-22 | 6-12 | GAINS [a] |
| Agriculture | | | | |
| Pre-regulation | 141 | 58 | 41 | (Kupiainen and Klimont, 2007) |
| | 89 | 49 | | (EEA, 2013) |
| Stage I | 20-39 | 16-21 | | (EEA, 2013) |
| Stage II | 15 | 11.5 | | (EEA, 2013) |
| Pre-regulation | 100-170 | 41-70 | 29-50 | GAINS [a] |
| Stage I | 57-96 | 23-40 | 16-27 | GAINS [a] |
| Stage II, III | 27-43 | 10-19 | 8-12 | GAINS [a] |
| Stage IV,V | 6-10 | 0.7-1.2 | 0.5-0.8 | GAINS [a] |
| Construction | | | | |
| Pre-regulation | 140 | 65 | 30 | (Kupiainen and Klimont, 2007) |
| | 103 | 56 | | (EEA, 2013) |
| Stage I | 85 | 47 | | (EEA, 2013) |
| Pre-regulation | 95-140 | 46-68 | 21-31 | GAINS [a] |
| Stage I | 57-76 | 26-39 | 12-18 | GAINS [a] |
| Stage II, III | 24-36 | 12-17 | 5-8 | GAINS [a] |
| Stage IV,V | 6-8 | 0.8-1.2 | 0.4-0.6 | GAINS [a] |

[a] the PM value represents PM2.5

**Table S4.4:** Comparison of selected measured emissions factors and ranges used in the GAINS model for 2-wheelers.

| | Emission factors (mg/MJ) | | | References |
|---|---|---|---|---|
| | PM | BC | OC | |
| **2-stroke** | | | | |
| Euro 0 mopeds | 250 (198-295) | | | (Spezzano et al., 2008), hot start |
| | 160 (121-878) | | | (Spezzano et al., 2008), cold start |
| Euro 1 mopeds | 169 (102-235) | | | (Spezzano et al., 2008), hot start |
| | 42 (26-71) | | | (Spezzano et al., 2008), cold start |
| Euro 2 mopeds | 147-217 | | | (Spezzano et al., 2008), hot start |
| | 13-215 | | | (Spezzano et al., 2008), cold start |
| CNG rickshaw, Delhi, India | 124-160 | | | (Grieshop et al., 2012) |
| Euro 0 mopeds | 132-1400 | 10-75 | 90-1015 | GAINS [a] |
| Euro 1 mopeds | 12-450 | 7-49 | 40-300 | GAINS [a] |
| Euro 2 mopeds | 37-280 | 6-45 | 23-172 | GAINS [a] |
| Euro 3 mopeds | 14-112 | 3-30 | 8-61 | GAINS [a] |
| **4-stroke** | | | | |
| Motorcycles | 2.6-3.7 | | | (Yang et al., 2005), cold start |
| Euro 0 motorcycles | 4 | | | (Spezzano et al., 2007) |
| Euro 1 motorcycles | 2 | | | (Spezzano et al., 2007) |
| Rickshaw, Delhi, India | 30-45 | | | (Grieshop et al., 2012) |
| CNG rickshaw, Delhi, India | 12-13 | | | (Grieshop et al., 2012) |
| Euro 0 motorcycles | 6-14 | 1-2 | 3-9 | GAINS [a] |
| Euro 1 motorcycles | 5-12 | 1-2 | 2-7 | GAINS [a] |
| Euro 2 motorcycles | 3-5 | 0.5-0.8 | 0.4-1.7 | GAINS [a] |
| Euro 3 motorcycles | 2-3 | 0.5-0.75 | 0.3-1.4 | GAINS [a] |

[a] the PM value represents PM2.5

**Table S4.5:** Summary of PM emission factor ranges used in the GAINS model for non-exhaust transport sources

| | Emission factors (mg/km) | | | |
|---|---|---|---|---|
| | $PM_{10}$ | $PM_{2.5}$ | BC | OC |
| Brake wear | | | | |
| Cars | 3.5 – 12 | 2.5 – 5 | 0.05 – 0.12 | 0.8 – 2.2 |
| Light duty vehicles | 3.5 – 19 | 2.5 – 8 | 0.05 – 0.2 | 0.8 – 3.5 |
| Heavy duty vehicles | 21 – 53 | 13 – 21 | 0.25 – 0.5 | 5 – 17 |
| Tyre wear | | | | |
| Cars | 1.5 – 9 | 0.15 – 0.7 | 0.2 – 1 | 0.5 - 2.4 |
| Light duty vehicles | 2.5 – 7 | 0.2 – 0.7 | 0.35 – 1 | 0.85 – 2.4 |
| Heavy duty vehicles | 40 – 47 | 4.2 – 4.7 | 6 – 7 | 15 – 17 |
| Road abrasion | | | | |
| Cars & Light duty vehicles | 7 - 10 | 3 – 5 | 0.15 – 0.6 | 0.7 – 1 |
| | 30 – 140 [a] | 20 – 80 [a] | 0.2 – 1.5 [a] | 4 – 14 [a] |
| Heavy duty vehicles | 38 – 50 | 18 – 27 | 0.7 – 1 | 3 – 5 |

[a] vehicles with studded tires; variation between estimates for Scandinavian and alpine countries

**S5 Industry**

GAINS model PM emission factors (as used for the ECLIPSE *V5a*) for brick making compared with values used in GAINS previously (UNEP/WMO, 2011) and recent set of measurements on typical kilns used in South Asia (Weyant et al., 2014).

5    **Table S5.1:** Comparison of emissions factors used in the GAINS model for brick kilns with selected other studies.

| Emission factors (g kg$^{-1}$ brick) | | | References |
|---|---|---|---|
| PM2.5 | BC | OC | |
| Clamp kiln | | | |
| 1.6 | 0.35 | 0.3 | (UNEP/WMO, 2011)[a] |
| 1 | 0.3 | 0.1 | GAINS (Asia) |
| 1 | 0.35 | 0.15 | GAINS (Latin America and Africa) |
| Downdraft kiln | | | |
| 0.49 | 0.19 | 0.07 | (Weyant et al., 2014) |
| 0.97 | 0.29 | 0.09 | GAINS (all regions) |
| Bull's trench kiln (BTK) | | | |
| 1.31 | 0.27 | 0.24 | (UNEP/WMO, 2011)[a] |
| 0.19 (0.08-0.33) | 0.15 (0.09-0.27) | 0.007 | (Weyant et al., 2014)[b] |
| 0.18/0.8 | 0.13/0.25 | 0.01/0.07 | GAINS (Asia); fixed /moving chimney |
| Vertical shaft brick kiln (VSBK) | | | |
| 0.77 | 0.175 | 0.15 | (UNEP/WMO, 2011)[a] |
| 0.07 (0.005-0.009) | 0.0015 (0.001-0.002) | 0.014 | (Weyant et al., 2014)[b] |
| 0.093 | 0.001-0.004 | 0.002-0.059 | GAINS (Asia) |
| 0.093 | 0.002 | 0.059 | GAINS (Latin America and Africa) |
| Zig-zag kiln | | | |
| 0.06 (0.03-0.06) | 0.01 (0.014-0.03) | 0.005 | (Weyant et al., 2014)[b] |
| 0.13 | 0.04 | 0.02 | GAINS (Asia) |
| Tunnel kiln (coal) | | | |
| 0.28 | 0.0035 | 0.003 | (UNEP/WMO, 2011)[a] |
| 0.24 | 0.001 | <0.00 | (Weyant et al., 2014) |
| 0.18 | 0.002 | 0.0035 | GAINS (all regions) |
| Hoffman kiln | | | |
| 0.08 | 0.003 | 0.005 | GAINS (all regions) |
| Marquez kiln (MK) | | | |
| 0.15 | 0.06 | 0.02 | GAINS (Latin America) |

[a] Previous version of the GAINS model was used
[b] Central value and ranges of average values; all measurement data provided in the original study

Brick sector production structure in Asia has been analysed in a number of studies addressing either the whole region where selected countries, typically key producers including China, India, Pakistan, Bangladesh, Vietnam, are discussed (AIT, 2003; BASIN, 1999; FAO, 1993; Heierli and Maithel, 2008; Maithel, 2014) or focusing on particular countries like China (Zhang, 1997), India (BASIN, 1998; Maithel et al., 2012; Verma and Uppal, 2013), Bangladesh (Croitoru and Sarraf, 2012; Guttikunda et al., 2013; World Bank, 2011), Cambodia (Rozemuller, 1999), Afghanistan (Samuel Hall Consulting, 2011), Nepal (Heierli et al., 2007). More recently, a number of development programs and local air pollution studies focused on this sector in the Latin America and Caribbean regions, including some where information about kiln structure was collected (Bellprat, 2009; EELA, 2011; Erbe, 2011; PRAL, 2012; Stratus Consulting, 2014; SwissContact, 2014a). Fewer assessments exist for Africa (Scott, 2013; SwissContact, 2014c). The updated and country specific data for Latin America and Caribbean (LAC) is included only in version *V5a* of ECLIPSE since the previous versions included just five regions for the whole LAC; Argentina, Brazil, Chile, Mexico, other LAC.

GAINS activity data has been built on the basis of several regional studies where production, energy efficiency, and sector structure were discussed, i.e., Asia (AIT, 2003; Co et al., 2009; Croitoru and Sarraf, 2012; FAO, 1993; Guttikunda et al., 2013; Heierli et al., 2007; Heierli and Maithel, 2008; Maithel, 2014; Maithel et al., 2012; Samuel Hall Consulting, 2011; Subrahmanya, 2006; Verma and Uppal, 2013; World Bank, 2011; Zhang, 1997), Africa (Alam, 2006; Scott, 2013; SwissContact, 2014c), Latin America and Caribbean (Bellprat, 2009; EELA, 2011; PRAL, 2012; Stratus Consulting, 2014; SwissContact, 2014b). For several countries where we found no regional analysis, the United Nations data on 'building bricks, made of clay' was used (http://unstats.un.org/unsd/industry/commoditylist2.asp). There are some differences between different versions of the ECLIPSE datasets; specifically during the development of the *V5a* version, the data for all countries in Latin America and Caribbean was revisited and updated, and a new version of the UN statistics was downloaded.

**Table S5.2:** Brick production in key regions; GAINS model assumptions - ECLIPSE *V5a*, Tg bricks year$^{-1}$

|  | 1990 | 1995 | 2000 | 2005 | 2010 |
|---|---|---|---|---|---|
| Global | 1542 | 2357 | 2688 | 3022 | 3574 |
| Asia | 1314 | 2130 | 2530 | 2819 | 3320 |
| *of which:* |  |  |  |  |  |
| *China* | 1050 | 1800 | 2106 | 2204 | 2508 |
| *India* | 131 | 178 | 254 | 406 | 553 |
| *Vietnam* | 20 | 20 | 27 | 46 | 65 |
| *Bangladesh* | 9 | 15 | 18 | 17 | 25 |
| *Pakistan* | 32 | 41 | 50 | 59 | 74 |
| *Other Asia* | 71 | 75 | 76 | 87 | 95 |
| Africa | 18 | 18 | 15 | 17 | 22 |
| Europe | 158 | 156 | 72 | 82 | 79 |
| Latin America and Caribbean | 29 | 30 | 43 | 75 | 127 |
| Other | 23 | 23 | 27 | 29 | 25 |

**S6 Emissions of PM species over time in ECLIPSE datasets**

The Fig S6.1 shows emissions of PM10, PM2.5, BC, and OC calculated with the GAINS model within different versions of the ECLIPSE dataset. These have been created between 2013 and 2015 and include a number of updates to activity data and emission factors; the methodology remained the same. The changes for PM10 and PM2.5 are similar, driven by updates of

5    activity data, i.e., the energy statistics from IEA were reimported for the whole time series for the version *V5* and *V5a* and for China the regional coal statistics were used. Control strategies have been updated continuously considering more up to date information available over time. Additionally, in version *V5a* Latin America and Caribbean were revised since higher spatial resolution was introduced in the GAINS model. Several of the above mentioned updates affected also emissions of BC and OC but the largest impact on the BC emissions was due to introduction of emissions from kerosene lamps which

10    were not specifically distinguished in *V4a*; this represents the key component of the higher emissions in *V5*, *V5a*. For OC the change is in the opposite direction and *V5a* has significantly lower emissions than previous versions which is due to update of the OC emission factor for residential cooking in Asia and Africa.

[Figure]

15    **Figure S6.1.** Global emissions of PM (excluding international shipping and open biomass burning) in the period 1990-2010 in different ECLIPSE scenarios; unit [Tg year$^{-1}$]

**Table S6.2:** Global anthropogenic (excluding international shipping & aviation) emissions of PM10 in ECLIPSE V5a; [Gg year$^{-1}$]

| Region | 1990 | 1995 | 2000 | 2005 | 2010 |
|---|---|---|---|---|---|
| 1 Canada | 333 | 315 | 345 | 337 | 334 |
| 2 USA | 2416 | 2158 | 1954 | 1920 | 1630 |
| 3 Mexico | 643 | 621 | 653 | 574 | 572 |
| 4 Rest Central America | 454 | 455 | 479 | 498 | 516 |
| 5 Brazil | 1228 | 1295 | 1250 | 1385 | 1456 |
| 6 Rest South America | 1018 | 1155 | 1138 | 1131 | 1192 |
| 7 Northern Africa | 1022 | 1152 | 1355 | 1144 | 1194 |
| 8 Other Africa | 4393 | 4993 | 5831 | 6425 | 7150 |
| 10 South Africa | 682 | 738 | 747 | 848 | 818 |
| 11 Western Europe | 3294 | 2458 | 2031 | 1747 | 1577 |
| 12 Central Europe | 2944 | 1608 | 1236 | 1046 | 1038 |
| 13 Turkey | 1007 | 756 | 525 | 477 | 571 |
| 14 Ukraine+ | 1854 | 856 | 679 | 707 | 680 |
| 15 Asia-Stan | 836 | 325 | 303 | 314 | 392 |
| 16 Russia+ | 5833 | 2434 | 2314 | 2316 | 2161 |
| 17 Middle East | 836 | 954 | 1055 | 962 | 996 |
| 18 India | 7828 | 8785 | 8654 | 7952 | 8061 |
| 19 Korea | 1227 | 913 | 844 | 816 | 768 |
| 20 China+ | 14057 | 17612 | 18205 | 21230 | 21976 |
| 21 Southeastern Asia | 2291 | 2855 | 2783 | 2451 | 2526 |
| 22 Indonesia+ | 1383 | 1576 | 1673 | 1768 | 1902 |
| 23 Japan | 545 | 435 | 354 | 319 | 267 |
| 24 Oceania | 295 | 303 | 354 | 354 | 342 |
| 25 Rest South Asia | 1695 | 1894 | 2211 | 2349 | 2533 |
| **Global** | **58112** | **56646** | **56974** | **59071** | **60651** |

**Table S6.3:** Global anthropogenic (excluding international shipping & aviation) emissions of PM2.5 in ECLIPSE V5a; [Gg year$^{-1}$]

| Region | 1990 | 1995 | 2000 | 2005 | 2010 |
|---|---|---|---|---|---|
| 1 Canada | 252 | 244 | 250 | 242 | 241 |
| 2 USA | 1629 | 1482 | 1296 | 1275 | 1027 |
| 3 Mexico | 495 | 498 | 526 | 459 | 454 |
| 4 Rest Central America | 395 | 394 | 416 | 428 | 446 |
| 5 Brazil | 938 | 974 | 933 | 1054 | 1098 |
| 6 Rest South America | 825 | 933 | 909 | 901 | 949 |
| 7 Northern Africa | 762 | 852 | 982 | 847 | 909 |
| 8 Other Africa | 4056 | 4606 | 5308 | 5887 | 6575 |
| 10 South Africa | 408 | 444 | 431 | 501 | 490 |
| 11 Western Europe | 2125 | 1700 | 1360 | 1157 | 1037 |
| 12 Central Europe | 1610 | 1020 | 843 | 752 | 775 |
| 13 Turkey | 585 | 480 | 388 | 356 | 425 |
| 14 Ukraine+ | 1072 | 531 | 464 | 483 | 455 |
| 15 Asia-Stan | 562 | 222 | 211 | 222 | 283 |
| 16 Russia+ | 3702 | 1614 | 1530 | 1495 | 1413 |
| 17 Middle East | 686 | 778 | 845 | 784 | 794 |
| 18 India | 5768 | 6453 | 6472 | 5957 | 6032 |
| 19 Korea | 784 | 600 | 547 | 565 | 529 |
| 20 China+ | 10863 | 13072 | 13633 | 15673 | 16096 |
| 21 Southeastern Asia | 1878 | 2257 | 2198 | 1974 | 2012 |
| 22 Indonesia+ | 1230 | 1371 | 1447 | 1510 | 1604 |
| 23 Japan | 337 | 295 | 236 | 203 | 160 |
| 24 Oceania | 188 | 193 | 210 | 201 | 188 |
| 25 Rest South Asia | 1455 | 1629 | 1859 | 1962 | 2065 |
| **Global** | **42606** | **42640** | **43294** | **44888** | **46055** |

**Table S6.4:** Global anthropogenic (excluding international shipping & aviation) emissions of PM1 in ECLIPSE V5a; [Gg year$^{-1}$]

| Region | 1990 | 1995 | 2000 | 2005 | 2010 |
|---|---|---|---|---|---|
| 1 Canada | 184 | 195 | 196 | 187 | 190 |
| 2 USA | 1163 | 1095 | 949 | 930 | 727 |
| 3 Mexico | 375 | 378 | 395 | 361 | 357 |
| 4 Rest Central America | 329 | 331 | 353 | 366 | 390 |
| 5 Brazil | 706 | 720 | 718 | 819 | 846 |
| 6 Rest South America | 657 | 732 | 708 | 712 | 764 |
| 7 Northern Africa | 447 | 476 | 514 | 485 | 542 |
| 8 Other Africa | 3724 | 4213 | 4838 | 5416 | 6064 |
| 10 South Africa | 285 | 309 | 307 | 354 | 354 |
| 11 Western Europe | 1397 | 1171 | 966 | 834 | 751 |
| 12 Central Europe | 894 | 667 | 619 | 579 | 607 |
| 13 Turkey | 386 | 341 | 286 | 263 | 311 |
| 14 Ukraine+ | 565 | 325 | 279 | 278 | 261 |
| 15 Asia-Stan | 292 | 154 | 146 | 154 | 198 |
| 16 Russia+ | 1988 | 1078 | 1011 | 936 | 852 |
| 17 Middle East | 501 | 562 | 596 | 614 | 615 |
| 18 India | 4500 | 4992 | 5016 | 4700 | 5031 |
| 19 Korea | 635 | 510 | 450 | 464 | 429 |
| 20 China+ | 9153 | 11251 | 11731 | 12473 | 11606 |
| 21 Southeastern Asia | 1800 | 2204 | 2093 | 1791 | 1803 |
| 22 Indonesia+ | 1135 | 1254 | 1315 | 1373 | 1453 |
| 23 Japan | 258 | 229 | 157 | 126 | 87 |
| 24 Oceania | 140 | 146 | 155 | 143 | 133 |
| 25 Rest South Asia | 1303 | 1445 | 1625 | 1714 | 1811 |
| **Global** | **32816** | **34780** | **35422** | **36073** | **36180** |

**Table S6.5:** Global anthropogenic (excluding international shipping & aviation) emissions of BC in ECLIPSE V5a; [Gg year$^{-1}$]

| Region | 1990 | 1995 | 2000 | 2005 | 2010 |
|---|---|---|---|---|---|
| 1 Canada | 44 | 49 | 51 | 49 | 49 |
| 2 USA | 311 | 291 | 281 | 279 | 201 |
| 3 Mexico | 76 | 77 | 82 | 84 | 88 |
| 4 Rest Central America | 52 | 54 | 61 | 65 | 71 |
| 5 Brazil | 143 | 148 | 160 | 171 | 179 |
| 6 Rest South America | 115 | 135 | 140 | 150 | 169 |
| 7 Northern Africa | 127 | 120 | 117 | 121 | 140 |
| 8 Other Africa | 752 | 836 | 942 | 1030 | 1135 |
| 10 South Africa | 57 | 59 | 57 | 74 | 72 |
| 11 Western Europe | 331 | 335 | 307 | 287 | 246 |
| 12 Central Europe | 126 | 112 | 112 | 121 | 134 |
| 13 Turkey | 60 | 59 | 53 | 51 | 67 |
| 14 Ukraine+ | 88 | 59 | 45 | 41 | 36 |
| 15 Asia-Stan | 50 | 28 | 33 | 38 | 55 |
| 16 Russia+ | 439 | 251 | 238 | 226 | 177 |
| 17 Middle East | 174 | 183 | 210 | 243 | 262 |
| 18 India | 853 | 931 | 884 | 908 | 1022 |
| 19 Korea | 135 | 84 | 71 | 84 | 74 |
| 20 China+ | 1348 | 1347 | 1655 | 1823 | 1924 |
| 21 Southeastern Asia | 300 | 299 | 304 | 328 | 333 |
| 22 Indonesia+ | 243 | 260 | 275 | 279 | 290 |
| 23 Japan | 67 | 74 | 66 | 50 | 29 |
| 24 Oceania | 30 | 32 | 35 | 35 | 33 |
| 25 Rest South Asia | 288 | 304 | 325 | 337 | 348 |
| **Global** | **6210** | **6129** | **6505** | **6872** | **7134** |

**Table S6.6:** Global anthropogenic (excluding international shipping & aviation) emissions of OC in ECLIPSE V5a; [Gg year$^{-1}$]

| Region | 1990 | 1995 | 2000 | 2005 | 2010 |
|---|---|---|---|---|---|
| 1 Canada | 72 | 77 | 77 | 72 | 74 |
| 2 USA | 448 | 434 | 388 | 379 | 308 |
| 3 Mexico | 162 | 162 | 164 | 158 | 155 |
| 4 Rest Central America | 144 | 149 | 159 | 169 | 181 |
| 5 Brazil | 251 | 258 | 275 | 311 | 314 |
| 6 Rest South America | 297 | 329 | 315 | 324 | 344 |
| 7 Northern Africa | 145 | 150 | 155 | 166 | 192 |
| 8 Other Africa | 1627 | 1842 | 2124 | 2408 | 2701 |
| 10 South Africa | 101 | 108 | 110 | 129 | 130 |
| 11 Western Europe | 495 | 422 | 343 | 284 | 253 |
| 12 Central Europe | 224 | 201 | 217 | 220 | 234 |
| 13 Turkey | 114 | 108 | 95 | 88 | 107 |
| 14 Ukraine+ | 149 | 102 | 82 | 77 | 72 |
| 15 Asia-Stan | 90 | 66 | 62 | 64 | 86 |
| 16 Russia+ | 509 | 332 | 304 | 256 | 231 |
| 17 Middle East | 190 | 217 | 220 | 237 | 229 |
| 18 India | 1530 | 1623 | 1596 | 1630 | 1755 |
| 19 Korea | 200 | 157 | 147 | 157 | 148 |
| 20 China+ | 3147 | 3264 | 3500 | 3564 | 3599 |
| 21 Southeastern Asia | 526 | 548 | 567 | 598 | 632 |
| 22 Indonesia+ | 431 | 473 | 514 | 551 | 595 |
| 23 Japan | 51 | 54 | 49 | 40 | 29 |
| 24 Oceania | 52 | 55 | 57 | 51 | 46 |
| 25 Rest South Asia | 502 | 562 | 628 | 680 | 726 |
| **Global** | **11456** | **11695** | **12150** | **12610** | **13140** |

The following charts in this section show emissions of PM species by key sectors for seven global regions defined as in Figure 3 in the main paper. These charts are the same as Figure 4 in the paper where global emissions are shown. Note that the scale varies across sectors and also across regions. For all regions, except Pacific, the units are million tons [Tg year$^{-1}$]; for Pacific they are [Gg year$^{-1}$].

[Figure]

**Figure S6.2.** Emissions of PM species by key sectors in *East Asia* in the period 1990-2010 [Tg year$^{-1}$]; ECLIPSE *V5a*

[Figure]

**Figure S6.3.** Emissions of PM species by key sectors in *Other Asia* in the period 1990-2010 [Tg year$^{-1}$]; ECLIPSE *V5a*

[Figure]

**Figure S6.4.** Emissions of PM species by key sectors in *Africa* in the period 1990-2010 [Tg year$^{-1}$]; ECLIPSE *V5a*

[Figure]

**Figure S6.5.** Emissions of PM species by key sectors in *Latin America and Caribbean* in the period 1990-2010 [Tg year$^{-1}$]; ECLIPSE *V5a*

[Figure]

**Figure S6.6.** Emissions of PM species by key sectors in *Europe and Russia* in the period 1990-2010 [Tg year$^{-1}$]; ECLIPSE *V5a*

[Figure]

**Figure S6.7.** Emissions of PM species by key sectors in *North America* in the period 1990-2010 [Tg year$^{-1}$]; ECLIPSE *V5a*

[Figure]

**Figure S6.8.** Emissions of PM species by key sectors in *Pacific* in the period 1990-2010 [Gg year$^{-1}$]; ECLIPSE *V5a*

**S7 Regional resolution**

The spatial resolution of the GAINS model is discussed section 2.4 of the paper and the list of all 170 regions can be obtained from the online model. In principle, GAINS distinguishes single countries in Europe (exception in Russia for which European and Asian part is included separately) North America, Australia and New Zealand, for Asia several larger

5 countries are divided into provinces or states (larger administrative units in, e.g., China, India, Indonesia, Japan, etc.) while Middle East represented as one region or (most recent versions) distinguishes Iran, Saudi Arabia, Israel, and the rest of Middle East. Africa is divided into four regions: South Africa, Egypt, North Africa, and other Africa. Latin America and Caribbean includes now 13 regions with all larger countries treated separately while Central America as well as Caribbean states are grouped in two regions. While such resolution of 170 regions is used for the calculation of emissions, the

10 presentation of data and results differs between the on-line models available for specific world regions, e.g., for Europe and Asia the full resolution is available, while in the global model application (http://magcat.iiasa.ac.at/gains/IAM/index.login) the data and results are presented for 25 regions (Fig. S7.1). This follows closely the IMAGE model[1] resolution; often used or compatible with several global integrated assessment models.

[Figure]

| Canada | Northern Africa | Ukraine + | China + |
| USA | Other Africa | Asia – Stan | Southeastern Asia |
| Mexico | South Africa | Russia + | Indonesia + |
| Rest Central America | Western Europe | Middle East | Japan |
| Brazil | Central Europe | India | Oceania |
| Rest South America | Turkey | Korea | Rest South Asia |

15 **Figure S7.1.** Regions distinguished in the global GAINS online application.
* * *
[1] http://themasites.pbl.nl/models/image/index.php/Region_classification_map

**S8 Sectoral resolution**

**Table S8.1:** Source sector resolution in the GAINS model for calculation of PM emissions

| Key source category | Source sectors | Fuel category or activity type |
|---|---|---|
| *Energy sector* | | |
| | Power plants (distinguishing small, large, old, new plants); Diesel generators; | Coal, oil, gas, biomass, waste |
| | Extraction and distribution of solid and liquid fuels (fugitive as well as combustion from gas flaring) | Coal, oil |
| | Briquette production | Production |
| *Residential combustion* | | |
| | Cooking stoves; Heating (distinguishing fireplaces, stoves, house boilers, mid-size residential boilers) | Coal, fuelwood, dung, oil, gas, agricultural residues, charcoal |
| | Kerosene lighting | Kerosene |
| | Waste (trash) burning | Waste |
| *Industrial combustion* | | |
| | Iron and Steel; Pulp and Paper; Chemical; Non-ferrous metals; Non-metallic minerals (excl. Bricks); Other | Coal, oil, gas, biomass, waste |
| *Industrial processes* | | |
| | Iron and steel industry divided into: Pig iron; Coke ovens; Agglomeration plants – pellets; Agglomeration plants – sinter; Open hearth; Electric Arc; Basic oxygen; Rolling mills; Cast Iron | Production |
| | Non-ferrous metals (copper and nickel smelters); Primary aluminium; Secondary aluminium; Cement; Lime; Carbon black production; Glass production; Mineral fertilizer production; Brick manufacturing; Pulp and paper | Production |
| | Refineries | Crude oil throughput |
| | Handling and storage of bulk industrial and agricultural products (fugitive) | Million tons of products |
| *Road transport* | | |
| | Passenger cars and vans; Light duty vehicles; Heavy duty vehicles; Busses; Motorcycles (4-stroke); Mopeds (2-stroke) | Gasoline, diesel, CNG, LPG, km driven (for calculation of non-exhaust emissions) |
| *Non-road transport* | | |
| | Agricultural and forestry; Construction and mining; Railways; Inland navigation; Coastal shipping; Aviation (landing and take-off); 2-stroke engines (e.g., in household, forestry, etc.); Other land based machinery | Diesel, gasoline, CNG, jet fuel and kerosene, heavy fuel oil, coal |
| *Agriculture* | | |
| | Arable land operations | Arable land area |
| | Livestock housing | Cattle, pigs, poultry |
| | Open burning of agricultural waste | Waste burned |
| *Other* | | |
| | Fireworks; Cigarette smoking; Barbeques; Cremation | Population |
| | Construction (fugitive) | Constructed area |

**S8 Comparison of regional estimates with selected studies**

The table S8.1 provides ECLIPSE *V5a* PM estimates for selected regions and years (from the period 2000-2010) and compares them with selected regional peer-reviewed studies.

5    **Table S8.1:** Comparison of regional estimates for anthropogenic [a] emissions of PM species, Gg year[-1]

| Region – (Source) – Year | PM10 | PM2.5 | PM1 | BC | OC |
|---|---|---|---|---|---|
| **Global** | | | | | |
| *This study – 1995* | 57830 | 43762 | 35902 | 6206 | 11949 |
| (Bond et al., 2004) - 1996 | | | | 4997 | 10481 |
| *This study - 2000* | 58366 | 44613 | 36741 | 6595 | 12449 |
| (Bond et al., 2013) - 2000 | | | | 4870 | |
| *This study - 2010* | 62537 | 47843 | 37819 | 7264 | 13548 |
| HTAP_v2 (Janssens-Maenhout et al., 2015) - 2010 | 50292 | 32761 | | 5525 | 13581 |
| **China** | | | | | |
| *This study - 2000* | 18061 | 13554 | 11685 | 1646 | 3487 |
| (Cao et al., 2006) - 2000 | | | | 1496 | 4211 |
| (Streets et al., 2003) - 2000 | | | | 1049 | 3385 |
| (Klimont et al., 2009) - 2000 | | | | 1345 | 3205 |
| (Lu et al., 2011) - 2000 | | | | 1244 | 2823 |
| (Ohara et al., 2007) - 2000 | | | | 1093 | 2563 |
| (Bond et al., 2013) - 2000 | | | | 1200 [b] | 2800 [b] |
| (Zhang et al., 2006) - 2001 | 17120 | 12100 | | | |
| *This study - 2005* | 21087 | 15593 | 12428 | 1813 | 3552 |
| (Zhang et al., 2009) - 2006 | 18223 | 13266 | | 1811 | 3217 |
| (Klimont et al., 2009) - 2005 | | | | 1366 | 2812 |
| *This study - 2010* | 21827 | 16019 | 11564 | 1915 | 3589 |
| (Lu et al., 2011) - 2010 | | | | 1838 | 3907 |
| (Kurokawa et al., 2013) - 2008 | 21606 | 14514 | | 1589 | 3081 |
| (Guan et al., 2014) - 2010 | | 12100 | | | |
| HTAP_v2 (Janssens-Maenhout et al., 2015) - 2010 | 16615 | 12199 | | 1764 | 3384 |
| (Kondo et al., 2011) - 2008 | | | | 1940 | |
| **India** | | | | | |
| *This study - 2000* | 8654 | 6472 | 5016 | 884 | 1596 |
| (Streets et al., 2003) - 2000 | | | | 600 | 2837 |
| (Ohara et al., 2007) - 2000 | | | | 795 | 3268 |

| Region – (Source) – Year | PM10 | PM2.5 | PM1 | BC | OC |
|---|---|---|---|---|---|
| (Klimont et al., 2009) - 2000 | | | | 842 | 1887 |
| (Lu et al., 2011) - 2000 | | | | 736 | 1990 |
| (Bond et al., 2013) - /2000 | | | | 500 [b] | 1600 [b] |
| (Reddy and Venkataraman, 2002a, 2002b) - 1998-99 | | 4300 | | 380 | 1250 |
| *This study - 2005* | 7952 | 5957 | 4700 | 908 | 1630 |
| (Zhang et al., 2009) - 2006 | 4002 | 3111 | | 344 | 888 |
| (Klimont et al., 2009) - 2005 | | | | 1029 | 2132 |
| *This study - 2010* | 8061 | 6032 | 5091 | 1022 | 1755 |
| (Lu et al., 2011) - 2010 | | | | 996 | 2582 |
| HTAP_v2 (Janssens-Maenhout et al., 2015) - 2010 | 8280 | 6230 | | 1019 | 2530 |
| (Kurokawa et al., 2013) - 2008 | 6651 | 4884 | | 713 | 2286 |
| Europe [c] | | | | | |
| *This study - 1995* | 6905 | 4584 | 3071 | 675 | 1021 |
| (Kupiainen and Klimont, 2007) - 1995 | | | | 717 | 1053 |
| (Schaap et al., 2004) - 1995 | | | | 760 | |
| (Bond et al., 2004) - 1996 | | | | 678 | 947 |
| *This study - 2000* | 5579 | 3843 | 2668 | 618 | 910 |
| (Kupiainen and Klimont, 2007) - 2000 | | | | 680 | 996 |
| (Kupiainen and Klimont, 2004) - 2000 | | | 2772 | 672 | 988 |
| *This study - 2010* | 5008 | 3471 | 2393 | 562 | 806 |
| TNO-MACCII (Kuenen et al., 2014) - 2009 | 4694 | 3199 | | 548 | 906 |
| HTAP_v2 (Janssens-Maenhout et al., 2015) [d] - 2010 | 2951 | 2133 | | 382 | 638 |
| LRTAP reporting (www.ceip.at) - 2010 | 4784 | 3250 | | | |
| Russian Federation | | | | | |
| *This study - 2010* | 2108 | 1368 | 815 | 170 | 213 |
| HTAP_v2 (Janssens-Maenhout et al., 2015) - 2010 | 562 | 313 | | 60 | 42 |
| (Huang et al., 2015) - 2010 | | | | 224 | |
| Russian Federation – European part only | | | | | |
| *This study - 2010* | 1090 | 734 | 427 | 71 | 122 |
| LRTAP reporting (www.ceip.at) - 2010 | 569 | 367 | | | |
| US | | | | | |
| *This study - 2000* | 1954 | 1296 | 949 | 289 | 388 |
| (Battye et al., 2002) - 1999 | | | | 430 | |
| (Reff et al., 2009) - 2000 | | | | 440 | 960 |
| (Bond et al., 2013) - 2000 | | | | 350 [b] | 500 [b] |

Field Code Changed

Field Code Changed

| Region – (Source) – Year | PM10 | PM2.5 | PM1 | BC | OC |
|---|---|---|---|---|---|
| *This study - 2010* | 1630 | 1027 | 727 | 201 | 308 |
| HTAP_v2 (Janssens-Maenhout et al., 2015) - 2010 | 1973 | 1640 | | 295 | 471 |
| (US EPA, 2016) [e] - 2011 | 2847 | 1909 | | | |
| (US EPA, 2016) [e] - 2014 | 2830 | 1875 | | 280 | 602 |

[a] Based on the information available in the quoted studies, all presented estimates exclude forest fires but include agricultural burning, unless stated otherwise; [b] Excluding agricultural burning; [c] Includes European part of Russian Federation (except HTAP_v2); [d] Excluding any territories of Russian Federation; [e]  [f] Excluding wildfires and prescribed burning, unpaved roads, and construction dust